# A virtual hydrological framework for evaluation of stochastic rainfall models

Bree Bennett[1], Mark Thyer[1], Michael Leonard[1], Martin Lambert[1], Bryson Bates[2]

[1]School of Civil, Environmental and Mining Engineering, University of Adelaide, North Terrace Campus, 5005, South Australia

[2] School of Agriculture and Environment, The University of Western Australia, Crawley, 6009, Western Australia

*Correspondence to*: Bree Bennett (bree.bennett@adelaide.edu.au)

**Abstract.** Stochastic rainfall modelling is a commonly used technique for evaluating the impact of flooding, drought or climate change in a catchment. While considerable attention has been given to the development of stochastic rainfall models (SRMs), significantly less attention has been paid to developing methods to evaluate their performance. Typical evaluation methods employ a wide range of rainfall statistics. However, they give limited understanding about which rainfall statistical characteristics are most important for reliable streamflow prediction. To address this issue a formal evaluation framework is introduced, with three key features: (i) streamflow-based — to give a direct evaluation of modelled streamflow performance, (ii) virtual — to avoid the issue of confounding errors in hydrological models or data, and (iii) targeted — to isolate the source of errors according to specific sites and seasons. The virtual hydrologic evaluation framework uses two types of tests, integrated tests and unit tests, to attribute deficiencies that impact on streamflow to their original source in the SRM according to site and season. The framework is applied to a case study of 22 sites in South Australia with a strong seasonal cycle. In this case study, the framework demonstrated the surprising result that apparently 'good' modelled rainfall can produce 'poor' streamflow predictions, whilst 'poor' modelled rainfall may lead to 'good' streamflow predictions. This is due to the representation of highly seasonally catchment processes within the hydrological model that can dampen or amplify rainfall errors when converted to streamflow. The framework identified the importance of rainfall in the 'wetting-up' months (months where the rainfall is high but streamflow low) of the annual hydrologic cycle (May and June in this case study) for providing reliable predictions of streamflow over the entire year despite their low monthly flow volume. This insight would not have been found using existing methods and highlights the importance of the virtual hydrological evaluation framework for SRM evaluation.

## 1    Introduction

Stochastic rainfall model (SRM) simulations are primarily used as inputs to a hydrological model, for simulating realisations of streamflow. Streamflow simulations are then used to assess hydrological risks, such as floods (e.g. Camici et al., 2011, Li et al., 2016) or droughts (e.g. Henley et al., 2013, Paton et al., 2013, Mortazavi-Naeini et al., 2015). When evaluating the efficacy of SRM's, current approaches that make comparisons to observed rainfall or streamflow have limited diagnostic

ability. They are unable to make a targeted evaluation of the SRM's ability to reproduce streamflow characteristics of practical interest. This paper introduces a new virtual framework that enables targeted hydrological evaluation of SRMs.

Observed-rainfall evaluation is the most common method for SRM evaluation (Rasmussen, 2013, Wilks, 2008, Baxevani and Lennartsson, 2015, Srikanthan and Pegram, 2009, Evin et al., 2018, Bennett et al., 2018). As shown in Fig. 1(a) it involves comparisons between observed and simulated rainfall typically using a large number of evaluation statistics. Often, this method shows 'mixed' performance where many statistics are reproduced well, but some are poor. While these assessments are useful, a drawback is that it is difficult to ascertain if the rainfall model's performance is sufficient in terms of predictions of practical interest, which are typically streamflow-based. This means it is unclear if it is necessary to invest time and effort to address instances of poor performance, when the majority of statistics are well reproduced (Bennett et al., 2018, Evin et al., 2018).

To overcome limitations in observed-rainfall evaluation methods, the conventional alternative is to evaluate the rainfall model's performance in terms of streamflow (e.g. Camici et al., 2011, Blazkova and Beven, 2002, 2009, McMillan and Brasington, 2008) and is referred to as 'observed-streamflow evaluation'. From Fig. 1(b), observed-streamflow evaluation typically involves (1) a SRM that produces simulations of rainfall, that are (2) input to a hydrological model to produce simulated streamflow, which is (3) converted to the predictions of interest (e.g. the flood frequency distribution), and (4) compared against the observed streamflow predictions of interest. A challenge with observed-streamflow evaluation is that when there is poor predictive performance (i.e. a significant discrepancy between the observed and predicted streamflow) it is difficult to ascertain if the poor performance was caused by the hydrological model or the SRM. Hydrological model predictive performance can vary substantially from catchment to catchment due to data errors (rainfall or streamflow) and model structural errors (Evin et al., 2014, Coxon et al., 2015, Andreassian et al., 2001, Kuczera and Williams, 1992, Renard et al., 2011, McInerney et al., 2017) which makes it difficult to evaluate the performance of the SRM and identify opportunities for improvement.

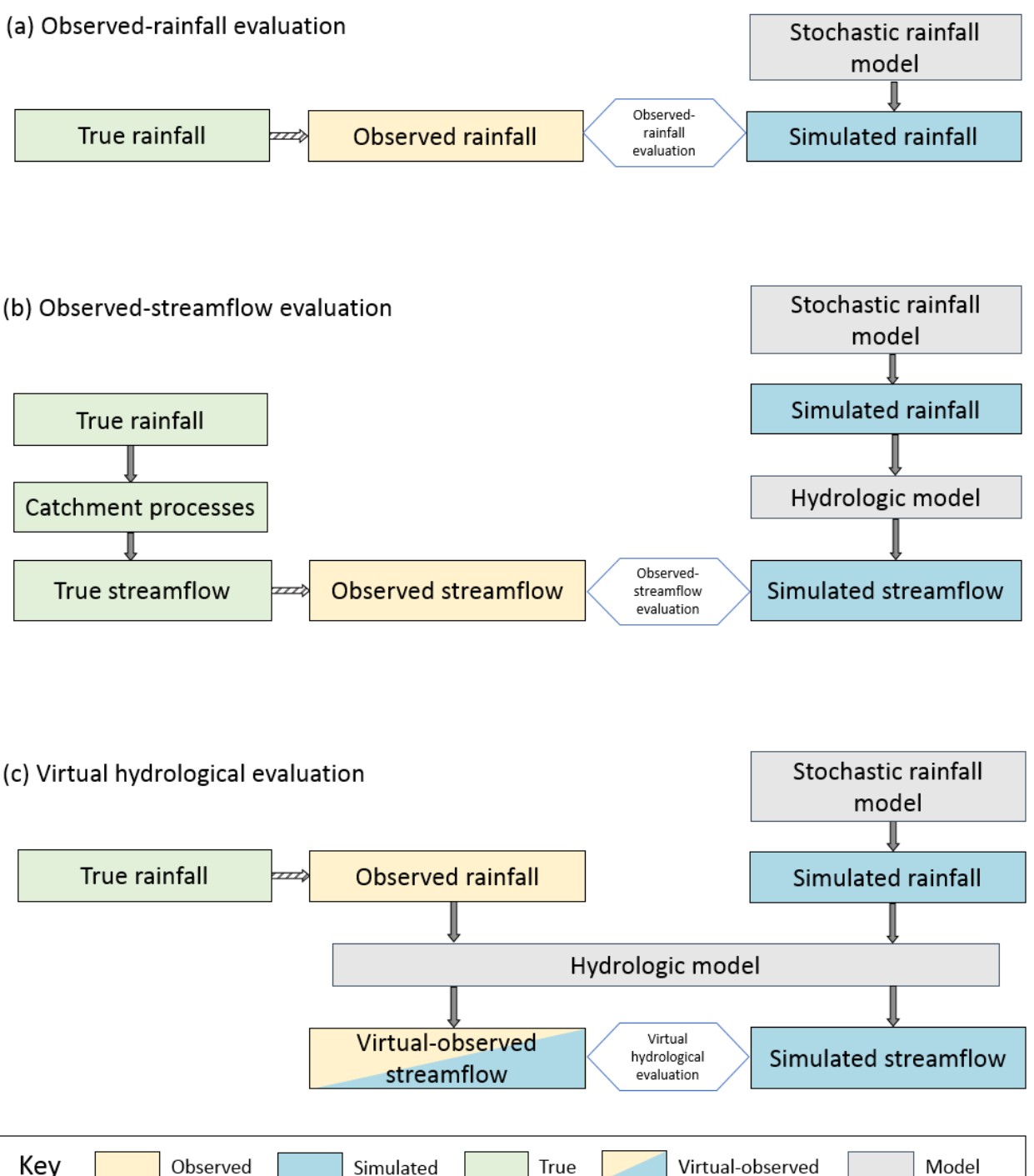

**Fig. 1** Schematic of (a) observed-rainfall evaluation where simulated rainfall is compared against observed rainfall and (b) observed-streamflow evaluation where simulated streamflow is compared against observed streamflow (c) virtual hydrological evaluation framework where simulated streamflow is compared against virtual-observed streamflow.

The focus of this paper is the development and application of a virtual hydrological evaluation framework for streamflow-based evaluation of SRMs. As shown in Fig. 1(c), a virtual hydrological evaluation involves the comparison of simulated streamflow statistics (produced by the hydrological model by inputting simulated rainfall from a SRM) against virtual-observed streamflow statistics (produced by the hydrological model by inputting observed rainfall). This framework is designed to focus on streamflow predictions of interest, similar to observed-streamflow evaluation, but to reduce the sources of error to only those introduced by the SRM. To illustrate this, Table 1 gives an overview of the sources of error for the three evaluation frameworks and indicates whether the evaluations are streamflow-based. The observed-rainfall evaluation framework is used to identify errors in the SRM, but is not able to determine their implications for streamflow. The observed-streamflow framework provides an absolute measure of performance, since ultimately the goal is to match streamflow observations or statistics. However, with this approach it is not possible to readily identify whether discrepancies in the simulated streamflow are attributed to the SRM, the streamflow observations or poor process representation within the hydrological model. In contrast, the virtual hydrological evaluation framework is a relative measure of performance, where the hydrological model is a common factor in the production of simulated streamflow and virtual-observed streamflow that is used as a baseline for comparison. By using a virtual baseline, observed streamflow is not directly required in the evaluation as both simulated and observed rainfall undergo transformation by the same process representation (i.e. the hydrological model). This enables discrepancies in the streamflow to be identified in terms of features of the SRM.

**Table 1 Comparison of the sources of error for observed-rainfall, observed-streamflow and virtual hydrological evaluation frameworks as well as whether the evaluation is streamflow-based.**

|  | Source of error | | | Streamflow-based evaluation |
|---|---|---|---|---|
|  | **Stochastic rainfall model** | **Hydrological model** | **Observed streamflow** | |
| **Observed-rainfall evaluation** | Yes | No | No | No |
| **Observed-streamflow evaluation** | Yes | Yes | Yes | Yes |
| **Virtual hydrological evaluation** | Yes | No | No | Yes |

To date, 'virtual experiments', that is, experiments that focus on comparisons between streamflow simulated under different conditions or inputs (i.e. virtual streamflow) without relying on comparisons to observed streamflow, have been used in a variety of contexts. Examples include (i) the evaluation of hydrological model sensitivity (e.g. Ball, 1994, Nicótina et al., 2008, Paschalis et al., 2013, Shah et al., 1996, Wilson et al., 1979) including the identification of rainfall features of interest in terms of hydrological behaviour (e.g. Sikorska et al., 2018), (ii) the development of new techniques for flood frequency analysis (e.g. Li et al., 2014, 2016), and (iii) the calibration, validation and selection of SRMs (e.g. Müller and Haberlandt, 2018, Kim and Olivera, 2011).

The framework presented in this paper is a significant advance from previously reported virtual experiments because it presents a formal framework to identify key deficiencies in the SRM by (1) extending the comprehensive and systematic evaluation (CASE) framework (developed by Bennett et al., 2018 for observed-rainfall evaluation and used by Evin et al., 2018, Khedhaouiria et al., 2018) that systematically categorises performance at multiple spatial and temporal scales using quantitative criteria for each statistic for use in virtual hydrological evaluations, and (2) utilising two types of virtual experiments that are able to identify the source of key deficiencies in SRM at specific locations and time periods.

The key objectives of this paper are:

1. To introduce a formalised framework for the virtual hydrological evaluation of SRMs: the new framework is a stepwise procedure that enables the identification poor performing sites, then poor performing time periods and then the key deficiencies in the SRM for those sites and time periods by drawing on the systematic application of quantitative performance criteria.

2. To present two different tests which are part of the framework: the integrated test and a new type of test, the unit test. Combined use of these tests allows streamflow discrepancies to be attributed to their original source in the SRM according to site and season.

3. To demonstrate the framework evaluation on a SRM and contrast the outcomes with conventional evaluation methods.

The virtual hydrologic evaluation framework is explained in Section 2 with the procedures for the integrated test and unit test outlined in Sections 2.2.2 and 2.4.1. SRMs have been developed for 22 sites in the Onkaparinga catchment, South Australia (Section 3) and are used to illustrate the procedure (Section 4). Discussion and conclusions emphasize the features of the framework and the different recommendations it can identify for improving the rainfall model (Sections 5 and 6).

## 2 Virtual hydrological evaluation framework

### 2.1 Overview

A virtual hydrological evaluation involves the comparison of simulated streamflow statistics to virtual-observed streamflow statistics (Fig. 1(c)), defined as:

- Simulated streamflow — is streamflow produced by the hydrological model by inputting simulated rainfall at a given site.
- Virtual-observed streamflow — is streamflow produced by the hydrological model by inputting observed rainfall at the same given site.

The virtual framework undertakes a relative assessment of the simulated and observed rainfall after its transformation by the same hydrological model to provide insight into the performance of the SRM. Because the hydrological evaluation is a relative comparison of the observed and simulated rainfall, it is important that all other model parameters and extraneous variables

(e.g. potential evapotranspiration) relating to the hydrological model are kept the same for the simulation of both virtual-observed and simulated streamflow. It is also important that the selected hydrological model is fit for purpose so that it can simulate the streamflow characteristics of interest.

The virtual hydrological evaluation framework is best used to augment and complement existing evaluation methods, rather than act as a replacement. The three evaluation frameworks could work together as follows, where: (i) observed-rainfall evaluation identifies any deficiencies in the SRM prior to any hydrological considerations; (ii) the virtual hydrological framework identifies which of these rainfall deficiencies impact on the key predictions of interest, that is, simulated streamflow; and (iii) observed-streamflow evaluation provides a final validation. Therefore, together they enable a more focused approach to identify opportunities for improvement of a SRM. This is because the ultimate goal of the SRM modelling process remains the same: to match observed streamflow for a catchment of interest.

The formal implementation of the virtual hydrological evaluation framework is summarised in Fig. 2. It uses a series of steps to identify poor performing sites, then poor performing time periods and then the key deficiencies in the SRM for those sites and time periods. It combines both observed rainfall-evaluation and virtual hydrological evaluation. The virtual hydrological evaluation includes two different types of tests, an 'integrated test' that isolates issues for a given site, and 'unit tests' that isolate issues for specific time periods. This enables the diagnosis of the key deficiencies in the simulated rainfall. The following sections explain the three steps in turn.

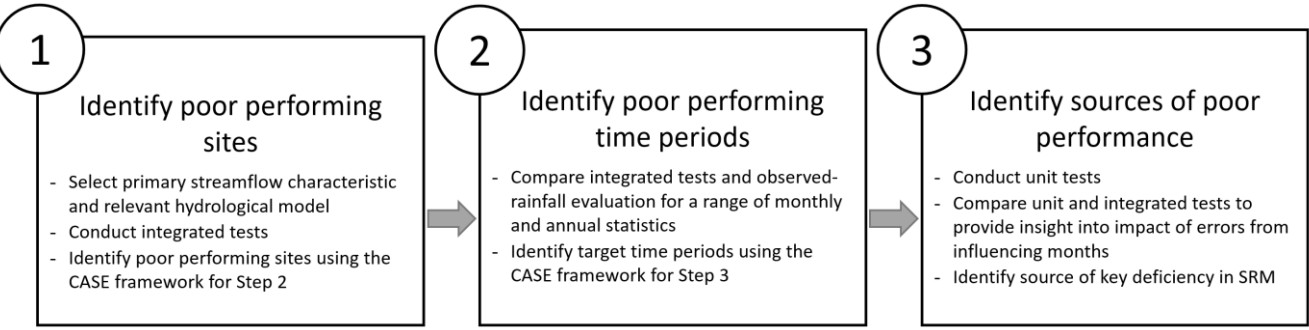

**Fig. 2 Virtual hydrological evaluation procedure.**

## 2.2     Step 1 – Identify poor performing sites

The first step focuses on using integrated tests to identify poor performing sites for further evaluation. Following the selection of a primary streamflow characteristic of interest and a suitable hydrological model, integrated tests are conducted for each rainfall site (described below in Section 2.2.2). The results of the integrated tests are then used to identify sites that are poor performing, according to the systematic application of quantitative performance criteria (see Section 2.2.3), for the primary streamflow characteristic.

### 2.2.1 Selection of primary streamflow characteristic and relevant hydrological model

In order to undertake an integrated test an appropriate hydrological model is required to simulate the streamflow. The hydrological model should be selected on the basis that it is capable of simulating streamflow for the timescales, magnitudes and physical processes of interest to the intended application. For example, a capability for simulating flow volumes is important for yield. A streamflow characteristic of interest, herein termed the 'primary streamflow characteristic', is then selected to enable a method for filtering sites and concentrating the investigation of the rainfall model on sites that perform poorly in terms of its intended application. For example, the distribution of annual total flow would be a suitable characteristic when investigating yield. Following the identification of the hydrological model and primary streamflow characteristic an integrated test is conducted for each rainfall site, which serves as an overall test of the SRM's performance.

### 2.2.2 Integrated test procedure

The integrated test proceeds for a single site by transforming the time series of observed and simulated rainfall, via the hydrological model (Fig. 1(c)). Consider the time series of observed, $R^{obs}$ daily rainfall for each year at a given site. This rainfall time series is transformed according to a hydrological model $g[\ ]$ to produce the virtual-observed streamflow, denoted as $Q^{vo}$ and '...' are additional inputs (e.g. potential evapotranspiration).

$$Q^{vo} = g[R^{obs}, ...] \tag{1}$$

Likewise, all replicates of the simulated, $R^{sim}$ daily rainfall for each year at a given site are transformed according the hydrological model $g[\ ]$ to produce simulated streamflow replicates, $Q^{sim}$.

$$Q^{sim} = g[R^{sim}, ...] \tag{2}$$

If there is a discrepancy between the simulated streamflow, $Q^{sim}$, and the virtual-observed streamflow, $Q^{vo}$, distributions, this indicates that there is a deficiency in the simulated rainfall for that site.

### 2.2.3 Identify poor performing sites using CASE framework

The integrated test results aim to identify the sites that are poor performing for the primary streamflow characteristic. Model performance is categorised using a CASE framework approach as 'good', 'fair' or 'poor' following Bennett et al. (2018). The quantitative tests for each performance category are provided in Table 2 alongside an illustration of each in Fig. 3. The quantitative tests proceed by comparing the statistics of the virtual-observed streamflow against those calculated from replicates of the simulated streamflow. Performance was categorised as 'good' if the selected statistic for the virtual-observed streamflow fell within the 90% limits of the statistic calculated from the simulated streamflow replicates (Fig. 3, case i), as 'fair' if the virtual-observed statistic fell outside the 90% limits of the simulated streamflow replicates but within the 99.7% limits (Fig. 3, case ii) and otherwise as 'poor' (Fig. 3, case iii).

**Table 2 CASE performance classification criteria. Adapted from Bennett et al. (2018).**

| Performance Classification | Test | Key |
|---|---|---|
| **'good'** | Observation lies within the 90% limits (case i) | 🟩 |
| **'fair'** | Observation lies outside the 90% limits but within the 99.7% limits (case ii) | 🟨 |
| **'poor'** | Otherwise (case iii) | 🟥 |

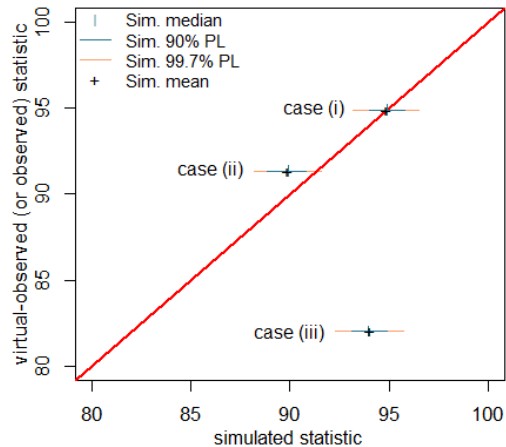

Fig. 3 Illustration of performance classification, case (i) shows 'good' performance, case (ii) shows 'fair' performance and case (iii) shows 'poor' performance. Adapted from Bennett et al. (2018).

## 2.3    Step 2 – Identify poor performing time periods

The second framework step is to identify poor performing time periods by conducting a detailed analysis of the integrated test results and comparing these results with an observed-rainfall evaluation at the monthly scale.

Evaluating monthly total flows is a valuable test of rainfall model performance as the production of monthly total flow volumes relies on the integration of many daily rainfall characteristics (amount, duration, persistence). For each of the poor performing sites, each of these statistics for each month are categorized into 'good', 'fair' and 'poor' using the CASE framework. See Section 2.2.3 for further explanation of the categorization procedure. This enables the identification of poor-performing time periods from the perspective of the virtual hydrological evaluation.

Poor performance in reproducing virtual-observed streamflow is then contrasted against an observed-rainfall evaluation so that specific poor-performing time periods can be identified for further investigation in Step 3. By contrasting CASE performance categories ('good', 'fair' and 'poor') for observed-rainfall evaluation against virtual-observation streamflow evaluation, poor performing time periods from both rainfall and streamflow perspectives can be identified. This comparison between the observed-rainfall evaluation and the virtual hydrological evaluation (integrated test) can be summarised graphically (e.g. see Fig. 7, Section 4.2).

## 2.4 Step 3 – Identify sources of poor performance

The third step of the framework is to identify sources of poor performance in streamflow according to deficiencies in the simulated rainfall. Step 2 identifies the poor performing time periods from a streamflow perspective. However, due to catchment 'memory', the poor performance in streamflow could be due to deficiencies in the simulated rainfall from a range of potential influencing months during or prior to the poor performing time period. For example poor streamflow performance in an evaluated month maybe due to the influence of: (i) rainfall deficiencies mostly in the same month (i.e. concurrent influencing months), (ii) rainfall deficiencies over a contiguous block of months including and preceding the evaluated month (i.e. prior and concurrent influencing months), or (iii) rainfall model deficiencies in a preceding month more so than in the evaluated month (i.e. prior influencing months). The integrated test cannot isolate which influencing months produce these deficiencies. Therefore, the unit test is designed to enable the identification of sources of poor performance in streamflow. The sources of poor performance are described in terms of which influencing months exhibit key deficiencies in simulated rainfall and therefore which SRM components should be improved.

### 2.4.1 Unit test procedure

The unit test investigates the impact of simulated rainfall in a given influencing month on the production of streamflow in an evaluated month of interest. This is achieved by splicing observed and simulated rainfall into a single time series which is used to produce simulated streamflow.

Following Fig. 4(a), consider the time series of observed, $R^{obs}$, and simulated, $R^{sim}$, daily rainfall for each year (and replicate) at a given site. Fig. 4(a) illustrates the embedding of simulated rainfall $R_k^{sim}$ in an influencing month, $k$, within observed rainfall $R_m^{obs}$ for all other months $m \in \{1, \dots, 12 | m \neq k\}$. The resulting spliced rainfall time series $R_{(k)}^{spl}$ is denoted with respect to the influencing month, $k$, and has the same length as the corresponding observed $R^{obs}$ and simulated $R^{sim}$ time series.

$$R_{(k)}^{spl} = \bigcup_{m=1}^{12} \begin{cases} R_m^{sim} \; ; m = k \\ R_m^{obs} \; ; m \neq k \end{cases} \tag{3}$$

For example, if June ($k = 6$) is selected as the influencing month, each year of the spliced time series, $R_{(6)}^{spl}$, would be composed as follows:

$$R_{(6)}^{spl} = \{R_1^{obs}, \dots, R_5^{obs}, R_6^{sim}, R_7^{obs}, \dots, R_{12}^{obs}\} \tag{4}$$

The ensemble of $k = 1, \dots, 12$ spliced rainfall time series $R_{(k)}^{spl}$ for all influencing months and additional inputs (e.g. potential evapotranspiration) indicated by '...' are transformed according to a hydrological model $g[\,]$ to produce an ensemble of simulated streamflow, $Q_{(k)}^{spl}$. This procedure is repeated for all simulated rainfall replicates.

$$Q_{(k)}^{spl} = g[R_{(k)}^{spl}, \dots] \tag{5}$$

By construction, the spliced rainfall is identical to the observed rainfall for all months other than the influencing month, so any errors in streamflow statistics can be attributed to the influencing month free from other factors.

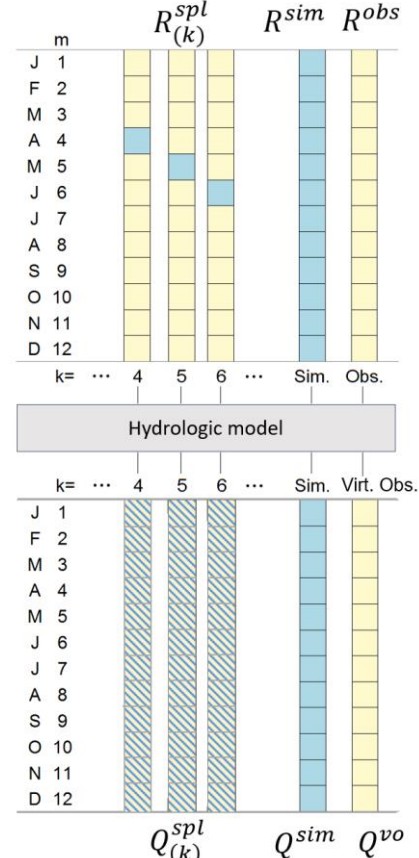

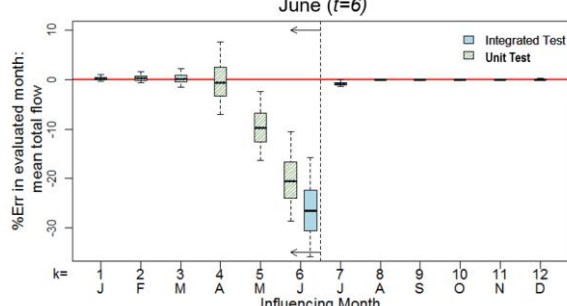

Fig. 4 Schematic of (a) the method of constructing a unit test by embedding simulated months in an observation time series, and (b) the error profile produced when using the integrated and unit tests for the evaluated time period of June ($t$=6) (box plot whiskers indicate the 90% limits of the simulated streamflow replicates). For the unit test the errors in the evaluated period ($t$) are calculated as the difference between $Q^{spl}_{(k)}$ and $Q^{vo}_{(t)}$. For the integrated test the errors are calculated as difference between $Q^{sim}$ and $Q^{vo}_{(t)}$.

The full set of spliced rainfall (e.g. spliced rainfall for each month designated as the influencing month $R_{(k)}^{spl}; k = 1, ... ,12$) is input to the hydrological model. This step is repeated for all available replicates of the spliced time series. The results of the unit test and the integrated test (Steps 1-2) are then investigated and compared selecting each month as the evaluated time period in turn as well as other key time periods (e.g. annual).

### 2.4.2   Compare unit tests and integrated tests

Side-by-side comparison of the results of the integrated test and unit tests are given in terms of the errors for selected monthly and annual statistics (see illustration in Fig. 4(b)). The comparison of errors from the unit test forms the basis of interpretation for hydrological insights and their relationship to the rainfall model.

A relative error metric, $\%Err$, is used to enable a comparison between the virtual-observed streamflow and the evaluated

streamflow replicates from the integrated or unit tests ($Q^{sim}$ or $Q_{(k)}^{spl}$) for time periods of interest (e.g. annual level or particular season or month). In this way the targeted hydrological evaluation centres on a specific subset of streamflows relating to the evaluated time period, $t$. In this paper, examples are provided for evaluated time periods at the monthly and annual scales.

Using the function $h[\ ]$ to denote a calculated statistic of interest (e.g. mean or standard deviation), the relative error in an evaluated time period $t$ (e.g. annual or particular month) is given by

$$\%Err_{(t)} = \frac{h\left[Q_{(t)}^{eval}\right] - h\left[Q_{(t)}^{vo}\right]}{h\left[Q_{(t)}^{vo}\right]} \times 100 \qquad (6)$$

where $Q_{(t)}^{vo}$ is the virtual-observed streamflow and $Q_{(t)}^{eval}$ is the simulated streamflow from the selected virtual hydrologic test (i.e. $Q^{sim}$ if integrated test or $Q_{(k)}^{spl}$ if unit test selected) in the evaluated time period $t$. This procedure is repeated for all replicates of the simulated streamflow such that a range of errors is reported for each test for the target time period.

Following the calculation of this error metric for all replicates of the integrated test and ensemble of unit tests ($k = 1, ... ,12$)

it is possible to investigate deficiencies in the simulated streamflow in terms of which influencing month(s) contribute more to the deficiencies in streamflow for the target time period based on that statistic of interest. Thus, for each site, statistic and evaluated time period there are 13 sets of errors to compare.

A typical error profile from integrated and unit tests is shown in Fig. 4(b) where mean monthly flow is selected as the statistic of interest for the evaluated time period of June ($t = 6$). In this figure the sets of errors from the integrated and unit tests are

summarised as boxplots with the boxplot whiskers indicating the 90% limits of the errors from the evaluated streamflow replicates. Fig. 4(b) shows the integrated test produced a median error of 27% (blue shaded boxplot) from all simulated rainfall replicates indicating a deficiency in the simulated streamflow for June. Examination of the unit tests (yellow and blue striped boxplots) for the target time period (June) shows that the median error is 20% when the influencing month is June ($k = 6$), the median error is 10% when the influencing month is May ($k = 5$) and when the influencing month is April ($k = 4$) the

median error is negligible. Therefore, the bias in mean June streamflow is primarily due to SRM deficiencies in June and May respectively.

### 2.4.3 Identify types of key deficiencies

Following a side-by-side comparison of integrated test and unit test results in terms of the relative errors, the sources of poor performance should be classified in terms of in which influencing months streamflow deficiencies originate (e.g. poor streamflow arises from rainfall deficiencies mostly in the same month, a prior month or a contiguous block of months). Differentiating between cases allows for SRM improvements to be targeted in terms of their ultimate impact on streamflow statistics. To complement this analysis a comparison of the virtual-observed flow duration curve for the evaluated time period with the flow duration curves resulting from unit tests for key influencing months is also recommended. Examples for each of case are presented in Section 1.

## 3 Case Study

The Onkaparinga catchment in South Australia is used as a case study (Fig. 5). The 323 km$^2$ catchment lies 25 km south of the Adelaide metropolitan area and contains the largest reservoir in the Adelaide Hills supplying the region (Mount Bold Reservoir). The catchment has a strong seasonal cycle (shown in Fig. 6) where the driest months (December, January and February) exhibit low rainfall and low streamflow, the wettest months (July, August and September) have high rainfall and high streamflow and the 'wetting-up' period (April, May and June) has high rainfall and lower streamflow. There is a strong rainfall gradient (Table 3), with average annual rainfall ranging from approximately 500 mm on the coast (Site No. 19) to over 1000 mm in the region of highest elevations (Site No. 20). A breakdown of the rainfall characteristics (annual total, number of wet days, daily average amounts, wet-spell and dry spell durations) at each site on a monthly basis is provided in Supplementary Material A.

The simulated daily rainfall was determined from the latent variable autoregressive daily rainfall model of Bennett et al. (2018) using at-site calibrated parameters. This rainfall model uses a latent variable concept, which relies on sampling from a normally distributed 'hidden' variable. The latent variable can then be transformed to a rainfall amount by truncating values below zero and by rescaling values above zero to match the observed rainfall's distribution. Here, the rainfall is rescaled using a power transformation.

To calibrate the model the rainfall data at a given site is partitioned on a monthly basis and separate parameters are fit for each month. The mean and standard deviation of rainfall amounts, as well as the proportion of dry days is calculated. These statistics are matched to the corresponding properties of the truncated power transformed normal distribution. The at-site lag-1 temporal correlation is then calculated based on the observed wet day periods for a given month. This statistic is transformed to the equivalent correlation of the underlying latent variable by accounting for the effects of truncation to determine the autocorrelation parameter. Full details of the calibration procedure are provided in Bennett et al. (2018). In this study the daily

rainfall model was calibrated and simulated at 22 locations throughout the catchment that have long, high-quality records (Table 3). 10,000 replicates of simulated rainfall covering a 73 year period (1914-1986) were used.

The hydrological model GR4J (Perrin et al., 2003) was used to simulate virtual-observed streamflow at a daily time step. GR4J is a daily lumped hydrological model that simulates daily streamflow in a parsimonious manner using four parameters. The GR4J model was calibrated according to the procedure set out in Westra et al. (2014b) for the stationary version of the GR4J hydrological model. The details are provided in (Westra et al., 2014a) and a short summary is provided here. The multi-site rainfall gauges were Thiessen weighted to calculate the catchment average rainfall. The hydrological model was calibrated to the daily streamflow data at Houlgrave Weir (see Fig. 5) using model calibration period of 15 years (1985-1999). The model parameters were estimated using maximum likelihood estimation procedure with a weighted least squares likelihood function. The set of hydrological model parameters that maximised the likelihood function were found using a multi-start quasi-Newton optimisation procedure with 100 random starts. Overall, the GR4J model was able to simulate streamflow with a good fit to the observed daily streamflow, with a Nash-Sutcliffe efficiency of 0.8. A similar type of hydrological model and calibration approach has been used for other virtual evaluation studies (Li et al. 2014; 2016). The same set of hydrological model parameters are used for both the unit and integrated tests so that the same transformation of rainfall to flow is used.

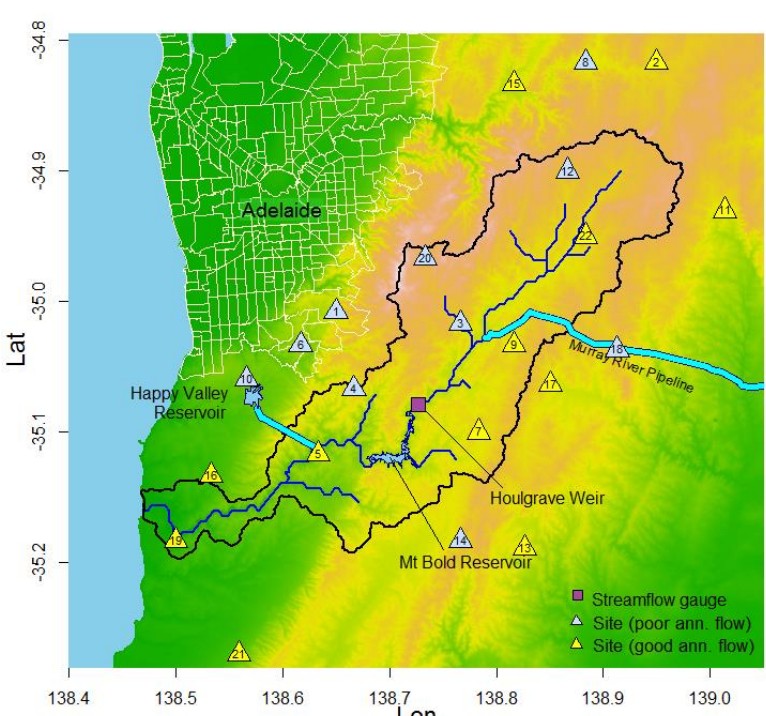

**Fig. 5 Onkaparinga catchment, South Australia. Sites indicated by blue triangles are explored in greater detail in this paper due to the relatively poorer ability of simulated rainfall to reproduce annual streamflow totals at these sites.**

**Table 3 Site names, locations and seasonal rainfall characteristic summary. Sites ordered from lowest to highest elevation.**

| Site No | Site Name | Elev (m) | Ann. Av. Rain (mm) | January | | | | | July | | | | |
|---|---|---|---|---|---|---|---|---|---|---|---|---|---|
| | | | | Total (mm) | No. Wet (days) | Daily Av. (mm) | Wet-spell (days) | Dry-spell (days) | Total (mm) | No. Wet (days) | Daily Av. (mm) | Wet-spell (days) | Dry-spell (days) |
| 19 | Old Noarlunga | 7 | 520 | 20 | 4.1 | 0.6 | 1.6 | 9.6 | 72 | 17 | 2.3 | 3.1 | 2.8 |
| 16 | Morphett Vale | 90 | 560 | 20 | 4.1 | 0.6 | 1.5 | 8.9 | 76 | 17 | 2.4 | 3.3 | 2.8 |
| 10 | Happy Valley | 148 | 640 | 22 | 4.8 | 0.7 | 1.7 | 8.2 | 88 | 18 | 2.8 | 3.6 | 2.6 |
| 21 | Willunga | 158 | 640 | 23 | 4 | 0.7 | 1.6 | 10 | 95 | 17 | 3 | 3.2 | 2.7 |
| 5 | Clarendon | 223 | 820 | 25 | 4.7 | 0.8 | 1.7 | 8.9 | 114 | 17 | 3.7 | 3.4 | 2.8 |
| 6 | Coromandel | 234 | 710 | 24 | 4.8 | 0.8 | 1.8 | 9.2 | 102 | 18 | 3.3 | 3.6 | 2.8 |
| 13 | Macclesfield | 302 | 730 | 28 | 5.3 | 0.9 | 1.8 | 7.9 | 99 | 17 | 3.2 | 3 | 2.7 |
| 15 | Cudlee Creek | 311 | 830 | 29 | 5 | 0.9 | 1.8 | 8.4 | 123 | 18 | 3.9 | 3.8 | 2.7 |
| 11 | Harrogate | 335 | 550 | 23 | 3.5 | 0.7 | 1.6 | 12 | 75 | 12 | 2.4 | 2.2 | 3.8 |
| 4 | Cherry gardens | 345 | 920 | 30 | 5.4 | 1 | 1.8 | 7.7 | 134 | 18 | 4.3 | 3.8 | 2.6 |
| 8 | Gumeracha | 346 | 790 | 27 | 5.3 | 0.9 | 1.8 | 7.8 | 108 | 18 | 3.5 | 3.5 | 2.6 |
| 9 | Hahndorf | 347 | 850 | 29 | 5.4 | 0.9 | 1.9 | 8.1 | 123 | 18 | 4 | 3.4 | 2.7 |
| 17 | Mount Barker | 349 | 770 | 28 | 5.9 | 0.9 | 1.9 | 7.2 | 104 | 18 | 3.3 | 3.3 | 2.6 |
| 7 | Echunga | 375 | 805 | 28 | 5 | 0.9 | 1.8 | 8.7 | 110 | 17 | 3.5 | 3.3 | 2.6 |
| 3 | Bridgewater | 376 | 1050 | 32 | 5.2 | 1 | 1.9 | 8.9 | 154 | 18 | 4.9 | 3.6 | 2.7 |
| 14 | Meadows | 384 | 870 | 30 | 4.8 | 1 | 1.7 | 8.5 | 122 | 17 | 3.9 | 3.2 | 2.7 |
| 2 | Birdwood | 385 | 720 | 25 | 4.4 | 0.8 | 1.8 | 9.6 | 104 | 17 | 3.4 | 3.4 | 2.8 |
| 1 | Belair | 386 | 790 | 28 | 4.6 | 0.9 | 1.8 | 9.8 | 111 | 16 | 3.6 | 3.2 | 3 |
| 22 | Woodside | 387 | 800 | 27 | 4.3 | 0.9 | 1.6 | 8.3 | 121 | 16 | 3.9 | 2.9 | 2.7 |
| 18 | Nairne | 403 | 680 | 28 | 4.7 | 0.9 | 1.6 | 8 | 93 | 16 | 3 | 2.8 | 2.8 |
| 12 | Lobethal | 470 | 880 | 28 | 4.9 | 0.9 | 1.8 | 8.4 | 133 | 18 | 4.3 | 3.5 | 2.6 |
| 20 | Uraidla | 499 | 1090 | 35 | 4.7 | 1.1 | 1.8 | 9 | 161 | 17 | 5.2 | 3.4 | 2.7 |

Note: Wet days are defined as days where the rainfall exceeded a 0.1 mm threshold with wet-spells defined as the number of days in a row above the threshold (and vice versa for dry-spells).

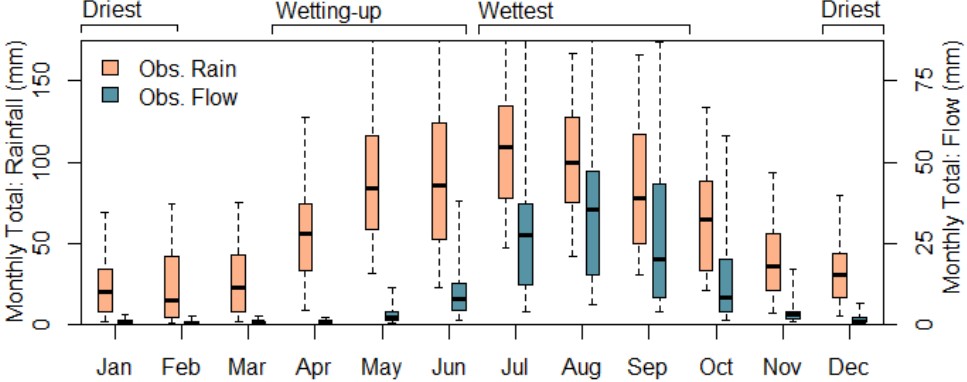

**Fig. 6 Seasonal variation of catchment average rainfall and flow at Houlgrave Weir. Boxplots show the variation across years. Wettest indicates (high rainfall, high flow), direst indicates (low rainfall, low flow) and wetting-up indicates (high rainfall, low flow).**

# 4 Results

## 4.1 Step 1 – Identify poor performing sites

To undertake Step1, annual total flow volumes were designated as the primary streamflow characteristic to narrow the number of sites investigated. Following the selection of the primary streamflow characteristic and selection of the hydrological model, GR4J, integrated tests were undertaken to evaluate the simulated rainfall at the 22 sites. The annual total flow distribution was used to give a broad indication of performance. This step categorised 10 of the 22 sites as 'poor' and 12 as 'good', which is in strong contrast to earlier evaluation efforts using observed-rainfall evaluation (Bennett et al., 2018) that categorised the majority of sites and statistics as 'good' (see Section 2.2.3 for category definitions).

The 10 sites categorised as 'poor' are the focus of subsequent hydrologic evaluation framework steps. These 'poor' performing sites are indicated by the blue triangles in Fig. 5.

## 4.2 Step 2 – Identify poor performing time periods

The poor performing sites identified in Step 1 were then compared in terms of both an observed-rainfall evaluation and virtual hydrological evaluation via an integrated test. Fig. 7 graphically summarises this comparison, with each row presenting monthly or annual performance of the following statistics:

- simulated daily rainfall statistics (mean ($m$) daily amounts, standard deviation ($sd$) of daily amounts, mean number of wet days ($nwet$) and the standard deviation of the number of wet days);
- aggregate rainfall statistics (mean and standard deviation of total rainfall); and
- aggregate streamflow statistics (mean and standard deviation of total flow).

The first to fourth columns of Fig. 7 summarise the observed-rainfall evaluation and the fifth and sixth of Fig. 7 summarise the virtual hydrological evaluation. The first column of Fig. 7 indicates that of the poor performing sites the SRM exhibited 'good' performance in simulating daily rainfall means and standard deviations as well as the mean number of wet days for all sites and months and at an annual level according to the observed-rainfall evaluation. Each of the three statistics presented in the first column are assessed separately but are presented together to avoid repetition. Whereas the second column indicates that there is mixed performance across sites and months in simulating the variability in the number of wet days ($sd(nwet)$). Likewise, the third and fourth columns indicate overall 'good' performance in simulating mean monthly totals and mixed performance in simulating the monthly or annual total standard deviations ($sd(total)$). Whereas the virtual hydrological evaluation (fifth and sixth) columns show mostly 'good' performance in all months other than those in the 'wettest' or 'wetting-up' periods.

A clear trend, from Fig. 7 is the contrast in performance between the observed-rainfall evaluation and the virtual hydrological evaluation. One constrast is that, in the driest months (Dec, Jan, Feb) 'poor' performance in simulating rainfall (based on observed-rainfall evaluation) did not necessarily translate to 'poor' performance in simulating streamflow (based on virtual

hydrological evaluation). For example, examining the first row of Fig. 7, the observed-rainfall evaluation shows that in January the SRM's ability to simulate variability in the number of wet days, *sd(nwet),* was 'poor' for all sites. However, in contrast the virtual hydrological evaluation shows that most sites had 'good' performance in simulating the January distribution of monthly total flow (i.e. *m(total)* and *sd(total)).*

A second contrast is that 'good' performance in the observed-rainfall evaluation does not necessarily translate to 'good' performance for the virtual hydrological evaluation, particularly for months in the 'wettest' and 'wetting-up' periods. For example, in Fig. 7 the rows summarising June and August show large percentages of 'poor' sites in the virtual hydrological evaluation of monthly total flow. This deficiency would have been difficult to infer using the observed-rainfall evaluation due to the 100% 'good' performance of *m(total)* rainfall and 'good/fair' performance of *sd(total)* rainfall in these months.

Likewise, by examining the bottom row of Fig. 7 that summarises annual performance, it can be seen that the observed-rainfall evaluation shows unbiased mean annual total, *m(total)*, rainfall (100% 'good') and yet the mean annual total flows showed only 10% of sites as 'good'. Discussion of the unit tests in the following section will investigate reasons why apparently 'good' rainfall can yield 'poor' flow.

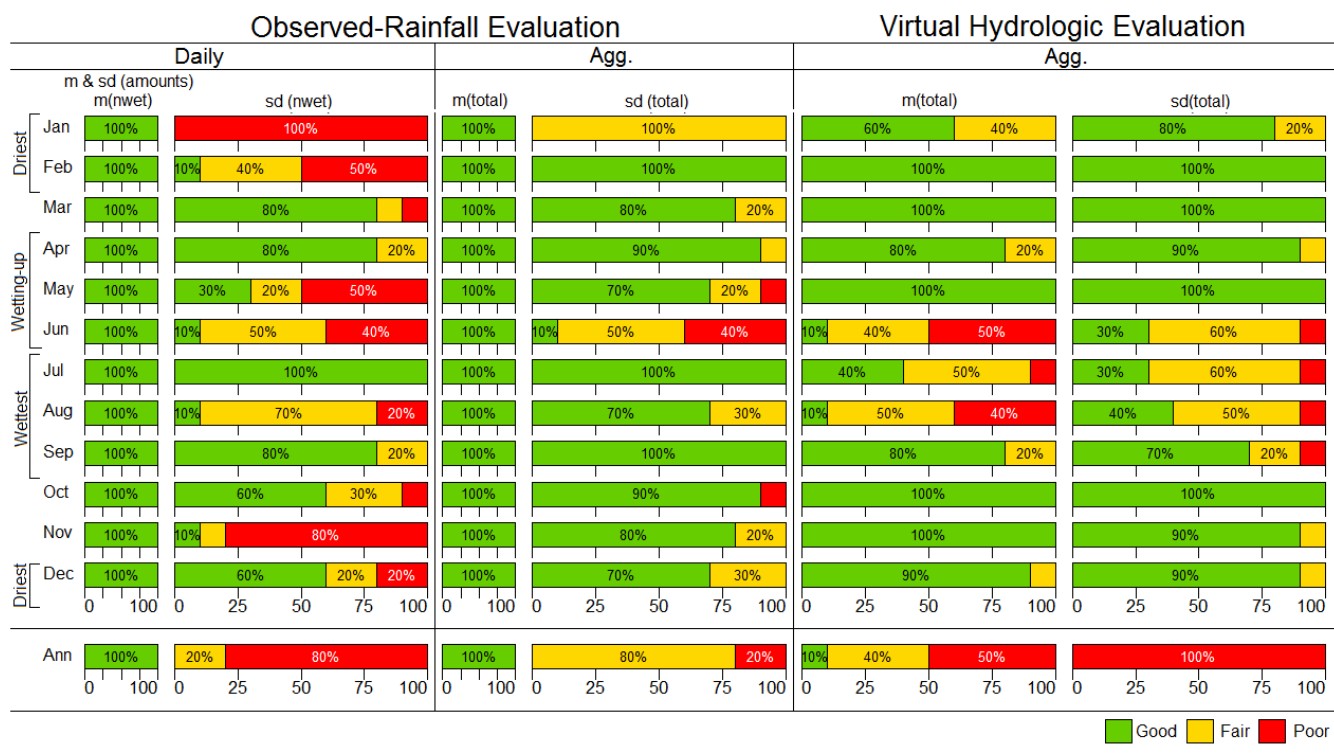

**Fig. 7 Integrated test, comparing observed-rainfall evaluation (left) with the virtual hydrologic evaluation (right). Comparison of daily and aggregate ('Agg.') rainfall statistics against aggregate flow statistics for individual months and years.**

### 4.3 Step 3 – Identify sources of poor performance

To undertake Step 3, unit tests were run to evaluate the source of deficiencies in poor performing time periods. The results of these test were compared against integrated tests in terms of their relative errors. From this comparison the source and type of key deficiencies in the simulated rainfall that lead to poor performance in simulated streamflow were identified. A comparison of the virtual-observed flow duration curve for the poor performing time periods and the flow duration curves resulting from unit tests for key influencing months was also undertaken to illustrate the impact of these key deficiencies on the daily flow duration curve.

Here, four examples of the different types of key deficiencies are illustrated using two locations, Site 12 and Site 10 (see Fig. 8 to Fig. 13). For completeness these results are presented together with the results of the observed-rainfall evaluation (panels (a) and (b) of Fig. 8 and Fig. 11).

#### 4.3.1 Streamflow errors mostly originate from rainfall model deficiencies in the evaluated month

A common case for streamflow errors is that they originate from rainfall in the same month. This case can be illustrated using Site 12 in Fig. 8 where left-side panels show results for the mean and right-side panels show the standard deviation and where panels (a) and (b) summarise the observed-rainfall evaluation, (c) and (d) summarise the integrated test. From panels (a) and (b), the simulated monthly rainfall is generally unbiased, but from (c) and (d) the mean and standard deviation of the simulated streamflow is lower than the virtual-observed flow from June to September. Here, September is selected as an illustrative case for an application of the unit test in Fig. 9 since it shows biased flow.

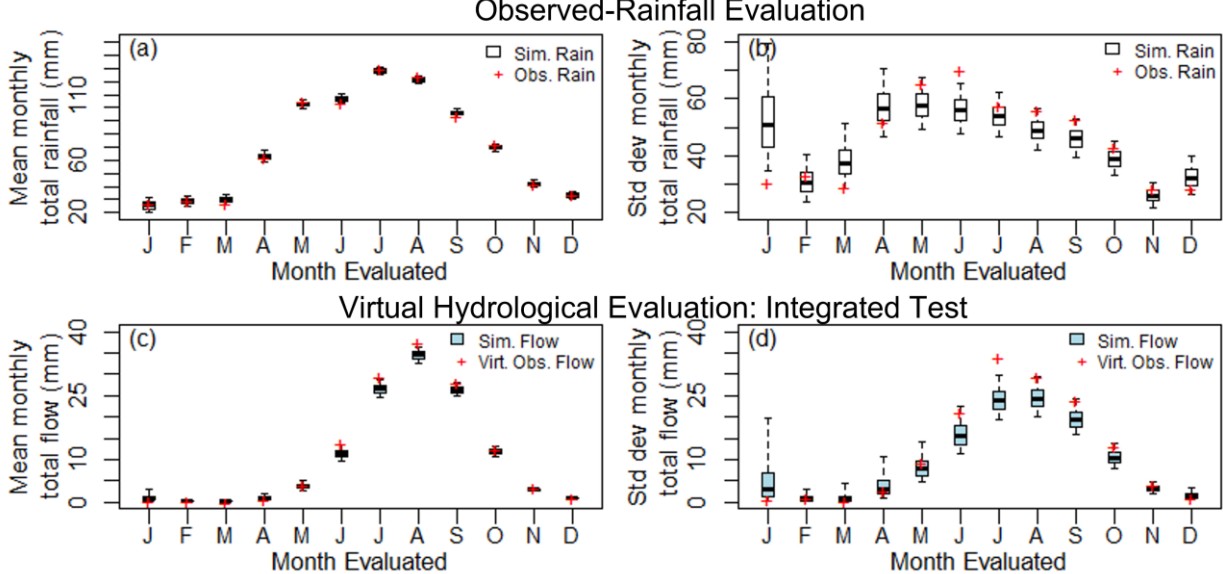

**Fig. 8 Lobethal, Site 12 (a) observed-rainfall evaluation mean monthly total rainfall, (b) observed-rainfall evaluation standard deviation of monthly total rainfall, (c) virtual hydrological evaluation (integrated test) mean monthly total streamflow, (d) virtual hydrological evaluation (integrated test) standard deviation of monthly total streamflow. Boxplot whiskers indicate the 90% limits of the simulated streamflow or rainfall replicates.**

Taking September as the evaluated month ($t=9$), Fig. 9 (a) and (b) compare the unit tests for all 12 influencing months (yellow and blue striped boxplots) with the integrated test (blue shaded boxplot) in terms of the error in the simulated flow. When the influencing month is September (i.e. the September rainfall is 'spliced' into the observed record, $k=9$) the resultant error is greatest and closest to the error for the integrated test for both the mean monthly total flow (Fig. 9 (a)) and standard deviation

of monthly total flow (Fig. 9 (c)). For the example of the standard deviation, when the influencing month is July (i.e. July rainfall is spliced into the observed record) the median error is less than 2%, whereas when September is taken as the influencing month the median error is approximately 16% (Fig. 9 (b)). Therefore, to improve September flows, September rainfall should be improved in preference to all other months.

This need to improve September in preference to preceding months is also illustrated via Fig. 9 (c) where the September daily

flow duration curves are shown for the cases where August (orange shading) and September (blue shading) are the influencing months compared against the virtual-observed September flow duration curve (purple dots). Where August is selected as the influencing month, the virtual-observed flow duration curve largely sits inside the 90% limits of the flow duration curves resulting from the unit testing procedure. Whereas, the virtual-observed flow duration curve is located outside the 90% limits of the unit test flow duration curve when September is taken as the influencing month. Thereby providing further evidence

that, to improve September flows, September rainfall should be improved in preference to other months.

Analysing other sites and months suggests that over 50% of the evaluations correspond to this case, and they typically occur in spring and summer months when the catchment is drying out.

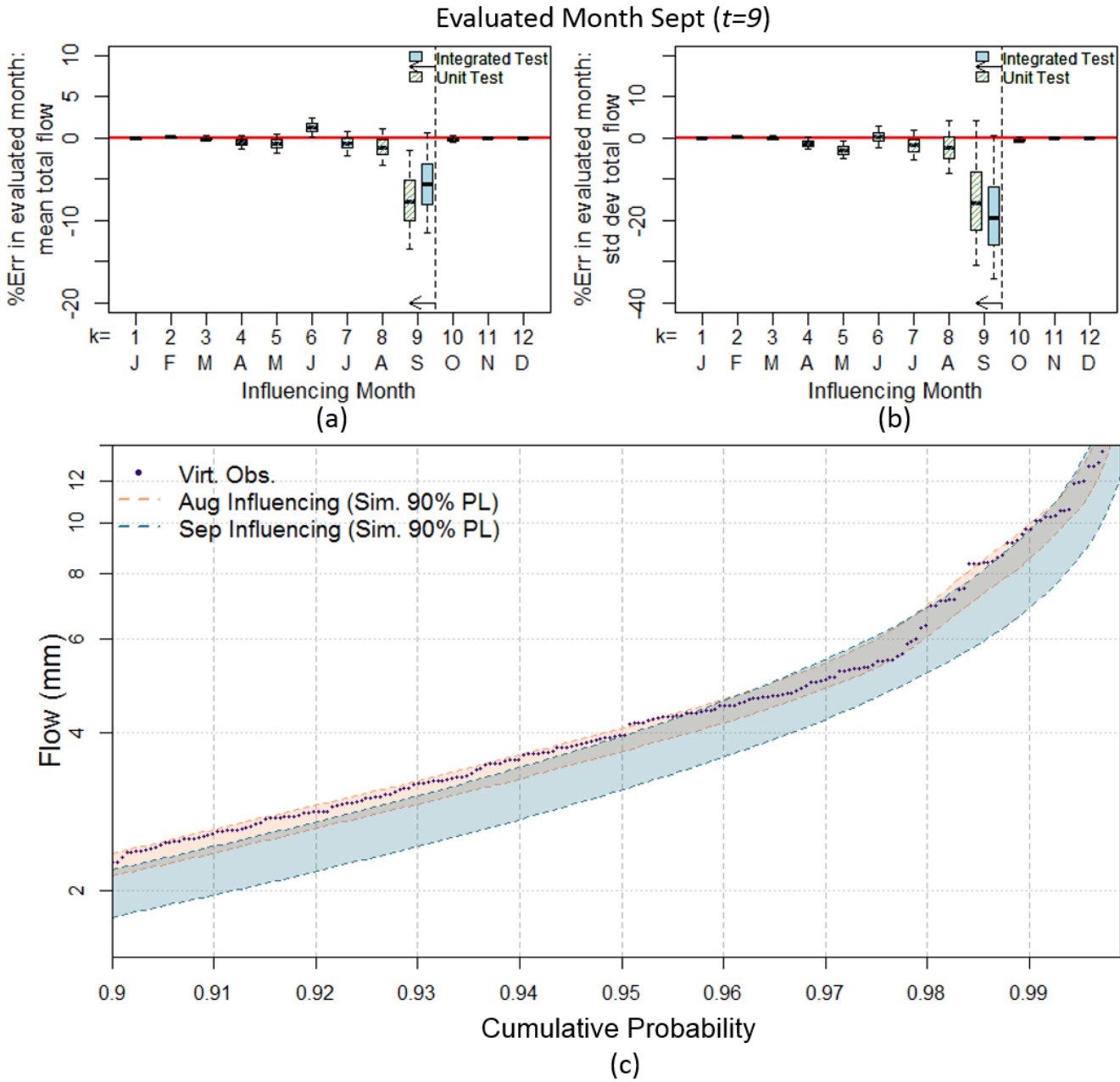

**Fig. 9** Lobethal, Site 12 (a) unit test error in mean monthly flow (September), (b) unit test error in standard deviation of monthly flow (September), (c) Unit test September flow duration curve when August and September are selected as influencing months (top 10% of flow days shown). Boxplot whiskers indicate the 90% limits of the simulated streamflow replicates.

### 4.3.2    Streamflow errors originate from rainfall model deficiencies over a contiguous block of months

An illustration of the case where streamflow errors originate from rainfall model deficiencies over a contiguous block of months is provided by Site 12, where July is selected as the evaluated month. Comparison of the July performance in the integrated and unit tests (Fig. 10 (a) and (b)) demonstrates that the errors in July streamflow do not originate in the July rainfall alone (unlike the case for September – see Section 4.3.1). Although the largest percentage error in flow is attributable to July (a median error of 8% in mean monthly total flow and 25% in the standard deviation of monthly total flow when the influencing month is July) a significant proportion of the error for July streamflow originates in prior months. June and May rainfall have a significant influence on the July flow with percentage errors of up to 15% in July flow when June or May are the influencing month. Therefore, to improve July flows, it is not just the July rainfall that should be improved but also the preceding two months.

This need to improve July and preceding months is also illustrated via Fig. 10 (c) where the July daily flow duration curves are shown for the cases where June (orange shading) and July (blue shading) are the influencing months compared against the virtual-observed July flow duration curve (purple dots). For both cases the virtual-observed flow duration curve is located outside the 90% limits of the flow duration curves resulting from the unit testing procedure.

Typically, 'wetting-up' and 'wettest' months fall in this case where streamflow errors originate from rainfall model deficiencies over a contiguous block of months, approximately 40% of the site/month combinations.

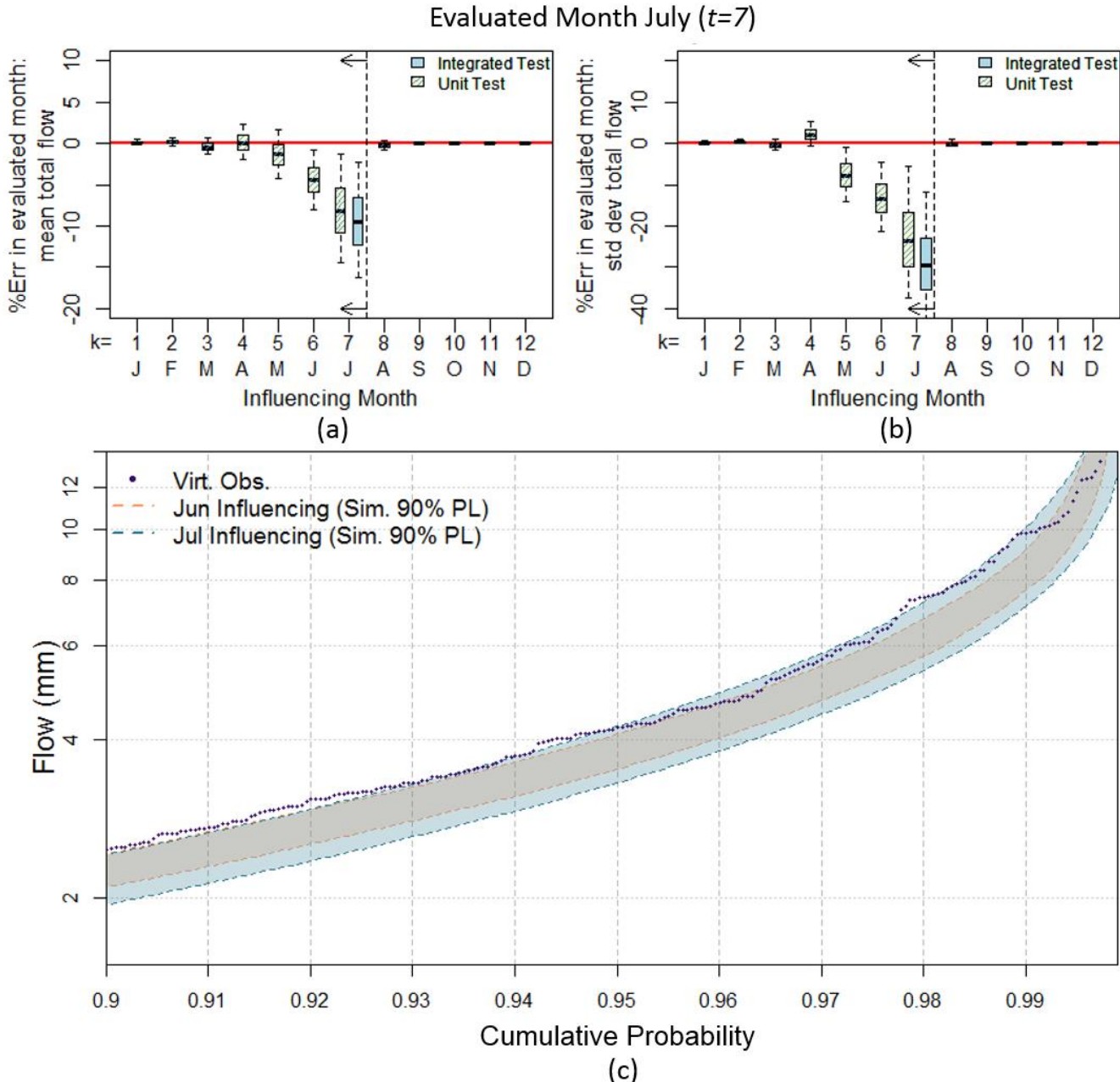

**Fig. 10** Lobethal, Site 12 (a) unit test error in mean monthly total flow (July), and (b) unit test error in standard deviation of monthly total flow (July), (c) July flow duration curve when June and July are selected as influencing months in unit test (top 10% of flow days shown). Boxplot whiskers indicate the 90% limits of the simulated streamflow replicates.

### 4.3.3 Streamflow errors originate from rainfall model deficiencies in a preceding month more so than evaluated month

An example of the case where the largest contribution to streamflow errors arises from rainfall deficiencies in a preceding month is provided by Site 10, where July is selected as the evaluated month. July is selected as an illustrative case for application of the unit test since it shows biased flow (see Fig. 11 (c) and (d)), but did not show any bias in the simulated rainfall (see Fig. 11 (a) and (b)).

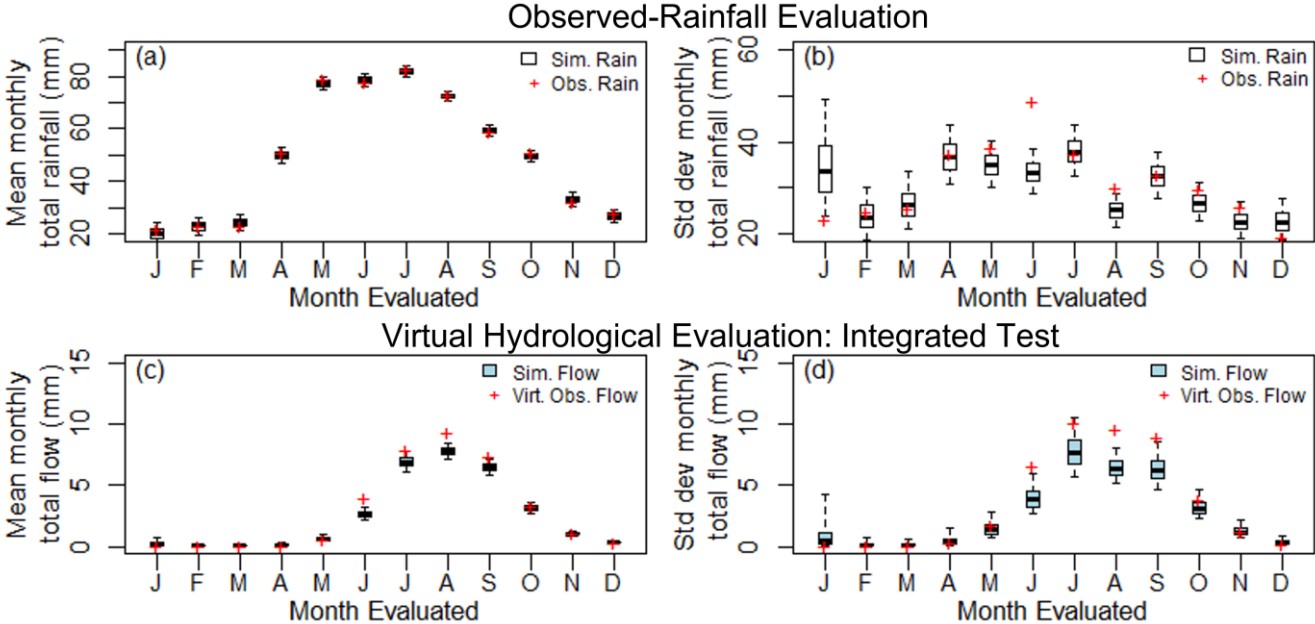

**Fig. 11 Happy Valley (Site 10) (a) observed-rainfall evaluation mean monthly total rainfall, (b) observed-rainfall evaluation standard deviation of monthly total rainfall, (c) virtual hydrological evaluation (integrated test) mean monthly total streamflow (d) virtual hydrological evaluation (integrated test) standard deviation of monthly total streamflow. Boxplot whiskers indicate the 90% limits of the simulated streamflow or rainfall replicates.**

The largest contributor to error in July flow is not July rainfall but June rainfall (Fig. 12 (a) and (b)). That is, the largest errors occur when there is observed rainfall for July spliced with simulated rainfall for June. In contrast, simulated July rainfall spliced with observed rainfall in other months, yields a smaller median error. This deficiency in June rainfall can also be seen in an examination of the July flow duration curves (Fig. 12 (c)) where the virtual-observed streamflow sits within the 90% limits of the simulated flow duration curve where July is designated as the influencing month, whereas when June is designated as the influencing month the virtual-observed streamflow sits outside the 90% limits for a number of the higher flow days.

While improving the July rainfall will improve the simulation of July flow, a more significant improvement will be obtained by focusing on improving the June rainfall. The category where streamflow errors originate from rainfall model deficiencies in a preceding month represents about 10% of the evaluated site/month combinations (i.e. those identified in Step 2).

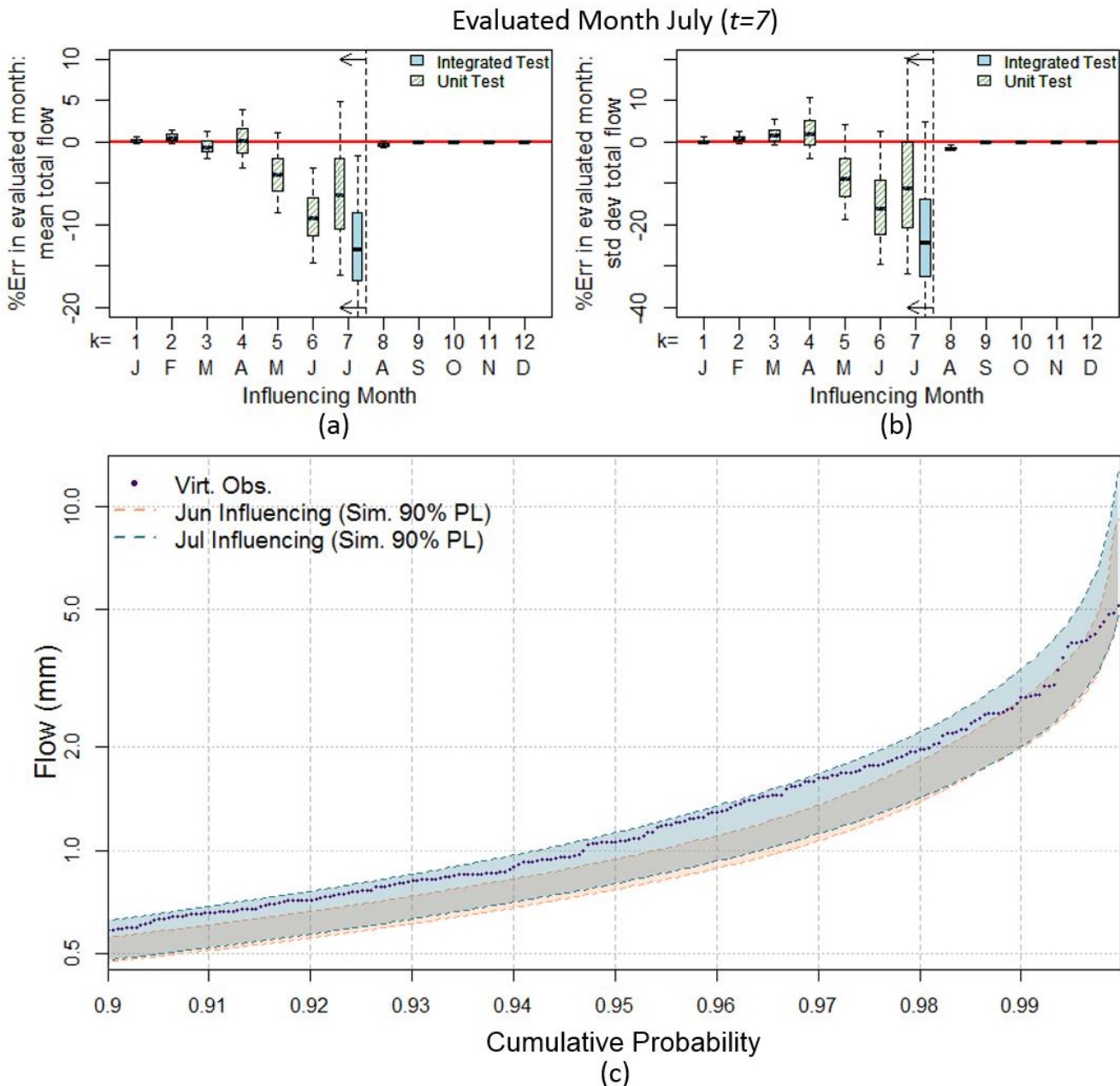

**Fig. 12 Happy Valley (Site 10) (a) unit test error in mean monthly flow (July), (b) unit test error in standard deviation of monthly flow, and (c) July flow duration curve when June and July are selected as influencing months in unit test (top 10% of flow days shown). Boxplot whiskers indicate the 90% limits of the simulated streamflow replicates.**

### 4.3.4    Influence of monthly rainfall on annual flow volumes

While annual simulated rainfall was unbiased, annual simulated streamflow was biased. An illustration of how errors in annual total streamflow arise from rainfall is shown for Site 10. Fig. 13 (a) and (b) show that when the months of May to August are assessed as the influencing month they produce the largest errors in distribution of annual total flow for Site 10. Splices of other months do not significantly degrade the simulation of total annual flow. This deficiency can also be seen via an examination of the flow duration curve (Fig. 13 (c)) in which the virtual-observed flow duration curve is located outside portions of the simulated flow duration curves where May or June are designated as the influencing month. Improvements to the simulation of annual total flow will therefore come from improving the rainfall model in the 'wetting-up' and wettest months of the seasonal catchment cycle (May to August). This insight from the use of unit testing would be difficult to obtain using other evaluation strategies (further discussed in Section 5.2).

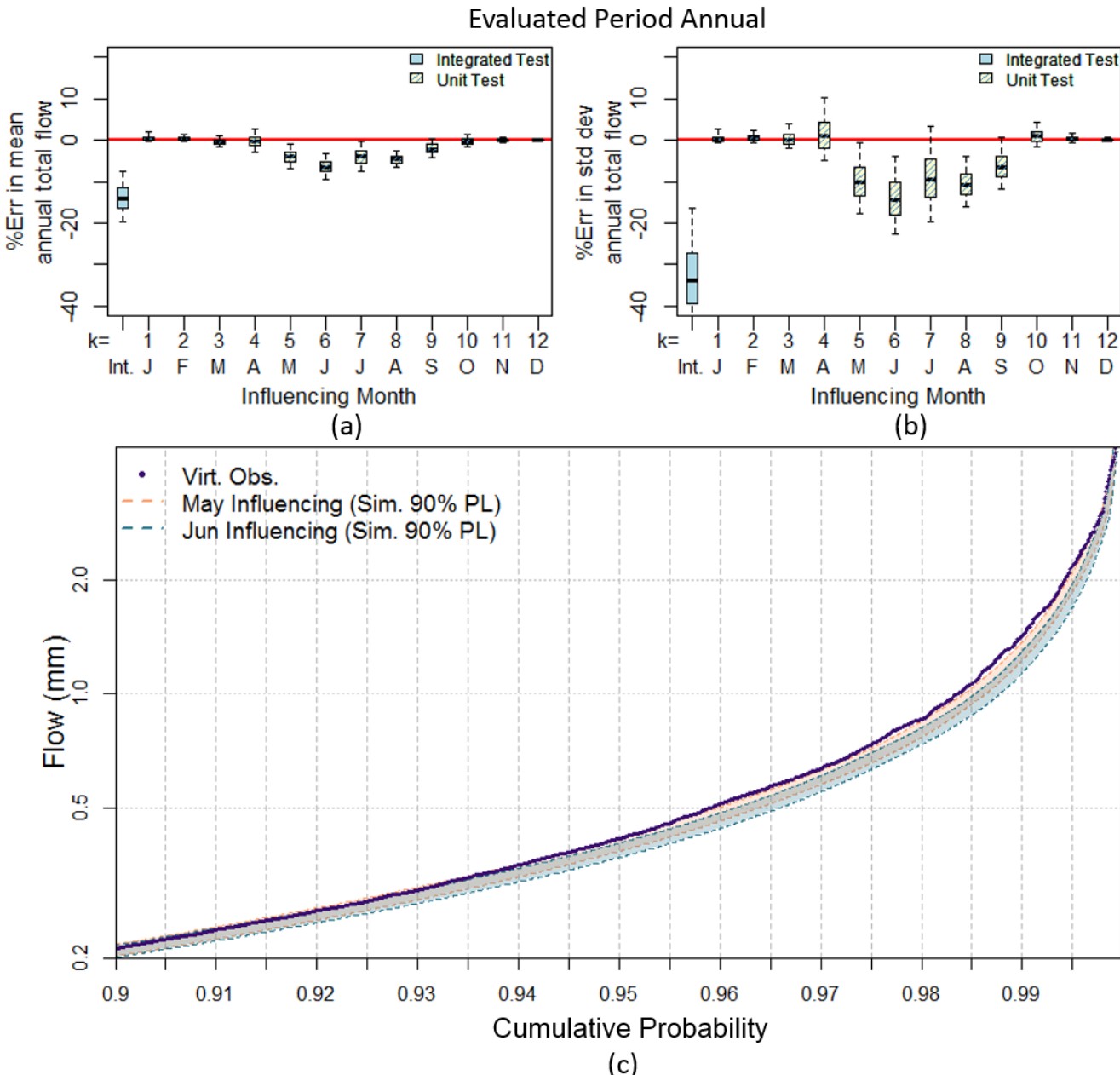

**Fig. 13 Happy Valley (Site 10) (a) unit test error in mean annual total flow, and (b) unit test error in standard deviation of annual total flow, and (c) annual flow duration curve when May and June are selected as influencing months in unit test (top 10% of flow days shown). Boxplot whiskers indicate the 90% limits of the simulated streamflow replicates.**

## 5    Discussion

### 5.1    The importance of streamflow-based evaluation

Streamflow arises from the integration of rainfall processes (e.g. rainfall amounts, occurrences and wet-dry patterns) over a catchment. Features of the catchment, such as catchment storage, thresholds and nonlinearities in the rainfall-streamflow response function, can either act to amplify or dampen the rainfall errors at different times of the year. These behaviours were clearly identified and demonstrated in Step 2 of the virtual hydrological evaluation framework which compares observed-rainfall evaluation and virtual hydrological evaluation (see Section 4.2).

In terms of amplification, the elasticity of the rainfall-streamflow response (Chiew, 2006) suggests that catchments can have strong sensitivities to discrepancies in rainfall. Given that the rainfall elasticity of streamflow to rainfall is a factor of 2 to 3.5 (Chiew, 2006), using the principles of error propagation (Ang and Tang, 2007), assuming linearity it follows that a 10% error in mean/standard deviation of rainfall could potentially be amplified to 20-35% error in the mean/standard of streamflow. This estimate represents a lower-bound of the potential amplification, since the non-linear nature of the rainfall-runoff transformation will likely produce a larger potential amplification of errors. This indicates that streamflow-based evaluation of rainfall models provides a stronger test than observed-rainfall evaluation in terms of the sensitivity of the statistics. For example, Fig. 7 shows that July rainfall statistics were classified as 'good', yet despite this, the streamflow response was 'poor' (see Section 4.2 for further discussion). It could be argued that the rainfall results presented in Fig. 7 were classified as 'good' because the observed-rainfall evaluation was limited, but the evaluation was methodical and used a comprehensive range of daily and monthly statistics (Bennett et al., 2018). While many rainfall statistics were preserved (means, standard deviation, extremes, marginal distributions of daily rainfall) the rainfall-streamflow response of the catchment exposes that there are deficiencies in the rainfall model not clearly identified by the observed-rainfall evaluation (Bennett et al., 2018).

In terms of dampened influence, catchment storages and high evapotranspiration can also act to suppress errors in the rainfall simulations. For example, Fig. 7 showed that the variability in the number of wet days, $sd(nwet)$, was 'poor' for all sites in January, yet this did not result in 'poor' streamflow. The high potential evapotranspiration in January indicates that the majority of rainfall in January is converted into actual evapotranspiration yielding little streamflow. Hence, any errors in rainfall do not noticeably impact on January streamflow.

It is clear that streamflow-based evaluation is beneficial in addition to conventional observed-rainfall evaluation.

### 5.2    The benefits of the virtual evaluation framework

A benefit of virtual hydrological evaluation is that it is a relative measure of performance, where the hydrological model is a common factor in the construction of virtual-observed and simulated streamflow. This enables discrepancies in the streamflow to be identified in terms of SRM features. In contrast, observed-streamflow evaluation is typically hampered by difficulties in

separating the impact of data errors, hydrological model predictive performance from the errors in the SRM. A further benefit is the ability to undertake streamflow-based evaluation at any site where rainfall is observed and simulated. This enables insights into the SRM performance for simulating streamflow on a site-by-site basis.

The use of a virtual hydrological framework for evaluation provides the unique opportunity to develop innovative tests that can target specific aspects of the SRM. This paper introduces an innovative unit test that was used as a method for isolating the influence of rainfall in a month (i.e. the influencing month) on streamflow in an evaluated month while excluding the possibility of deficiencies from other rainfall months. The test enables a procedure for targeting months that are influential in terms of streamflow production rather than interpret model performance based on blunt evaluation of rainfall or streamflow.

This unit test provides added value over and above the integrated test because it identifies which are the influencing months that have deficiencies in the modelled rainfall that produce poor streamflow predictions. For example, Section 4.3.2 illustrated that while the integrated test identified that there was poor streamflow in July for Site 12, the unit test was able to identify that the simulated rainfall in the prior influencing months of both May and June (Fig. 10) made significant contributions (10-15% errors) to July's poor streamflow. A second example is shown in the influence of monthly rainfall on the errors in annual flow volumes in Section 4.3.4. If the modeller had focussed on improving the rainfall model by focusing on months with the highest contribution to annual total flow, July to September would have been identified as important, whereas the unit test identifies a different focus (May-August). The unit tests in Section 4.3.4 show that May and June combined make up 13% of the total annual flow volume (Fig. 11 (c)). However, they contribute to 11% of the error in the mean annual total flow (Fig. 13 (a)) and 24% error in the standard deviation (Fig. 13 (b)). By contrast, September is a high flow month contributing 21% of the annual total flow, but only 2% error in the mean and 6% error in the standard deviation. Without the unit test, it would have been less clear that the 'wetting-up' months such as May and June were a more important focus for SRM improvement than a high-flow month such as September.

### 5.3    Limitations and future research

The virtual hydrological framework for SRM evaluation provides opportunity for further improvements in the future, including:

(i) Using multiple, well-tested hydrological models - a potential limitation of the virtual hydrologic evaluation framework is that it is reliant on the use of a single hydrological model. Hydrological structural errors may potentially skew interpretation of the SRM evaluation if the hydrological model poorly represents the catchment processes. To reduce these impacts the steps taken in this study included (a) using a well-tested hydrological model that has demonstrated good performance on a wide range of catchments (e.g. the GR4J model has been widely tested–see Perrin et al. (2003) and Coron et al. (2012)); (b) calibrating and evaluating the hydrological model on a catchment close to the observed rainfall sites to ensure it provided sufficiently good performance  (e.g. GR4J was calibrated to the Onkaparinga catchment–see Westra et al. (2014b) and Westra et al. (2014a)). Future research will use multiple, well-tested hydrological models with sufficiently good performance to reduce

the reliance on a single hydrological model and ensure the identification of SRM deficiencies is not dependent on a single hydrological model.

(ii) Comparison of SRMs – this framework can be extended to provide more direct guidance on which rainfall features (in terms of components of the SRM) should be modified to improve streamflow performance. This can be done by comparing multiple rainfall model variants (parametrically, or via bootstrap techniques) which are designed to have contrasting features of a key characteristic (e.g. intermittency, rainfall correlation). Such an approach was undertaken by Evin et al. (2018) using an observed-rainfall evaluation approach. If the SRMs have monthly/seasonal autocorrelation (these were not significant for the rainfall in the Onkaparinga catchment) the unit testing approach would need to be extended by conditionally sampling the simulated rainfall in a manner that preserves monthly correlations.

(iii) Evaluation of temporal non-stationarity – this framework can be extended to evaluate the impact of non-stationarity on SRM model performance by applying it on a selected non-stationary period. Care would be needed in the selection of statistics to identify model performance (since the performance in different sub-periods could be masked when evaluating an overall period). A related issue is that the hydrological model should provide adequate performance across the range of non-stationary climate forcings to which it is subjected.

(iv) Evaluation of spatial performance – there are multiple opportunities to develop tests for spatial performance including (a) repeating the integrated test for all sites and for catchment average rainfall means it would be possible to diagnose whether specific locations or the spatial dependence causes poor reproduction of streamflow statistics; (b) developing a spatial unit test (which is analogous to the temporal unit test but extended to space) where different combinations of sites are 'spliced' in the construction of catchment average rainfall – to evaluate the impact of 'mixed' performance in the SRMs between sites on the catchment average rainfall; and (c) these spatial unit tests could be used to evaluate stochastic weather generators more generally as well as spatially distributed rainfall generators – though these would require a spatially distributed hydrological model.

## 6    Conclusions

This paper has introduced a virtual hydrologic evaluation framework that enables targeted hydrological evaluation of SRMs. The framework formalises virtual streamflow investigations by (1) using a comprehensive and systematic evaluation approach to evaluate performance (2) introducing two key innovations, an integrated test and a unit test. The integrated test compares simulated streamflow and virtual-observed streamflow to detect overall deficiencies in the ability of at-site stochastic rainfall to reproduce streamflow statistics. The unit test enables the attribution of detected streamflow errors to specific months of stochastic rainfall. The integrated and unit tests enabled different conclusions to be reached in terms of priorities for improving the rainfall model. These conclusions would not otherwise have been possible with conventional evaluation methods that focus either on rainfall statistics, or on high streamflow months. The integrated test demonstrated that while large discrepancies were identified in low rainfall months these did not translate to deficiencies in streamflow due to the dry state of the catchment. The

test also indicated instances where modelled rainfall categorised as 'good' translated to 'poor' flow due to the influence of catchment 'memory' and rainfall from prior months. The unit test identified the importance of the simulated rainfall in the transition months of May and June (late autumn/early winter) during the 'wetting-up' phase of the catchment cycle for producing low errors in subsequent high streamflow months (July/August/September) and the annual streamflow distribution.

The virtual hydrologic evaluation framework provides valuable additional diagnostic ability for the development and application of SRMs, not available by using rainfall-based evaluation techniques alone.

## 7  Acknowledgements

This work was supported by an Australian Research Council Discovery grant: A new flood design methodology for a variable and changing climate DP1094796. Additional support was provided by the CSIRO Climate Adaptation Flagship. We thank

the anonymous reviewers for their constructive comments and feedback that helped substantially improved the manuscript.

## 8  Author contributions

BSB conceived and executed the analysis, with input from MT, ML, MFL and BCB. MT, ML, MFL and BCB have contributed to assisting with method development, interpretation and analysis. BSB, MT and ML mostly wrote, reviewed and revised the paper.

## 9  Data availability

All the data used in this study can be requested by contacting the corresponding author Bree Bennett at bree.bennett@adelaide.edu.au.

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
