# Peer review of "A virtual hydrological framework for evaluation of stochastic rainfall models"

_Hydrology and Earth System Sciences, 2018_

## Referee Comment (RC1) · Anonymous Referee #1 · 17 Oct 2018

General comments:

The manuscript introduces a new framework for the evaluation of generated rainfall time series in terms of their ability to reproduce runoff time series characteristics. This is done by two tests, an integrated test and a unit test. This topic is of broad interest for the hydrological scientific community and suitable for a publication in HESS.

However, I consider the integrated test not as a novelty, since it has been applied before in different studies, but the unit test is useful for rainfall model evaluation. Hence, I suggest to move the focus to the unit test and extend the validation by other runoff characteristics. Also, the theoretical elements of the paper are very long, the application and validation of the test should be extended and there is a lack of some crucial information reagrding the applied r-r model and its calibration procedure (for details

please see my specific comments). Due to the resulting workload I suggest a major revision of the manuscript.

Specific comments:

- P2l20-23 The so-called "virtual-observed streamflow"-approach and the integrated test is not new and a widely used evaluation method, especially in data-sparse regions or research fields. For example in urban hydrology, where measured runoff characteristics are not often available, the simulation of a reference streamflow is very common (e.g. Müller and Haberlandt, 2018). The authors even mention other studies using the integrated test (Li et al., 2014, 2016). However, the unit test is interesting and indeed provides useful insights into the rainfall-runoff (r-r) transformation process. It would be useful to move the focus on this test and proof it with additional runoff characteristics, e.g. flow duration curves, not using only the monthly runoff amount. Therefore, no new simulations are necessary, only additional analyses of the existing r-r simulation results.

P2l23-25 The sentence is not clear without the explanations given in section 2. Either here more information are provided or the sentence is left out.

P3l9-14 The idea behind the example provided by the authors is clear. Nevertheless, some of the rainfall characteristics mentioned are not clear and, since it is only an example, can be left out or can be replaced by other rainfall characteristics:

- rainfall on wet days - What does this characteristic represent (the daily total rainfall itself is mentioned later)?;

- Extreme value analysis on a monthly basis and autocorrelation on an annual basis are from my understanding rather uncommon rainfall characteristics for the evaluation of rainfall time series P3l20-21 The details provided in brackets can be left out, since without reading the reference there are no additional information for the reader.

P3l7-P4l20 The motivation for the introduction of the new evaluation strategy is quite

long and can be shortened by the half. I think the majority of the community is quite aware of the issue with overlapping errors. Also Fig. 1 and Fig. 2 are quite clear from the text and could be left out. If kept, a box with "True rainfall" should be added in Fig. 1a) to be consistent with Fig. 1b ("True streamflow")

P5l10 "to match streamflow observations" -> "to match streamflow observations or statistics"

P6Table1 The authors should include a definition of the applied symbols in the caption, since the difference between "x" and "-" is not too intuitive (from only the table). Is in the last line, first column something missing (virtual hydrological. . .)?

P6Table 1 From my opinion the results from the virtual-observed streamflow approach can still be biased by the applied r-r model. For example, rainfall is generated in space and two rainfall generation methods show differences in terms of rainfall characteristics, but not in the simulated streamflow. After what I've read in the introduction and methods section, the conclusion is that the compared rainfall characteristics are then not practicable ("no impact") und useless (for the study region). But this also depends on i) the model choice (including e.g. spatial resolution, model type (fully / semi-distributed), several model approaches) and ii) the parameter identification. In a semi-distributed model differences in spatial rainfall could be dampened, while they are (maybe) not dampened in a fully-distributed model. The parameters have to be chosen a priori – a calibration on one of the rainfall data sets is not possible to avoid biases. Will the parameters be calibrated by an additional rainfall data set (the observed data) and if so, how can be avoided that this calibration introduces a bias (e.g. maybe the observed rainfall data is more similar to rainfall data set A under investigation than to B)? So all of the results depend on the chosen setup for the r-r simulations and drawn conclusions are only valid in context with the model setup and parameter set. This is of course always the case in hydrology, but it becomes more important if a virtual runoff time series is applied, since the "relation" between the model output and reality gets lost. However, the authors point these issues out later in their investigation (p20), but it

should be communicated earlier to the reader.

P9Fig3b Maybe the authors can spend a more detailed explanation of the two different indices k and t. For me the difference was not quite clear at the beginning. Also, it is clear that rainfall in June can affect the runoff in July (or from April by filling storages and hence affecting runoff in July). But how can rainfall in July affect runoff in June, although the months August to January obviously don't? Is the rainfall information transformed into runoff over such a long period in the model? Since there is no rain in the summer half year, shouldn't the storages run empty?

P9Section 2.4 It would be useful for the reader to illustrate the implementation of the framework with a flow chart, since the authors use step 1, step 6 and so on throughout the section (and the manuscript).

P10l14-17 What is the 90 % limit of the simulated statistic? If m=10 mm, everything between 1 mm and 19 mm is considered as good? Here an additional explanation is required.

P11l9-10 From Table 2 it cannot be seen, how long the time series used for the calibration of the rainfall generator are. It would be useful to the reader to characterize the time series more in detail (wet spell durations and amount, dry spell durations and maybe even on a monthly basis, since further investigations are carried out on a monthly basis). At least a hint to Fig. 6 and Fig. 7, which include some monthly observations, would be useful.

P11l16 For the calibration of the model the reader is referred to Westra et al. (2014), which is a non-reviewed technical report with 100+ pages, as far as I can see. In context with my former specific comment it is necessary to provide information in the actual manuscript, how the model has been calibrated. Which rainfall data was used for the calibration? If all 22 stations have been applied, how was the areal rainfall estimated as input for the lumped r-r model?

[Figure]

P11Section 3 Although the observed discharge time series is not used in the investigation, it would be useful for the reader to provide some runoff characteristics (e.g. mean discharge) to get a feeling for the catchment.

P11l14-18 On p9l15-17 you mention "The hydrological model should be selected on the basis that it is capable of simulating streamflow for the timescales, magnitudes and physical processes of interest to the intended application." Is the lumped model able to simulate the physical processes of a catchment with a few 100 km$^2$ catchment area (I could not find the catchment area in the manuscript).

P12l8-10 Which result is analyzed? Integrated test or unit test?

P19l19 In Fig. 5 the results for rainfall are worse than for runoff (for mean values).

P19 Results-section Before it was mentioned that also the influence of spatial rainfall patterns can be evaluated. Since this is not done in the manuscript, it can be moved to the outlooks of the manuscript. Otherwise a spatial analyses can be implemented in the manuscript (what I would recommend), to show further advantages of the unit test.

P20l20-21 With the introduced framework it is still not possible to identify, which rainfall characteristics are important for streamflow prediction. Based on the high non-linearity of the rainfall-runoff transformation process, a single rainfall characteristic cannot be sufficient to draw conclusions about the impact on the resulting runoff. If this would be the case, r-r models wouldn't have to be used anymore. However, could the authors identify, based on their analysis, which rainfall characteristics are important for the resulting runoff behaviour? (of course, the results depend on the study site, model choice and so on, but nevertheless. . .)

P21l32-p22l1 This example is hard to follow, maybe the authors can extend it. From my understanding it depends on the calibration of the storage coefficients. If storage coefficients are small, the results from the monthly rainfall will be transfered to runoff immediately. This would be possible with the "traditional" approach.

Technical corrections:

P23 There is a reference of Li et al. (2015b), but no Li et al. (2015a). Also Li et al. (2016) is mentioned before Li et al. (2015b)

References:

Müller, H., Haberlandt, U. (2018): Temporal rainfall disaggregation using a multiplicative cascade model for spatial application in urban hydrology, Journal of Hydrology 556, 847-864

Sikorska, A. E., Vivrioli, D., Seibert, J. (2018): Effective precipitation duration for runoff peaks based on catchment modelling, J. Hydrol., 556, 510–522

---

## Referee Comment (RC2) · Anonymous Referee #2 · 22 Oct 2018

The authors present a streamflow-based evaluation framework to assess the adequacy of hydrologic predictions from stochastic rainfall generators (SRGs). This is a "virtual framework" in that it benchmarks these predictions against streamflows produced by historical continuous simulations, rather than observed streamflow timeseries. The authors point out that this avoids the complicating issues of model structural errors. This is a useful approach to benchmarking SRGs, and could perhaps be applied to other fluxes of interest (not just streamflow) for which long-term observation records aren't just available.

I agree with the first reviewer, who stated that "the theoretical elements of the paper are very long". There seems to be a fair bit of repetition, or at least over-explanation, of the motivation, and I strongly recommend that the authors look closely at how Section

2 can be shortened.

Generally, I think that the demonstration would be more illuminating if the authors used it to compare two or more SRGs and/or hydrologic models.

The demonstration of these methods is provided at the monthly timescale. While this timescale might be useful for applications of water supply, it is not meaningful for flood processes in all but the very largest watersheds. It is easy to picture a hydrologic model that produces adequate performance in terms of monthly flows, but not daily or subdaily extremes, while the opposite is also possible. Similarly, it is also probably an easier task to create a stochastic rainfall generator that works well for producing monthly means and associated variability than fine-scale extremes. Thus, the virtual framework in this manuscript may not be as broadly useful for extremes as the authors claim (at least flood extremes, droughts might be a different story). I thus recommend that the authors acknowledge this shortcoming, and "tone down" the framework's purported usefulness for flood risk (e.g. page 9 lines 17), since this remains unproven.

It isn't clear how the boxplots (e.g. figure 3) are constructed. Is it the "13 errors" mentioned in page 8 line 24? Or is it somehow derived from the 10,000 synthetic rainfall years? Or the 73 years of observed data with synthetic rainfall "spliced in"? Either way, it isn't clear that the authors have avoided the proliferation of error metrics that they identify as a limitation of previous measures on page 3. If this method is applied to a large number of sites, it still seems like a not-entirely compact evaluation scheme. Perhaps the authors could clarify how this compares to other methods in this respect.

Relatedly, in Figure 6 and 7, shouldn't the "obs rain" and "virtual obs. flow" be a range, rather than a single value? There are 73 years of monthly rainfall and simulated flows. . . this variability would be valuable context for evaluating the variability of the stochastic realizations.

It is unclear how other meteorological forcings (temperature, etc.) are handled in this

framework. The authors focus on stochastic rainfall generators, as opposed to stochastic weather generators, meaning that the other forcings must be supplied independently of the rainfall. I would imagine that this could create some serious issues in some cases if synthetic rainfall is spliced together with inappropriate series of temperature or other forcings; one can imagine getting strange results in terms of precipitation vs. ET balances, with unclear consequences for the evaluation results.

I have never developed my own SRG, but I imagine that it might be hard to know exactly how to use the results from this analysis to refine that generator, despite the authors' claim that this is a valuable use of the framework. It identifies performance by month, rather than by "rainfall characteristics" (pg. 21 line 9). It is useful to know whether the SRG performs well for some months than others, but what next? If the authors plan to continue research on this topic, I would suggest that a method that "tracks" the propagation of rainfall through the model might be more effective. To me, the most clear way of doing this is to track how different rainfall statistical moments translate to different statistical moments in the streamflow, using both historical and synthetic rainfall. Such an approach would be amenable to changing the evaluation timescale. For these reasons, I recommend that the authors delete the statement that this framework "should be an essential step in the development and application of stochastic rainfall models" (page 21 line 22-23). On a related note, the authors should comment on how this technique would apply to distributed (i.e. high-resolution gridded) SRGs and hydrologic models.

I wonder if this framework should consider the autocorrelation in monthly rainfalls when doing this splicing. I don't know too much about the climate of South Australia, but I can imagine that autocorrelation at least in dry periods can be quite important, and this is likely not preserved during the splicing. It's not clear what the implications would be for the resulting evaluation.

Figure 3 and elsewhere: I don't understand what "(90% limits shown)" means.

Section 3: Mention basin size. Also, why are stations outside the watershed used? More importantly, is the rainfall hydroclimate stationary? If not, then it seems as though this whole issue of stochastic generation and comparison of resulting streamflows against a nonstationary continuous simulation would be more complicated. Please comment on this.

Minor comments: Page 1 line 9: change "is" to "has been" Page 1 line 10: change "is given" to "has been paid" Page 1 line 12: delete "whenever the simulated rainfall are poor" Page 1 line 15: change "months" to "seasons"-that is a more broadly relevant term. Hydrology varies seasonally, months are an arbitrary construct (this comment applies elsewhere in the paper, such as page 2 line 31) Page 1 line 19: change "catchment cycle" to "annual hydrologic cycle" Throughout paper: I recommend introducing an acronym for stochastic rainfall models and using it throughout. Page 1 line 24-25: I recommend deleting "risks" after "floods" and "droughts" and changing it to "hazards" after "hydrologic" Page 1 line 28: delete comma after "targeted" Page 2 line 10: "and/or" is not appropriate in technical writing. Use "or" Page 2 line 12: put "virtual experiments" in quotations when mentioned for the first time, for emphasis Page 3 line 16: add comma after "poor" Page 5 line 10: The goal is not to match streamflow observations. It is to match the statistics of streamflows Page 3 line 12: add "model" before parameters Page 3 line 13: Why would you call ET "extraneous"? It is generally very very important. Page 10 line 15: I think that "observed/virtual" is a strange term. Observations have very little usage in this study... Page 11 line 16: grammar problem "was fit good" Page 14 line 14: This sentence is a bit awkward. It isn't perhaps so "common and obvious" to the reader. Page 19 line 3-4: streamflow arises from more than just rainfall integration over a catchment area-what about ET, etc.? Page 20 line 11-13: I don't understand this sentence. Certainly model performance depends on the chosen model. Page 20 line 13: good place to mention that multiple SRGs could be used too, not just multiple hydrologic models. Page 20 line 18: put "memory" in quotations.

---

## Referee Comment (RC3) · Anonymous Referee #3 · 2 Nov 2018

**OVERVIEW**

The manuscript proposes a "virtual hydrological" framework useful for the performance evaluation of stochastic rainfall generators (SRGs). Differently from other studies involved on this topic, this work proposes 1) to evaluate the rainfall performances directly in terms of discharge by considering as benchmark the "virtual observed streamflow", i.e. the streamflow obtained by running the observed rainfall into the hydrological model, 2) to use two different tests to highlight discrepancies between observed and

**GENERAL COMMENTS**

Although the topic is surely of interest for the readership of HESS, a major revision is required before to consider the manuscript suitable for the publication. Indeed, throughout the manuscript some important information are missing (for details see specific comments below) whereas the section 2 and details about the virtual hydrological framework should be shortened. Moreover, the outcomes of this study seem linked to the specific case study and the authors should discuss how the results could be generalized for different SRGs and hydrological models.

Abstract section. This section should be made clearer concerning both the explanation of the virtual hydrological framework features and the results obtained in the work. Specifically, lines 12-15 and 18-20 in page 1 are not clear without reading the paper.

Section 2. This section should be shortened deleting multiple repetitions about the framework description in sub-sections 2.2 and 2.3. Moreover, Figure 1 and 2 could be merged into one figure. Conversely, the section 2.4 should be improved (also adding a flowchart) to allow the readers to easily follow the section "results".

Section 3. Some important details are missing in this section. In addition to the area of the catchment and the temporal resolution of the simulated rainfall, the authors should specify how the GR4J model is forced by observed rainfall and how it is calibrated. Is the observed catchment average rainfall used to force and calibrate the hydrological model? How many years of observed discharge data are used for calibration? is this set of parameters used to simulate streamflow within the unit and integrated test?

Finally, major details should be added to this section about the rainfall statistics used for the calibration of the SRG of Bennet et. al. (2018).

HESSD
Section 4. In this section the authors should address the following points:

1) as the authors know, the rainfall simulated by the SRG are function of the rainfall statistic properties used to estimate the model parameters. According to the authors, in which way the rainfall statistical properties and the results obtained by the unit and integrated test are linked? if different rainfall statistical properties are used for the SRG calibration, are the results different? For instance, is the identification of the 10 "poor" sites sensitive a variation of rainfall statistics? If different statistics are used for SRG calibration, is it possible to reduce the streamflow errors? These aspects should be demonstrated/discussed by the authors in order to provide to the readers a general framework not tailored for a specific case study.

2) as the streamflow generation is a results of the mean areal (rather than single-site) rainfall over the basin, before to apply the integrated test to identify sites for which the rainfall simulation is not good, it could be interesting to estimate the streamflow errors coming from the mean areal rainfall, evaluated as average over the 22 sites. How good are these streamflow time series with respect to the "virtual-observed" ones (obtained by the rainfall observed over the 22 sites and averaged over the catchment)? More interesting, the authors should highlight the benefits deriving from the use of integrated test. For instance, they should show what are the streamflow errors if only the 12 "good" sites are retained to evaluate the mean areal rainfall. Is it better than using all 22 sites?

**SPECIFIC COMMENTS**

Page 2, lines 19-21: the example of Bennet et al. (2018) is not clear without reading the paper.

Page 7, lines 14-15: This sentence is not clear. Please rephrase it.

Page 8, line 24: Why the authors write "13 errors to compare"? Are the authors considering also the integrated test? It is not clear.
Page 8, lines 27-28. This example related to the integrated test is difficult to understand in this section. The reason is that in the previous section, where the authors describe the test there is no mention to the fact that the evaluation of the integrated test is carried out also at the monthly scale.

Figure 4: the position of the streamflow gauging station should be added in the figure.

Page 15, lines 19-20: the conclusion about the transitional months should be drawn carefully. Indeed, moving from dry to wet conditions the process of formation of flood is very sensitive to the antecedent soil moisture conditions. Is the hydrological model able to reproduce observed streamflow in the transition period?

---

## Author Comment (AC1) · 15 Nov 2018

**Response to Reviewer 1**

> **Comment 1:**
>
> *The manuscript introduces a new framework for the evaluation of generated rainfall time series in terms of their ability to reproduce runoff time series characteristics. This is done by two tests, an integrated test and a unit test. This topic is of broad interest for the hydrological scientific community and suitable for a publication in HESS.*
>
> *However, I consider the integrated test not as a novelty, since it has been applied before in different studies, but the unit test is useful for rainfall model evaluation. Hence, I suggest to move the focus to the unit test and extend the validation by other runoff characteristics. Also, the theoretical elements of the paper are very long, the application and validation of the test should be extended and there is a lack of some crucial information regarding the applied r-r model and its calibration procedure (for details please see my specific comments). Due to the resulting workload I suggest a major revision of the manuscript.*

**Response 1:** We are pleased that the reviewer found our manuscript suitable and we are grateful for the insightful and constructive comments. They have been very helpful, thank you. The suggestions provided show careful consideration and will lead to an improved revision of the manuscript. Regarding the specific matters raised:

*Novelty of the integrated test* – we will provide better referencing and discussion on existing uses of virtual-observed streamflow evaluation (calibration, validation, model selection and diagnosis) in the introduction, including the rewording presentation of objective 2. However, we feel that there are some important aspects of our implementation of our framework that distinguish it from existing presentations of virtual-observed streamflow evaluations. In particular, the presentation of the integrated test in this paper is the first time, a virtual-observed streamflow evaluation has been formalised used using a comprehensive and systematic evaluation (CASE) framework approach (Bennett et al. 2018). This distinguishing feature and others are further discussed in response to comment 2. We will take the reviewer's advice and emphasize the novelty of the unit test and its diagnostic ability in the revised manuscript.

*Evaluation using other runoff characteristics* – we will examine if additional runoff characteristics such as flow duration curves provide additional insight on the deficiencies of the rainfall model, over and above what is already presented and incorporate discussion of these insights where appropriate.

*Length of theoretical elements* – we will reduce the length of the relevant sections.

*Information on the r-r model* – we will provide better explanation, including references of the calibration and validation procedure of this rainfall-runoff model.

We further elaborate on these items in response to subsequent comments made.

**Comment 2:**

*P2l20-23: The so-called "virtual-observed streamflow"-approach and the integrated test is not new and a widely used evaluation method, especially in data-sparse regions or research fields. For example in urban hydrology, where measured runoff characteristics are not often available, the simulation of a reference streamflow is very common (e.g. Müller and Haberlandt, 2018). The authors even mention other studies using the integrated test (Li et al., 2014, 2016). However, the unit test is interesting and indeed provides useful insights into the rainfall-runoff (r-r) transformation process.*

**Response 2:**

Thank you for supplying the references: Müller and Haberlandt, 2018; Sikorska et al. 2018. We will include them in the revised text along with additional references (e.g. Kim and Olivera, 2012).

We agree that the concept of a virtual-observed streamflow evaluation is not new and we will revise the introduction, including the presentation of objective 2, to make this clear and discuss that the approach has been used in a variety of contexts (e.g. calibration, validation, model selection and as a diagnostic tool).[1] However, there are some important aspects of our framework that distinguish it from existing presentations of virtual-observed streamflow evaluation, as outlined below.

1. This is the first time the virtual-observed streamflow evaluation approach has been formalised using a comprehensive and systematic evaluation (CASE) framework (pioneered by Bennett et al., 2018 and used by Evin et al. 2018, Khedhaouiria et al. 2018) to evaluate stochastic rainfall models in terms of the ability to produce key runoff statistics of interest. The integrated tests presented in this paper follow the CASE approach because they (i) present a comprehensive range of key statistics of interest, (ii) systematically categorise performance at specific spatial and temporal scales using quantitative criteria for each statistic, and (iii) systematically categorise aggregate performance over multiple spatial and/or temporal scales.

Previous papers (Müller and Haberlandt, 2018; Sikorska et al. 2018, Kim and Olivera, 2012) have used a virtual-observed streamflow evaluation approach, but have not used a CASE framework to evaluate the performance of stochastic rainfall model at multiple rainfall sites in terms of its ability capture key streamflow statistics of interest. For example, Müller and Haberlandt (2018) established the need for spatial consistency of rainfall generation in modelling sewer networks by comparing rainfall disaggregation approaches with or without spatial consistency. This virtual-observed streamflow evaluation is performed for identified extreme rainfall events only and therefore does not use a CASE approach that considers multiple temporal scales and the longer term effects of the applied rainfall on the translation of subsequent rainfall to streamflow. Sikorska et al. (2018)
* * *
[1] We will rewrite the literature review to point out that this test is not new and that it has been employed in a variety of contexts, including:

*Calibration* – Using virtual streamflow to directly improve the calibration of a rainfall model. For example, Kim & Olivera (2012) derived weights to reflect the importance of various rainfall statistics in terms of streamflow. As another example, Li et al. (2014, 2016) used catchment simulations to estimate soil moisture distributions as part of a new technique to derive flood frequencies.

*Validation* – to establish a model as fit-for-purpose together with other validation tests (Kim & Olivera, 2012).

*Model selection* – to identify key rainfall features of multiple competing models or model options in terms of hydrological behaviour. For example, Müller and Haberlandt (2018) established the need for spatial consistency of rainfall generation in modelling sewer networks.

*Diagnostic* – to identify rainfall features of interest in a given rainfall model in terms of hydrological behaviour. For example, Sikorska et al. (2018) found that detailed rainfall time series were not needed to reproduce peaks in the modelled catchments and that simple rainfall disaggregation approaches were sufficient.

focused on identifying rainfall features of interest in terms of resultant hydrological behaviour for the purposes of determining the effective daily precipitation duration with a view to selecting a suitable rainfall disaggregation scheme. Their evaluations determined that detailed temporal rainfall time series were not needed to reproduce annual or seasonal peaks in their modelled catchments. Although the evaluations presented are comprehensive the motivation of the Sikorska et al. (2018) paper is different and does not provide a general formalised framework for systematically categorising stochastic rainfall model performance at specific and aggregate temporal and spatial scales. Kim & Olivera (2012) used virtual-observed streamflow evaluation as part of a larger calibration and validation approach in which various weights were trialled to reflect the importance of various rainfall statistics within a modified Bartlett-Lewis rectangular pulse (MBLRP) model. However the focus was on the improvement and validation of the MBLRP model rather than the presentation of separate framework for model evaluation. Finally, Li et al. (2014, 2016) used a virtual-observed streamflow evaluation approach, to evaluate the ability of range of techniques to estimate the derived annual flood frequency distribution - they did not use a CASE approach to evaluate stochastic rainfall models. In the revised paper we will improve the presentation of the approach, highlighting the key points above to more clearly demonstrate the novelty.

Additionally, the formalisation of virtual-observed streamflow evaluation using a comprehensive and systematic evaluation (CASE) approach, the integrated test, forms a baseline for subsequent application of the unit test which has greater ability to pinpoint issues with respect to the source of the rainfall error on a monthly basis.

2. As identified by the reviewer, we introduce an innovative unit test, which has never been used before in a virtual-observed streamflow evaluation approach. The key advantage of this unit test is that by splicing together the observed and simulated rainfall in a systematic manner, it is able to develop new insights on which months have deficiencies in simulated rainfall that produce poor runoff performance. We will put greater emphasis on this new innovative unit test in the revised manuscript.
* * *
**Comment 3:**

*It would be useful to move the focus on this test and proof it with additional runoff characteristics, e.g. flow duration curves, not using only the monthly runoff amount. Therefore, no new simulations are necessary, only additional analyses of the existing r-r simulation results.*
* * *
**Response 3:**

Good idea, we will examine if additional runoff characteristics such as flow duration curves provide additional insight on the deficiencies of the rainfall model, over and above what is already presented. Where appropriate, we will add them to the manuscript and/or supplementary material with additional discussion.
* * *
**Comment 4:**

*P2l23-25 The sentence is not clear without the explanations given in section 2. Either here more information are provided or the sentence is left out.*
* * *
**Response 4:**

Thank you. We will leave out the sentence.

**Comment 5:**

*P3l9-14 The idea behind the example provided by the authors is clear. Nevertheless, some of the rainfall characteristics mentioned are not clear and, since it is only an example, can be left out or can be replaced by other rainfall characteristics:*

*- rainfall on wet days - What does this characteristic represent (the daily total rainfall itself is mentioned later)?;*

*- Extreme value analysis on a monthly basis and autocorrelation on an annual basis are from my understanding rather uncommon rainfall characteristics for the evaluation of rainfall time series*

**Response 5:**

Thank you. We will modify the example to be clearer, by only using rainfall statistics that are well-known and require no additional explanation.

To address the specific questions regarding our original choice of statistics, our interest in some of these rainfall statistics arises from our context. For example, (i) rainfall on wet days is important because the calibration should match the moments of the truncated and power-transformed Gaussian upper tail; (ii) extreme value analysis of months is of interest to strongly seasonal locations (e.g. Leonard et al., 2008); and (iii) auto-correlation of annual totals is of interest due to teleconnections in the rainfall signal (Thyer and Kuczera, 2000).

**Comment 5:**

*P3l20-21 The details provided in brackets can be left out, since without reading the reference there are no additional information for the reader.*

**Response 5:**

We agree that the details in brackets can be left out and will do so in the revised manuscript.

**Comment 7:**

*P3l7-P4l20 The motivation for the introduction of the new evaluation strategy is quite long and can be shortened by the half. I think the majority of the community is quite aware of the issue with overlapping errors. Also Fig. 1 and Fig. 2 are quite clear from the text and could be left out. If kept, a box with "True rainfall" should be added in Fig.1a) to be consistent with Fig. 1b ("True streamflow")*

**Response 7:**

We will shorten the explanation while maintaining the key points of the introduction. Based on our experiences explaining this work, we feel the figures in section 2 are helpful to avoid misconceptions. We prefer to retain them and will amended them as suggested.

**Comment 8:**

**P5l10 "to match streamflow observations" -> "to match streamflow observations or statistics"**

**Response 8:**

Thank you. We will modify the sentence as suggested.

**Comment 9:**

*P6Table1 The authors should include a definition of the applied symbols in the caption, since the difference between "x" and "-" is not too intuitive (from only the table). Is in the last line, first column something missing (virtual hydrological…)?*

**Response 9:**

Thank you for this suggestion. Table 1 will be revised so that text is used in place of the original symbols ('Yes', 'No', 'Not Applicable'). The Table caption will also be amended to clarify this also (i.e. 'Yes' indicates that a source of error in included in the evaluation, 'Not Applicable' indicates that a source of error is not relevant to the evaluation and 'No' indicates a source of error is not included in the evaluation). The last line, first column will be amended to read 'virtual hydrological evaluation'.

**Comment 10:**

*P6Table 1 From my opinion the results from the virtual-observed streamflow approach can still be biased by the applied r-r model. For example, rainfall is generated in space and two rainfall generation methods show differences in terms of rainfall characteristics, but not in the simulated streamflow. After what I've read in the introduction and methods section, the conclusion is that the compared rainfall characteristics are then not practicable ("no impact") und useless (for the study region). But this also depends on i) the model choice (including e.g. spatial resolution, model type (fully / semi-distributed), several model approaches) and ii) the parameter identification. In a semi-distributed model differences in spatial rainfall could be dampened, while they are (maybe) not dampened in a fully-distributed model. The parameters have to be chosen a priori – a calibration on one of the rainfall data sets is not possible to avoid biases. Will the parameters be calibrated by an additional rainfall data set (the observed data) and if so, how can be avoided that this calibration introduces a bias (e.g. maybe the observed rainfall data is more similar to rainfall data set A under investigation than to B)? So all of the results depend on the chosen setup for the r-r simulations and drawn conclusions are only valid in context with the model setup and parameter set. This is of course always the case in hydrology, but it becomes more important if a virtual runoff time series is applied, since the "relation" between the model output and reality gets lost. However, the authors point these issues out later in their investigation (p20), but it should be communicated earlier to the reader.*

**Response 10:**

The reviewer has raised some excellent discussion points. The immediate response is that while we have discussed some of these points later in the investigation (Section 5.2), we will communicate the key issues earlier to the reader. We appreciate the centrality of the issue raised and will highlight it in Section 2.2. We provide specific responses below to the discussion points raised.

1. "*Virtual-observed streamflow approach can still be biased by the applied r-r model*" – Yes, we agree. This is a very important matter to consider.
2. "*Differences in terms of rainfall characteristics, but not in the simulated streamflow*" – Yes, there is the potential that a chosen rainfall-runoff model is insensitive to certain important differences in modelled rainfall. Further to this, it is important to mention that a unit test can still get insights using a lumped rainfall-runoff model. This initial test is a necessary, but not sufficient condition for spatial rainfall models. If a rainfall model, cannot get the virtual-observed streamflow statistics from a well calibrated, well-known lumped rainfall-runoff model right, there is limited value in examining the spatial statistics.
3. "*But this also depends on i) the model choice … and ii) the parameter identification*" – Yes, as with the comments in (2), all elements of the modelling method can potentially introduce bias. The end-user's impact of interest and associated modelling process can influence an outcome. These observations reinforce the need for care when applying the framework. We have chosen a widely applied model, GR4J. We also adopted rigorous calibration which is presented compactly in a journal paper (Westra

et al. 2014a), but with a detailed report also available (Westra et al. 2014b). The calibration of the hydrological model is further discussed in the response to comment 15.

4. "*If so, how can be avoided that this calibration introduces a bias*" – This is an excellent question, which we will address with further discussion. While best-practice models and methods are important, this does not necessarily guard against the possibility that a model poorly represents key processes of interest. One remedy for this limitation would be to use multiple rainfall-runoff models and this is discussed in Section 5.2.
* * *
**Comment 11:**

*P9Fig3b Maybe the authors can spend a more detailed explanation of the two different indices k and t. For me the difference was not quite clear at the beginning. Also, it is clear that rainfall in June can affect the runoff in July (or from April by filling storages and hence affecting runoff in July). But how can rainfall in July affect runoff in June, although the months August to January obviously don't? Is the rainfall information transformed into runoff over such a long period in the model? Since there is no rain in the summer half year, shouldn't the storages run empty?*
* * *
**Response 11:**

We will more clearly indicate the meaning of indices $k$ and $t$ in the descriptive text, Figure 3b and its caption to aid the reader.

A unit test is undertaken by evaluating the ensemble of simulated streamflows from transforming the spliced rainfall from the 12 potential influencing months for an evaluated month, $t$. We believe it is necessary to evaluate all 12 potential influencing months because *a priori* the impacts of 'poor' rainfall can have long-term impacts on streamflow statistics due to catchment storage in the rainfall-runoff model. Some catchment models have short-term stores to represent features such as depressions, basins, and channels, but other catchments models have long-term stores to represent the long-term memory in subsurface catchment storages) that can have memory over multiple months. For the case study catchment the storages do not run empty each year in summer, so there is potential for persistence at longer timescales due to this 'memory' in the catchment. Therefore, it is plausible that rainfall from 12 months prior can influence the current state of a catchment (especially if that month/season was anomalously wet or dry). An additional figure of monthly rainfall and streamflow boxplots will be provided to illustrate the highly seasonal nature of the case study catchment in Section 3 (also see response to comment 14).

**Comment 12:**

*P9Section 2.4 It would be useful for the reader to illustrate the implementation of the framework with a flow chart, since the authors use step 1, step 6 and so on throughout the section (and the manuscript).*

**Response 12:**

We agree, good idea. We will incorporate a flow chart to illustrate the implementation of the framework in Section 2.4.

**Comment 13:**
*P10l14-17 What is the 90 % limit of the simulated statistic? If m=10 mm, everything between 1 mm and 19 mm is considered as good? Here an additional explanation is required.*

**Response 13:**
Thank you. We will add additional information to explain the 90% limit test as requested both in text and graphically (potentially as supplementary material). The relevant information is available in Bennett et al. (2018), but to be more accessible this information will be reproduced in the current paper.

**Comment 14:**
*P11l9-10 From Table 2 it cannot be seen, how long the time series used for the calibration of the rainfall generator are. It would be useful to the reader to characterize the time series more in detail (wet spell durations and amount, dry spell durations and maybe even on a monthly basis, since further investigations are carried out on a monthly basis). At least a hint to Fig. 6 and Fig. 7, which include some monthly observations, would be useful.*

**Response 14:**
We agree with the reviewer and recognise that the high-level summaries need more tangible details on the rainfall and streamflow statistics on a monthly basis to help the reader understand the seasonal behaviour of the case study catchment. We will revise Table 2 to characterise the rainfall time series in more detail including the addition of columns that present rainfall statistics (total rainfall, no. of wet days, average daily rainfall, average wet day length, average dry spell durations) in different seasons – for brevity in this table we will show two months: January to represent the dry summer and July to represent the wet winter. Further detail on these statistics for all months at each site will be also provided as supplementary material. In addition, a new figure that shows seasonal variation of catchment average rainfall and streamflow on a monthly basis will be added to Section 3 (Case Study) of the main paper to address the suggestion of the reviewer.

**Comment 15:**
*P11l16 For the calibration of the model the reader is referred to Westra et al. (2014), which is a non-reviewed technical report with 100+ pages, as far as I can see. In context with my former specific comment it is necessary to provide information in the actual manuscript, how the model has been calibrated. Which rainfall data was used for the calibration? If all 22 stations have been applied, how was the areal rainfall estimated as input for the lumped r-r model?*

**Response 15:**
A paper (Westra et al., 2014a) will now be cited alongside the report. The paper provides a compact peer-reviewed summary of the model and its calibration – for a neighbouring catchment (Scott Creek). This paper was acknowledged with a Research Spotlight Award from American Geophysical Union (top 5% of papers in AGU). The reference to the report, Westra et al. (2014b), is also retained since it gives details specific to the Onkaparinga catchment used in this paper and because it is comprehensive. The Scott Creek and Onkaparinga catchments were calibrated as part of the same project using consistent models and techniques.

The model development and calibration was comprehensive and considered a range of aspects including multiple sources of uncertainty (input, output, parameters, etc.)[2] and it is beyond of the scope of this paper to include all the details. Instead relevant aspects of the calibration and model-selection will be added to the manuscript. The relevant features to be included will be: calibration approach, including parameter optimisation method and objective used, calibration and validation results (NSE etc.), and an explanation of the rainfall and runoff data used for calibration and validation.
* * *
**Comment 16:**
*P11Section 3 Although the observed discharge time series is not used in the investigation, it would be useful for the reader to provide some runoff characteristics (e.g. mean discharge) to get a feeling for the catchment.*
* * *
**Response 16:**
Thanks. Details of the catchment's runoff characteristics at the annual and seasonal level will be added to the revised Table 2 and the new figure (see response to comment 13).
* * *
**Comment 17:**
*P11l14-18 On p9l15-17 you mention "The hydrological model should be selected on the basis that it is capable of simulating streamflow for the timescales, magnitudes and physical processes of interest to the intended application." Is the lumped model able to simulate the physical processes of a catchment with a few 100 km2 catchment area (I could not find the catchment area in the manuscript).*
* * *
**Response 17:**
The catchment area is 323 km$^2$ and will be mentioned in the revision (Section 3).

It is important that the chosen hydrological model is fit for purpose (see also discussion of comment 10). The GR4J model used in this paper is for catchment inflows to the Mount Bold reservoir and is appropriate for analysis of catchment yield (i.e. focussed on means and variances of inflow)[3]. However, if we were examining impacts on instantaneous peak flows impacts, this model would not be suitable and if we wanted to look at impacts of distributed rainfall, we would need a distributed rainfall-runoff model. However, for the purpose of this paper, which is to demonstrate the virtual hydrological framework (including the unit test) for evaluating the ability of a stochastic rainfall model to estimate catchment yield, the model is deemed sufficient.
* * *
[2] The hydrological model calibration considered 24 model variants combined with likelihood estimation of a heteroskedastic error model. The calibration separated out multiple sources of uncertainty (input uncertainty from gauges and radar, output uncertainty associated with streamflow gauges, and model uncertainty). All 22 rainfall stations were used to estimate areal rainfall. The areal rainfall was interpolated using kriging with external drift on a daily basis using a similar latent-variable Gaussian model as the stochastic model from Bennett et al. (2018). The areal rainfall estimation was performed using Thiessen weights for comparison. The number of gauges is relatively dense and the uncertainty due to rainfall inputs was also assessed relative to other sources of uncertainty (Westra et al. 2014b; pg. 7).

[3] The GR4J model has a calibrated Nash Sutcliffe of 0.8 (reported in the original manuscript). This model (and its non-stationarity variants) were used to project climate change impact on the Onkaparinga catchment (Westra et al. 2014b).

**Comment 18:**
*P12l8-10 Which result is analyzed? Integrated test or unit test?*

**Response 18:**

We agree this is not clear, thank you for pointing it out. The result is from the integrated test (step 4). This will be clarified in the revised text.

**Comment 19:**

*P19l19 In Fig. 5 the results for rainfall are worse than for runoff (for mean values).*

**Response 19:**
There are many interesting features of Fig 5 like this. The fact that there is not a direct correspondence between 'good' rainfall and 'good' runoff, and/or 'poor' rainfall and 'poor' runoff is one of the motivations for the virtual hydrological evaluation framework in addition to observed rainfall-based evaluation. Figure 5 shows that it is possible for seemingly 'poor' rainfall to yield 'good' runoff (as also 'good' rainfall can yield 'poor' runoff). We note that the discrepancy in Figure 5 is not in terms of mean values, but for the standard deviation of monthly aggregates (see Figure 5, *sd*(total) for rainfall and runoff in Jan, Mar, May, Jun, Oct, Nov, Dec). For drier months (Nov-Mar) the lack of correspondence (i.e. 'poor' rainfall producing 'good' runoff) is due to the low amount of runoff. While in wetter months (May-Oct) the relationship is more complicated as shown in the unit test demonstrations (Section 4.2). This will be fully explained in the revised Section 4.1 of the paper.

**Comment 20:**
*P19 Results-section Before it was mentioned that also the influence of spatial rainfall patterns can be evaluated. Since this is not done in the manuscript, it can be moved to the outlooks of the manuscript. Otherwise a spatial analyses can be implemented in the manuscript (what I would recommend), to show further advantages of the unit test.*

**Response 20:**
This concept has not been demonstrated in the main paper, therefore it will be deferred to the discussion on outlooks (Section 5.2). We believe the unit test has sufficient novelty to represent a substantial contribution, hence we will the leave spatial rainfall evaluation for future developments.

**Comment 21:**

*P20l20-21 With the introduced framework it is still not possible to identify, which rainfall characteristics are important for streamflow prediction. Based on the high non-linearity of the rainfall-runoff transformation process, a single rainfall characteristic cannot be sufficient to draw conclusions about the impact on the resulting runoff. If this would be the case, r-r models wouldn't have to be used anymore. However, could the authors identify, based on their analysis, which rainfall characteristics are important for the resulting runoff behaviour? (of course, the results depend on the study site, model choice and so on, but nevertheless…)*

**Response 21:**
The comment is correct that the proposed method does not identify the impact of specific singular rainfall characteristics on the resulting runoff. However, the framework does provide a clear approach to isolate which set of components of the rainfall model require further attention. The integrated test focuses attention on hydrological properties, and the unit test can isolate deficiencies in rainfall by month. When applied to the case study in our paper, the limitations of the model are in the variability of the rainfall and not in the rainfall mean. The initial motivation for the approach can be seen in Figure 5 where the mean of the annual rainfall is 'good', but the mean of the annual runoff is 'poor'. Figure 5 also shows that this is mostly attributed to 'poor' mean runoff in June, July and August. Unit tests were then used to show that the rainfall in the catchment 'wetting-up' period (May-June) is of key importance. This is greater insight than could have been achieved with observed-rainfall evaluation and is greater insight than could be gathered from other virtual-observed streamflow approaches.

However, we agree with the reviewer that the framework cannot currently distinguish between particular features of the rainfall (e.g. "Is it rainfall correlation, magnitude, or intermittency that causes a low standard deviation in monthly streamflow?"). Nonetheless, the framework has significant potential to be extended to diagnose which are rainfall characteristics are. This can be done by comparing multiple rainfall model variants (parametrically, or via bootstrap techniques) which are designed to have contrasting features of a key characteristic (e.g. intermittency, rainfall correlation). Such an approach was undertaken by Evin et al. (2018) using an observed-rainfall evaluation approach to compare model variants. In the revised paper, this will now be identified as a limitation and this extension will be highlighted for further research.

**Comment 22:**
*P21l32-p22l1 This example is hard to follow, maybe the authors can extend it. From my understanding it depends on the calibration of the storage coefficients. If storage coefficients are small, the results from the monthly rainfall will be transferred to runoff immediately. This would be possible with the "traditional" approach.*

**Response 22:**
Thank you for pointing out that the example is hard to follow. We will add additional information on catchment seasonality in the case study description to better explain the importance of the 'wetting-up' months and storage in the catchment (also see responses to comments 14 and 16). We will then revisit this example to explain the concept more concretely.

**Comment 23:**
*P23 There is a reference of Li et al. (2015b), but no Li et al. (2015a). Also Li et al. (2016) is mentioned before Li et al. (2015b)*

**Response 23:**
Thank you, this will be corrected.

**References**

Bennett, B., Thyer, M., Leonard, M., Lambert, M., and Bates, B. (2018). A comprehensive and systematic evaluation framework for a parsimonious daily rainfall field model, Journal of Hydrology, 556, 1123-1138.

Evin, G., Favre, A.-C., and Hingray, B. (2018): Stochastic generation of multi-site daily precipitation focusing on extreme events, Hydrology and Earth System Sciences, 22, 655-672, 2018.

Khedhaouiria, D., Mailhot, A. and Favre, A.C. (2018). Daily Precipitation Fields Modeling across the Great Lakes Region (Canada) by Using the CFSR Reanalysis. Journal of Applied Meteorology and Climatology, 57(10), pp.2419-2438.

Kim, D. and F. Olivera (2012). "Relative Importance of the Different Rainfall Statistics in the Calibration of Stochastic Rainfall Generation Models." Journal of Hydrologic Engineering 17(3): 368-376.

Leonard, M., Metcalfe, A. and Lambert, M. (2008) Frequency analysis of rainfall and streamflow extremes accounting for seasonal and climatic partitions. Journal of hydrology, 348(1-2), pp.135-147.

Li, J., Thyer, M., Lambert, M., Kuczera, G., and Metcalfe, A (2014): An efficient causative event-based approach for deriving the annual flood frequency distribution, J Hydrol, 510, 412-423.

Li, J., Thyer, M., Lambert, M., Kuzera, G., Metcalfe, A., 2016. Incorporating seasonality into event-basedjoint probability methods for predicting flood frequency: A hybrid causative event approach. J. Hydrol., 533: 40-52.

Sikorska, A. E., Vivrioli, D., Seibert, J. (2018): Effective precipitation duration for runoff peaks based on catchment modelling, J. Hydrol., 556, 510–522

Thyer, M. and Kuczera, G. (2000). Modeling long-term persistence in hydroclimatic time series using a hidden state Markov Model. Water resources research, 36(11), pp.3301-3310.

Westra, S., Thyer, M. Leonard, M., Kavetski, D. and Lambert, M. (2014a), A strategy for diagnosing and interpreting hydrological model nonstationarity, Water Resour. Res., 50, 5090–5113, doi:10.1002/2013WR014719.

Westra, S., Thyer, M., Leonard, M., Kavetski, D., and Lambert, M. (2014b) Impacts of climate change on surface water in the Onkaparinga catchment-Final report volume 1: hydrological model development and sources of uncertainty, 1839-2725.

---

## Author Comment (AC2) · 19 Nov 2018

**Response to Reviewer 2**

**Major Comments**

| |
|---|
| **Comment 1:** |
| *The authors present a streamflow-based evaluation framework to assess the adequacy of hydrologic predictions from stochastic rainfall generators (SRGs). This is a "virtual framework" in that it benchmarks these predictions against streamflows produced by historical continuous simulations, rather than observed streamflow timeseries. The authors point out that this avoids the complicating issues of model structural errors. This is a useful approach to benchmarking SRGs, and could perhaps be applied to other fluxes of interest (not just streamflow) for which long-term observation records aren't just available.* |

**Response 1:**

Thank you. We are pleased that the reviewer found the approach useful for evaluating stochastic rainfall models and appreciated the wider potential application of the approach for other fluxes of interest.

| |
|---|
| **Comment 2:** |
| *I agree with the first reviewer, who stated that "the theoretical elements of the paper are very long". There seems to be a fair bit of repetition, or at least over-explanation, of the motivation, and I strongly recommend that the authors look closely at how Section 2 can be shortened.* |

**Response 2:**

Thank you, we will shorten Section 2 while retaining the key points.

| |
|---|
| **Comment 3:** |
| *Generally, I think that the demonstration would be more illuminating if the authors used it to compare two or more SRGs and/or hydrologic models.* |

**Response 3:**

We support this idea. In the future the framework can be used to compare two or more SRG's for particular hydrological applications. Furthermore, by utilising two or more hydrological models in the virtual evaluation framework, it would reduce the dependence on the choice of hydrological model (which was raised by reviewer #1 see comment 10), because one could look for patterns of errors for a single SRG across two hydrological models.

However, our preference is not to include this in this paper, for the following reasons:

1. To include the details of a second SRG and/or a second hydrological model, as well as providing a complete explanation of the details of the framework, observed-rainfall evaluation, virtual observed streamflow evaluation, two different tests, the integrated and unit tests, would make the paper and/or analysis overly long. Reviewer #1 has asked for extra details and additional figures/tables to explain a wide range of details – including the hydrological model calibration (see reviewer #1 comment 15), extra streamflow analysis (see reviewer #1 comment 3)[1]. This is over and above the seven figures already included. If we included another SRG and hydrological model, the number of the figures could
* * *
[1] At least two additional figures will be incorporated to provide information on the seasonality of the catchment (rainfall and runoff). Table 2 will also be extended to characterise the rainfall time series at each site in more detail and supplementary material provided to give further detail again. Additionally, we will be examining the flow duration curves to see if they provide additional insight on the deficiencies of the rainfall model, over and above what is already presented. Where appropriate, we will add them to the manuscript and/or supplementary material with additional discussion.

increase dramatically making the paper overly long and lose focus on the presentation of the framework.

2. The framework has not been presented before, in particular the unit test. Therefore, in the manuscript we present the evaluation of a single stochastic rainfall model to demonstrate the framework. This application to a single model has demonstrated some new insights; that the errors in streamflow for a particular month can be affected by errors in the in rainfall from the previous 2 to 3 months. This innovation is something that has not been previously identified in the literature.

Once the framework has been established and explained in this paper, future work will undertake multiple SRG and/or multi-hydrological model comparisons, as suggested by the reviewer. A comment on this will be incorporated into the discussion (Section 5).
* * *
**Comment 4:**

*The demonstration of these methods is provided at the monthly timescale. While this timescale might be useful for applications of water supply, it is not meaningful for flood processes in all but the very largest watersheds. It is easy to picture a hydrologic model that produces adequate performance in terms of monthly flows, but not daily or subdaily extremes, while the opposite is also possible. Similarly, it is also probably an easier task to create a stochastic rainfall generator that works well for producing monthly means and associated variability than fine-scale extremes. Thus, the virtual framework in this manuscript may not be as broadly useful for extremes as the authors claim (at least flood extremes, droughts might be a different story). I thus recommend that the authors acknowledge this shortcoming, and "tone down" the framework's purported usefulness for flood risk (e.g. page 9 lines 17), since this remains unproven.*

**Response 4:**

Thank you.

We will tone done the discussion of the framework's usefulness for flood risk applications as this is not demonstrated in the manuscript. For example, on page 9 line 17 we will use an example not related to flood risk in explaining considerations relevant for choosing an appropriate hydrological model.

Additionally, reviewer 1 (see comment 3) has suggested that we examine additional streamflow characteristics, in particular flow duration curves. We plan to do this and include the flow duration curves, where they provide insight. This will provide a broader demonstration of the framework through an application that considers statistics at the finer (daily) scale.

**Comment 5:**

*It isn't clear how the boxplots (e.g. figure 3) are constructed. Is it the "13 errors" mentioned in page 8 line 24? Or is it somehow derived from the 10,000 synthetic rainfall years? Or the 73 years of observed data with synthetic rainfall "spliced in"? Either way, it isn't clear that the authors have avoided the proliferation of error metrics that they identify as a limitation of previous measures on page 3. If this method is applied to a large number of sites, it still seems like a not-entirely compact evaluation scheme. Perhaps the authors could clarify how this compares to other methods in this respect.*

**Response 5:**

Thank you for pointing this out. We will revise the example in Section 2.3.2 which explains how the unit test figure (those like Figure 3b) are constructed and used as a diagnostic. The Figure caption will also be improved. The revised text will clarify that each boxplot is a summary of the error (the difference between the simulated and virtual-observed performance statistic, Eq. 4) for the 10,000 replicates of the simulated 73 year time series. The "13 errors" relates to the number of boxplots displayed in figures of this type (the unit test for 12 influencing months and 1 integrated test). As you point out, this description of "13 errors" on page 8 line 24 is confusing. It will be revised to make clearer that there are 13 error summary boxplots, and also set out how the boxplots are constructed (a related response on Figure 3 is given to reviewer 1, comment 11).

The integrated test is designed to be a compact evaluation that includes multiple sites and statistics. Once the integrated test is completed and problems identified in the simulated streamflow, the more detailed unit test is applied to sites of interest (Step 6 of the framework). Figure 3 describes the unit tests, which are not designed to be undertaken on a large number of sites – they are designed to be more probing and are only undertaken on certain sites with problems, such as those identified by the integrated test.

The reviewer is right to point out that the example on page 3 does make it look like the proliferation of error metrics is identified as a limitation. We can see that the accompanying example emphasises the large number of statistics rather than our intended key point: that there are difficulties in assessing trade-offs or the relative importance of statistics. The example will be revised to clarify that the key issue for observed-rainfall evaluation is the difficulty in understanding the relative importance of rainfall features in terms of streamflow generation and what to do when performance is 'mixed'.

**Comment 6:**

*Relatedly, in Figure 6 and 7, shouldn't the "obs rain" and "virtual obs. Flow" be a range, rather than a single value? There are 73 years of monthly rainfall and simulated flows… this variability would be valuable context for evaluating the variability of the stochastic realizations.*

**Response 6:**

Yes, the variability of the rainfall and simulated flows is valuable context.

In Figures 6 and 7 we present a higher level summary to evaluate the model performance considering both mean conditions and their variability (i.e. standard deviations). The statistics presented in Figures 6 and 7 are the observed rain and the virtual-observed flow means (left column) and standard deviations (right column) over the full 73 years, calculated for the 12 months respectively. There are 12 monthly means and 12 monthly standard deviations per realisation. We are not calculating a separate statistic for each year of the timeseries. The boxplots show the range of these monthly statistics for the 10,000 stochastic rainfall model replicates. This convention is common to other papers in the field (e.g. Bennett et al. 2018, Khedhaouiria et al. 2018, Evin et al. 2018, Frost et al. 2011, Frost et al. 2004, Srikanthan et al. 2004, etc.)
* * *
**Comment 7:**

*It is unclear how other meteorological forcings (temperature, etc.) are handled in this framework. The authors focus on stochastic rainfall generators, as opposed to stochastic weather generators, meaning that the other forcings must be supplied independently of the rainfall. I would imagine that this could create some serious issues in some cases if synthetic rainfall is spliced together with inappropriate series of temperature or other forcings; one can imagine getting strange results in terms of precipitation vs. ET balances, with unclear consequences for the evaluation results.*
* * *
**Response 7:**

The reviewer is correct, as the focus is on evaluating stochastic rainfall generators, the other forcings are supplied independently. In our case study, the potential evapotranspiration (PET) time series (our only other meteorological forcing) is unchanged from the observed values in all hydrological simulations (i.e. the same PET time series is used in the simulation of the virtual-observed streamflow, integrated tests and unit tests). This is important as the hydrological evaluation is a relative comparison of the observed and simulated rainfall, hence all other time series and parameters relating to the hydrological model are kept the same in all instances. This approach was also taken in Sikorska et al. (2018), where the impact of using different rainfall disaggregation schemes on resultant flow was tested using a hydrological model. For all these tests the historical observed temperature time series was used to enable a comparison between the rainfall elements only.

To assess this assumption for the Onkaparinga case study we have evaluated the rainfall-PET correlation in all months. There is a negative relationship, which accounts for a small portion of the variance, up to $R^2 = 0.11$ in drier summer months. Figure 1 shows the rainfall-PET correlations for a drier summer month (January) and a wet winter month (June).

[Figure]

**Figure 1: Rainfall-PET correlation (left) January and (right) June.**

While there is some non-zero relationship, we do not consider it to undermine the case study (since all other statistics of PET are reproduced and the relationship is mild). However, this may not be the case for other locations where the model is applied. We will therefore discuss the matter in the paper, and suggest a method to identify whether it is significant. The method is to perform a bootstrap of the observed rainfall, for example, shuffle the order of years of rainfall but keep the same sequence of PET–thus breaking the rain-PET correlation–and test for differences in terms of virtual-observed streamflow compared to the observed time series of rainfall and PET.

We will include the reviewer's recommendation to apply the framework more generally to stochastic weather generators. The application would require care to ensure that the PET generator does not introduce other deficiencies.

**Comment 8:**

*I have never developed my own SRG, but I imagine that it might be hard to know exactly how to use the results from this analysis to refine that generator, despite the authors' claim that this is a valuable use of the framework. It identifies performance by month, rather than by "rainfall characteristics" (pg. 21 line 9). It is useful to know whether the SRG performs well for some months than others, but what next? If the authors plan to continue research on this topic, I would suggest that a method that "tracks" the propagation of rainfall through the model might be more effective. To me, the most clear way of doing this is to track how different rainfall statistical moments translate to different statistical moments in the streamflow, using both historical and synthetic rainfall. Such an approach would be amenable to changing the evaluation timescale. For these reasons, I recommend that the authors delete the statement that this framework "should be an essential step in the development and application of stochastic rainfall models" (page 21 line 22-23). On a related note, the authors should comment on how this technique would apply to distributed (i.e. high-resolution gridded) SRGs and hydrologic models.*

**Response 8:**

The reviewer has raised some excellent discussion points. We provide specific responses below to the discussion points raised.

1. "*It is useful to know whether the SRG performs well for some months than others, but what next?*" – Thank you. We agree that it is a useful feature and it provides much more information than observed-rainfall evaluation alone. This is a key innovation of the paper. Following an observed-rainfall evaluation the focus would have been on the months Jan, Feb, Nov, Dec, May and June (Bennett et al., 2018). However, based on the results of the virtual hydrologic framework, we now know that May-July are the key months when considering the hydrology and that the problems with modelled rainfall in Jan-Feb, Nov-Dec are less important. Also, we now know that rainfall in preceding months is important and not just the month in which the flow is evaluated, which is more information than before.

   We agree that it does not tell us exactly which rainfall characteristics to focus on. However, it is unlikely to be that simple – a single rainfall statistic is unlikely to translate into a single runoff statistic because streamflow integrates a range of rainfall processes (see also reviewer #1, comment 21).

   Now that we know in which months deficiencies originate, we can focus on those months and trial various alternatives to the rainfall model to address the problem. This is left for future research, as mentioned in Section 5.3.

2. "*I would suggest that a method that "tracks" the propagation of rainfall through the model might be more effective … most clear way of doing this is to track how different rainfall statistical moments translate to different statistical moments in the streamflow, using both historical and synthetic rainfall*" – This idea offers scope to extend the framework, and is something to consider in the future. In this paper, we describe the integrated test and then introduce the unit test in terms of 'splicing' monthly blocks of rainfall. However, the reviewer is right that the approach could be formulated differently to use different 'splicing' approaches. For example, to examine the percentage changes in resultant streamflow as a function of a particular change in the inputted rainfall. We will clarify that the framework in its currently presented form is not the only way to undertake this type of investigation.

3. "*I recommend that the authors delete the statement that this framework "should be an essential step in the development and application of stochastic rainfall models"*" – We will soften the wording of this statement.

4. "*On a related note, the authors should comment on how this technique would apply to distributed (i.e. high-resolution gridded) SRGs and hydrologic models.*" – This is an important topic. We will comment on how this technique could apply to distributed rainfall and hydrologic models in Sections 5.2 and 5.3.

**Comment 9:**

*I wonder if this framework should consider the autocorrelation in monthly rainfalls when doing this splicing. I don't know too much about the climate of South Australia, but I can imagine that autocorrelation at least in dry periods can be quite important, and this is likely not preserved during the splicing. It's not clear what the implications would be for the resulting evaluation.*

**Response 9:**

This is a valid and interesting point. Bennett et al. (2018) demonstrated that the monthly autocorrelations are small for the Onkaparinga catchment (from -0.2 in drier summer months to 0.3 in the wetter winter months), and as a result this issue was not considered in the presentation of the framework.[2] We can appreciate that monthly/seasonal autocorrelation is a significant feature of other locations and that could be a limitation when applying this method. We briefly suggested in Section 5.3 that the issue of monthly autocorrelation could be explored as an extension of the model – we will provide further details to explain how this might be achieved. For example, rather than naïvely splicing rainfall it might be possible to conditionally sample the simulated rainfall in a manner that preserves monthly correlations. The efficacy of this technique would require some exploration since there may be limitations arising from the conditional sampling.

**Comment 10:**

*Figure 3 and elsewhere: I don't understand what "(90% limits shown)" means.*

**Response 10:**

The reference to 90% limits indicates that the boxplot whiskers extend to from the 5th to 95th percentile values of the metrics based on the 10,000 replicates. The initial description of these figures (Section 2.3.2) and all figure captions will be revised to indicate that the "(90% limits shown)" indicates that the boxplot whiskers extend to the 90% limits of the 10,000 simulations for the presented statistic. Please also see the response to comment 5.
* * *
[2] Bennett at al. (2018) also demonstrated that the model sufficiently reproduced these small monthly autocorrelations.

**Comment 11:**

*Section 3: Mention basin size. Also, why are stations outside the watershed used?*

**Response 11:**

Thank you, the basin size will be included in the revised manuscript (323 km$^2$).

All the sites identified in Figure 2 were used to estimate the catchment average rainfall (in the revised Section 3 of the manuscript we will better explain this) for the rainfall-runoff modelling calibration. When estimating catchment average rainfall it is fairly common to use sites outside the catchment, to better represent the spatial variability and to avoid boundary effects. It is therefore important that a stochastic rainfall model is able to reproduce the rainfall statistics at all of the sites outlined in Figure 2. This is why we evaluated the stochastic rainfall model at all the sites indicated in Figure 2.

There are further reasons why this is valid for a virtual approach. Most notably, because there is no comparison made with observed streamflow. The virtual hydrological evaluation uses the calibrated hydrological model as a tool to process the observed and simulated rainfall for comparison. The virtual-observed streamflow can be thought of as a virtual stream flow gauge. The virtual stream gauges have no physical location that they are trying to replicate. Instead the virtual stream gauges enable a synthetic test of the simulated rainfall.

Virtual hydrological evaluation of a single rainfall site is analogous to treating the information at the selected rainfall gauge (observed and simulated) as being representative of the catchment rainfall. This 'catchment rainfall' is then routed through the chosen hydrological model to produce simulated and virtual-observed streamflow at the 'virtual catchment outlet'.

This type of virtual approach was used in a different context (the development of new techniques for flood frequency estimation) in which a calibrated hydrological model was 'moved' all over Australia (Li et al., 2016).

**Comment 12:**

*More importantly, is the rainfall hydroclimate stationary? If not, then it seems as though this whole issue of stochastic generation and comparison of resulting streamflows against a nonstationary continuous simulation would be more complicated. Please comment on this.*

**Response 12:**

Thank you, we will comment on this issue in Section 5.3.

The reviewer is right to point out the complicated nature of comparing stochastically generated rainfall against a nonstationary continuous simulation. In this paper we took steps to minimise this impact by careful selection of the observed rainfall period.[3] Evaluating non-stationarity is considered an extension to this framework and therefore left for future research. It is conceivable that the same general framework can be applied for a selected non-stationary period, but care would be needed in the selection of statistics to identify model performance (since the performance in different sub-periods could be masked when evaluating an overall period). A related issue is that the hydrological model should provide adequate performance across the range of non-stationary climate forcings to which it is subjected.
* * *
[3] The catchment experiences a significant rainfall decline in the early 2000's (see Westra et al 2014a and 2014b) due to the 'millennium drought'. This is why we choose an earlier rainfall period that finishes in 1986. Although this does not mean we have eliminated the impact of non-stationarity it has been reduced by taking this step.

**Minor Comments**

> **Comment 13:**
>
> *Page 1 line 15: change "months" to "seasons"- that is a more broadly relevant term. Hydrology varies seasonally, months are an arbitrary construct (this comment applies elsewhere in the paper, such as page 2 line 31)*

**Response 13:**

Thank you for pointing this out. We will change 'months' to 'seasons'.

> **Comment 14:**
>
> *Throughout paper: I recommend introducing an acronym for stochastic rainfall models and using it throughout.*

**Response 14:**

Thank you, we will introduce and acronym for stochastic rainfall models and use it throughout the manuscript.

> **Comment 15:** *Page 1 line 9: change "is" to "has been"*
>
> **Comment 16:** *Page 1 line 10: change "is given" to "has been paid"*
>
> **Comment 17:** *Page 1 line 12: delete "whenever the simulated rainfall are poor"*
>
> **Comment 18:** *Page 1 line 19: change "catchment cycle" to "annual hydrologic cycle"*
>
> **Comment 19:** *Page 1 line 28: delete comma after "targeted"*
>
> **Comment 20:** *Page 2 line 10: "and/or" is not appropriate in technical writing. Use "or"*
>
> **Comment 21:** *Page 2 line 12: put "virtual experiments" in quotations when mentioned for the first time, for emphasis*
>
> **Comment 22:** *Page 1 line 24-25: I recommend deleting "risks" after "floods" and "droughts" and changing it to "hazards" after "hydrologic"*
>
> **Comment 23:** *Page 3 line 16: add comma after "poor"*
>
> **Comment 24:** *Page 5 line 10: The goal is not to match streamflow observations. It is to match the statistics of streamflows*
>
> **Comment 25:** *Page 3 line 12: add "model" before parameters*
>
> **Comment 26:** *Page 3 line 13: Why would you call ET "extraneous"? It is generally very very important.*
>
> **Comment 27:** *Page 10 line 15: I think that "observed/virtual" is a strange term. Observations have very little usage in this study…*
>
> **Comment 28:** *Page 11 line 16: grammar problem "was fit good"*
>
> **Comment 29:** *Page 19 line 3-4: streamflow arises from more than just rainfall integration over a catchment area-what about ET, etc.?*
>
> **Comment 30:** *Page 20 line 13: good place to mention that multiple SRGs could be used too, not just multiple hydrologic models.*
>
> **Comment 31:** *Page 20 line 18: put "memory" in quotations.*
>
> **Comment 32:** *Page 14 line 14: This sentence is a bit awkward. It isn't perhaps so "common and obvious" to the reader.*
>
> **Comment 33:** *Page 20 line 11-13: I don't understand this sentence. Certainly model performance depends on the chosen model.*

**Response 15-33:**

We thank the reviewer for pointing out the above editorial corrections (comments 15 - 33) and for their thorough consideration. We will incorporate these corrections in the revised manuscript.

**References**

Bennett, B., Thyer, M., Leonard, M., Lambert, M., and Bates, B. (2018). A comprehensive and systematic evaluation framework for a parsimonious daily rainfall field model, Journal of Hydrology, 556, 1123-1138.

Evin, G., Favre, A.-C., and Hingray, B. (2018). Stochastic generation of multi-site daily precipitation focusing on extreme events, Hydrology and Earth System Sciences, 22, 655-672, 2018.

Frost, A.J., Cowpertwait, P. and Srikanthan, R. (2004). Stochastic generation of point rainfall data at subdaily timescales: a comparison of DRIP and NSRP. CRC for Catchment Hydrology.

Frost, A.J., Charles, S.P., Timbal, B., Chiew, F.H., Mehrotra, R., Nguyen, K.C., Chandler, R.E., McGregor, J.L., Fu, G., Kirono, D.G. and Fernandez, E. (2011). A comparison of multi-site daily rainfall downscaling techniques under Australian conditions, Journal of Hydrology, 408(1-2), pp.1-18.

Khedhaouiria, D., Mailhot, A. and Favre, A.C. (2018). Daily Precipitation Fields Modeling across the Great Lakes Region (Canada) by Using the CFSR Reanalysis, Journal of Applied Meteorology and Climatology, 57(10), pp.2419-2438.

Li, J., Thyer, M., Lambert, M., Kuzera, G., Metcalfe, A. (2016). Incorporating seasonality into event-based joint probability methods for predicting flood frequency: A hybrid causative event approach, Journal of Hydrology, 533, 40-52.

Sikorska, A. E., Vivrioli, D., Seibert, J. (2018). Effective precipitation duration for runoff peaks based on catchment modelling, Journal of Hydrology, 556, 510–522

Srikanthan, R., Chiew, F. and Frost, A. (2004). Stochastic Climate Library, User Guide.

Westra, S., Thyer, M. Leonard, M., Kavetski, D. and Lambert, M. (2014a). A strategy for diagnosing and interpreting hydrological model nonstationarity, Water Resour. Res., 50, 5090–5113, doi: 10.1002/2013WR014719.

Westra, S., Thyer, M., Leonard, M., Kavetski, D., and Lambert, M. (2014b). Impacts of climate change on surface water in the Onkaparinga catchment-Final report volume 1: hydrological model development and sources of uncertainty, 1839-2725.

---

## Author Comment (AC3) · 19 Nov 2018

Our response to reviewer 3 is available as a pdf supplement via the link below.

Please also note the supplement to this comment:
https://www.hydrol-earth-syst-sci-discuss.net/hess-2018-489/hess-2018-489-AC3-supplement.pdf
* * *

---

## Author Response (AR1)

**Overall Response to Editor**

This document describes our response to the reviewers' comments and our revision of the manuscript entitled 'A virtual hydrological framework for evaluation of stochastic rainfall models' (HESS 2018-489). We agreed with the majority of the issues raised by the reviewers and editor. The revised paper is a significant improvement on the original submission. The key modifications we have made in the revised paper include:

- Clarifying and emphasising the novelty of the virtual hydrological evaluation framework more strongly (see Section 1: Introduction). In particular, we have provided better referencing and discussion on existing uses of virtual-observed streamflow evaluation and have emphasised the significance of formalising virtual-observed streamflow evaluation using a comprehensive and systematic evaluation (CASE) framework approach (originally developed in Bennett et al. 2018), as well as the novelty of the new unit test.
- Restructuring and shortening the motivation for the virtual hydrological evaluation approach as well as descriptions of the theoretical elements (see Sections 1 and 2).
- Extending the presented analysis to include additional runoff characteristics (i.e. flow duration curves) in the diagnosis of the sources of deficiencies in streamflow (see Section 4.3). This evaluation of flow duration curves has been incorporated in to the virtual hydrological evaluation framework as part of Step 3 (see Section 2.4.3.).
- Streamlining the virtual evaluation framework steps (from seven steps down to three), summarising this procedure graphically using a flow chart (see Figure 2) and restructuring Section 2 to better integrate the framework steps and the specific virtual tests (i.e. integrated and unit tests). Section 4 (Results) has also been re-structured to follow the streamlined framework procedure.
- Improving the explanations of the stochastic rainfall model and the hydrological model used in the case study, including the providing descriptions of the calibration approaches (see Section 3).

Please find below our response to specific comments.

**Response to Editor**

> *Thank you for posting your responses to the three referees' reports. The referees raised some critical comments and suggestions that I urge you to consider. I feel your work, if thoroughly revised, could ultimately be accepted for publication in HESS. I invite you to upload a revised manuscript, incorporating the proposed changes and additions, and making any other modifications where you see fit. Alongside a thoroughly revised version of your manuscript, please provide a detailed response ("item-by-item") to each of the referees' remarks.*
>
> *In addition to the comments from the referees, I kindly ask you to add "Author contribution" and "Data availability" sections to the manuscript, as indicated in the guidance for authors (see https://www.hydrology-and-earth-system-sciences.net/for_authors/manuscript_preparation.html).*

**Response:**

We have thoroughly revised the manuscript in light of the reviewer comments. 'Author contribution' and 'Data availability' sections have been added to the revised manuscript.

Our item-by-item responses appear below.

**Response to Reviewer 1**

**Comment 1:**

*The manuscript introduces a new framework for the evaluation of generated rainfall time series in terms of their ability to reproduce runoff time series characteristics. This is done by two tests, an integrated test and a unit test. This topic is of broad interest for the hydrological scientific community and suitable for a publication in HESS.*

*However, I consider the integrated test not as a novelty, since it has been applied before in different studies, but the unit test is useful for rainfall model evaluation. Hence, I suggest to move the focus to the unit test and extend the validation by other runoff characteristics. Also, the theoretical elements of the paper are very long, the application and validation of the test should be extended and there is a lack of some crucial information regarding the applied r-r model and its calibration procedure (for details please see my specific comments). Due to the resulting workload I suggest a major revision of the manuscript.*

**Response 1:** We are pleased that the reviewer found our manuscript suitable for publication and we are grateful for the insightful and constructive comments. They have been very helpful. The suggestions provided show careful consideration and have led to an improved revision of the manuscript. Regarding the specific matters raised:

- *Novelty of the integrated test* –  we have reworded objective 2[1] and provided better referencing and discussion on existing uses of virtual-observed streamflow evaluation (calibration, validation, model selection and diagnosis) in the Introduction. The revised text is:

  Page 4, Line 21- 25: *"To date, 'virtual experiments' have been used in a variety of contexts, including (i) the evaluation of hydrological model sensitivity (Ball, 1994, Nicótina et al., 2008, Paschalis et al., 2013, Shah et al., 1996, Wilson et al., 1979) including the identification of rainfall features of interest in terms of hydrological behaviour (Sikorska et al., 2018), (ii) for developing new techniques for flood frequency analysis (Li et al., 2014, 2016), and (iii) to support SRM selection (Müller and Haberlandt, 2018) as well as calibration and validation (Kim and Olivera, 2011) through a comparison to virtual streamflow."*

  We feel that there are some important aspects of the implementation of our framework that distinguish it from existing presentations of virtual-observed streamflow evaluations. In particular, the presentation of the integrated test in this paper is the first time, a virtual-observed streamflow evaluation has been formalised used using a comprehensive and systematic evaluation (CASE) framework approach (Bennett et al. 2018). This has been clarified in the text:

  Page 4, Line 26 – Page 5, Line 3: *"The framework presented in this paper is significant advance from previously reported virtual experiments because it presents a formal framework to identify key deficiencies in the SRM by utilising (1) A comprehensive and systematic evaluation (CASE) framework (developed by Bennett et al., 2018 and used by Evin et al., 2018, Khedhaouiria et al., 2018) that systematically categorises performance at multiple spatial and temporal scales using quantitative criteria for each statistic, and (2) two types of virtual experiments that are able to identify the source of key deficiencies in SRM at specific locations and time periods."*
* * *
[1] Page 5, Line 6, *Objective 2: "*To present two different tests which are part of the framework: the integrated test as well as introduce a new test, the unit test. Combined use of these tests allows streamflow discrepancies to be attributed to their original source in the SRM according to site and season."

This distinguishing feature and others are further discussed in response to comment 2. We have taken the reviewer's advice and emphasized the novelty of the unit test and its diagnostic ability in the revised manuscript.

- *Evaluation using other runoff characteristics* – we have included an examination of flow duration curves to provide additional insight into the deficiencies of the rainfall model. This evaluation has been incorporated in to the virtual hydrological evaluation framework as part of Step 3 (see Section 2.4.3):

    Page 12, Lines 1-8: *"Following a side-by-side comparison of integrated test and unit test results in terms of the relative errors the sources of poor performance should be classified in terms of in which influencing months streamflow deficiencies originate (e.g. poor streamflow arises from rainfall deficiencies mostly in the same month, a prior month or a contiguous block of months) … To complement this analysis a comparison of the virtual-observed flow duration curve for the evaluated time period with the flow duration curves resulting from unit tests for key influencing months is also recommended."*

- *Length of theoretical elements* – the motivation and description of the theoretical elements (now Sections 1 – 2.1) have been shortened by 40 lines  (from 167 lines down to 127 lines).
- *Information on the r-r model* – we have improved the explanation of the rainfall-runoff model, including adding references of the calibration and validation procedure (see Section 3). The revised text is:

    Page 13, Lines 1-12: "*The hydrological model GR4J (Perrin et al., 2003) was used to simulate virtual-observed streamflow at a daily time step. GR4J is a daily lumped hydrological model that simulates daily streamflow in a parsimonious manner using four parameters. The GR4J model was calibrated according to the procedure set out in Westra et al. (2014b) for the stationary version of the GR4J hydrological model. The details are provided in (Westra et al., 2014a) and a short summary is provided here. The multi-site rainfall gauges were Thiessen weighted to calculate the catchment average rainfall. The model was calibrated to the streamflow data at Houlgrave Weir (see Figure 4) using model calibration period of 15 years (1985-1999). The parameters were estimated using maximum likelihood estimation procedure with a weighted least squares likelihood function. The model parameters that maximised the likelihood function were found using a multi-start quasi-Newton optimisation procedure with 100 random starts. Overall, the GR4J model was a good fit to the observed streamflow, with a Nash-Sutcliffe efficiency of 0.8. A similar type of hydrological model and calibration approach has been used for other virtual evaluation studies (Li et al. 2014; 2016). The same set of hydrological model parameters are used for both the unit and integrated tests so that the same transformation of rainfall to flow is used.*"

We further elaborate on these items in response to subsequent comments made.
* * *
**Comment 2:**

*P2l20-23: The so-called "virtual-observed streamflow"-approach and the integrated test is not new and a widely used evaluation method, especially in data-sparse regions or research fields. For example in urban hydrology, where measured runoff characteristics are not often available, the simulation of a reference streamflow is very common (e.g. Müller and Haberlandt, 2018). The authors even mention other studies using the integrated test (Li et al., 2014, 2016). However, the unit test is interesting and indeed provides useful insights into the rainfall-runoff (r-r) transformation process.*
* * *
**Response 2:**

Thank you for supplying the references: Müller and Haberlandt, 2018; Sikorska et al. 2018. We have included them in the revised introduction text along with additional references (e.g. Kim and Olivera, 2012).

We agree that the concept of a virtual-observed streamflow evaluation is not new and we have revised the introduction, including the presentation of objective 2, to make this clear and discuss that the approach has been used in a variety of contexts (e.g. calibration, validation, model selection and as a diagnostic tool). The revised text is:

> Page 4, Line 21- 25: "*To date, 'virtual experiments' have been used in a variety of contexts, including (i) the evaluation of hydrological model sensitivity (Ball, 1994, Nicótina et al., 2008, Paschalis et al., 2013, Shah et al., 1996, Wilson et al., 1979) including the identification of rainfall features of interest in terms of hydrological behaviour (Sikorska et al., 2018), (ii) for developing new techniques for flood frequency analysis (Li et al., 2014, 2016), and (iii) to support SRM selection (Müller and Haberlandt, 2018) as well as calibration and validation (Kim and Olivera, 2011) through a comparison to virtual streamflow.*"

However, there are some important aspects of our framework that distinguish it from existing presentations of virtual-observed streamflow evaluation, as outlined below.

1.  This is the first time the virtual-observed streamflow evaluation approach has been formalised using a Comprehensive and Systematic Evaluation (CASE) framework (pioneered by Bennett et al., 2018 and used by Evin et al. 2018, Khedhaouiria et al. 2018) to evaluate stochastic rainfall models in terms of the ability to produce key runoff statistics of interest. The integrated tests presented in this paper follow the CASE approach because they (i) present a comprehensive range of key statistics of interest, (ii) systematically categorise performance at specific spatial and temporal scales using quantitative criteria for each statistic, and (iii) systematically categorise aggregate performance over multiple spatial and/or temporal scales. This novelty has been clarified in the revised manuscript:

    > Page 4, Line 26 – Page 5, Line 3: "*The framework presented in this paper is significant advance from previously reported virtual experiments because it presents a formal framework to identify key deficiencies in the SRM by utilising (1) A comprehensive and systematic evaluation (CASE) framework (developed by Bennett et al., 2018 and used by Evin et al., 2018, Khedhaouiria et al., 2018) that systematically categorises performance at multiple spatial and temporal scales using quantitative criteria for each statistic, and (2) two types of virtual experiments that are able to identify the source of key deficiencies in SRM at specific locations and time periods.*"

    In contrast, previous papers (Müller and Haberlandt, 2018; Sikorska et al. 2018, Kim and Olivera, 2012) have used a virtual-observed streamflow evaluation approach, but have not used a CASE framework to evaluate the performance of stochastic rainfall model at multiple rainfall sites in terms of its ability capture key streamflow statistics of interest. For example, Müller and Haberlandt (2018) established the need for spatial consistency of rainfall generation in modelling sewer networks by comparing rainfall disaggregation approaches with or without spatial consistency. This virtual-observed streamflow evaluation is performed for identified extreme rainfall events only and therefore does not use a CASE approach that considers multiple temporal scales and the longer term effects of the applied rainfall on the translation of subsequent rainfall to streamflow. Sikorska et al. (2018) focused on identifying rainfall features of interest in terms of resultant hydrological behaviour for the purposes of determining the effective daily precipitation duration with a view to selecting a suitable rainfall disaggregation scheme. Their evaluations determined that detailed temporal rainfall time series were not needed to reproduce annual or seasonal peaks in their modelled catchments. Although the evaluations presented are comprehensive the motivation of the Sikorska et al. (2018) paper is different and does not provide a general formalised framework for systematically categorising stochastic rainfall model performance at specific and aggregate temporal and spatial scales. Kim & Olivera (2012) used virtual-observed streamflow evaluation as part of a larger calibration and validation approach in which various weights were trialled to reflect the importance of various rainfall statistics within a modified Bartlett-Lewis rectangular pulse (MBLRP) model. However, the focus was on the improvement and validation of the MBLRP model rather than the presentation of separate framework for model evaluation. Finally, Li et al. (2014, 2016) used a virtual-observed streamflow evaluation approach, to evaluate the ability of range of techniques to estimate the derived annual flood frequency distribution - they did not use a CASE approach to evaluate stochastic rainfall models.

2.  The formalisation of the integrated test using a comprehensive and systematic evaluation (CASE) approach, forms a baseline for subsequent application of the unit test which has greater ability to pinpoint issues with respect to the source of the rainfall error on a monthly basis. For example, in Step 3 of the framework the following comparison is made:

> Page 11, Lines 4-6: "*Side-by-side comparison of the results of the integrated test and unit tests are given in terms of the errors for selected monthly and annual statistics (see illustration in Fig. 3(b)). The comparison of errors from the unit test forms the basis of interpretation for hydrological insights and their relationship to the rainfall model.*"

3. As identified by the reviewer, we introduce an innovative unit test, which has never been used before in a virtual-observed streamflow evaluation approach. The key advantage of this unit test is that by splicing together the observed and simulated rainfall in a systematic manner, it is able to develop new insights on which months have deficiencies in simulated rainfall that produce poor performance in streamflow. We have put greater emphasis on this new innovative unit test in the revised manuscript including extending the unit test to evaluate streamflow deficiencies via an examination of daily flow duration curves.
* * *
**Comment 3:**

*It would be useful to move the focus on this test and proof it with additional runoff characteristics, e.g. flow duration curves, not using only the monthly runoff amount. Therefore, no new simulations are necessary, only additional analyses of the existing r-r simulation results.*
* * *
**Response 3:**

We have included an examination of flow duration curves to provide additional insight into the deficiencies of the rainfall model in the revised manuscript (see Section 4.3). An evaluation of flow duration curves has been incorporated in to the virtual hydrological evaluation framework as part of Step 3 (see Section 2.4.3) to augment the unit test evaluations. An example of the examination and discussion of the flow duration curves and a flow duration curve figure (Fig. 9) are reproduced below:

> Page 18, Lines 9-15: "*This need to improve September in preference to preceding months is also illustrated via Fig. 9 (c) where the September daily flow duration curves are shown for the cases where August (orange shading) and September (blue shading) are the influencing months compared against the virtual-observed September flow duration curve (purple dots). Where August is selected as the influencing month, the virtual-observed flow duration curve largely sits inside the 90% limits of the flow duration curves resulting from the unit testing procedure. Whereas, the virtual-observed flow duration curve sits outside the 90% limits of the unit test flow duration curve when September is taken as the influencing month. Thereby providing further evidence that to improve September flows that September rainfall should be improved in preference to other months.*"

[Figure]

**Fig. 1** Lobethal, Site 12 (90% limits shown) (a) unit test error in mean monthly flow (September), (b) unit test error in standard deviation of monthly flow (September), (c) Unit test September flow duration curve when August and September are selected as influencing months (top 10% of flow days shown).

**Comment 4:**

*P2l23-25 The sentence is not clear without the explanations given in section 2. Either here more information are provided or the sentence is left out.*

**Response 4:**

Thank you. The sentence has been left out in the revised manuscript.

**Comment 5:**

*P3l9-14 The idea behind the example provided by the authors is clear. Nevertheless, some of the rainfall characteristics mentioned are not clear and, since it is only an example, can be left out or can be replaced by other rainfall characteristics:*
*- rainfall on wet days - What does this characteristic represent (the daily total rainfall itself is mentioned later)?;*
*- Extreme value analysis on a monthly basis and autocorrelation on an annual basis are from my understanding rather uncommon rainfall characteristics for the evaluation of rainfall time series*

**Response 5:**

Due to a restructure of the paper the example does not appear in the revised manuscript.

**Comment 6: P3l20-21 The details provided in brackets can be left out, since without reading the reference there are no additional information for the reader.**

**Response 6:** Due to the shortening of the introduction and motivation of the new evaluation framework (see comment 7 below) the sentence no longer appears in the revised manuscript.

**Comment 7:**

*P3l7-P4l20 The motivation for the introduction of the new evaluation strategy is quite long and can be shortened by the half. I think the majority of the community is quite aware of the issue with overlapping errors. Also Fig. 1 and Fig. 2 are quite clear from the text and could be left out. If kept, a box with "True rainfall" should be added in Fig.1a) to be consistent with Fig. 1b ("True streamflow")*

**Response 7:**

The motivation for the new evaluation strategy has been restructured and shortened by 50 lines (from 167 lines to 127 lines) while maintaining the key points of the introduction and section 2.1 of the original manuscript. We have retained Fig. 1 and 2 but merged them (now Fig. 1 in the revised manuscript) and made the amendments suggested above. Based on our experiences explaining this work these figures are helpful to avoid misconceptions. For convenience the merged figure is reproduced below.

[Figure]

**Fig. 2 Schematic of (a) observed-rainfall evaluation where simulated rainfall is compared against observed rainfall and (b) observed-streamflow evaluation where simulated streamflow is compared against observed streamflow (c) virtual hydrological evaluation framework where simulated streamflow is compared against virtual-observed streamflow.**

**Comment 8:**

**P5l10 "to match streamflow observations" -> "to match streamflow observations or statistics"**

**Response 8:** The sentence has been modified as suggested.
* * *
**Comment 9:**

*P6Table1 The authors should include a definition of the applied symbols in the caption, since the difference between "x" and "-" is not too intuitive (from only the table). Is in the last line, first column something missing (virtual hydrological…)?*
* * *
**Response 9:**

To avoid ambiguity Table 1 has been revised so that text (i.e. 'Yes' and 'No') is used in place of the original symbols and an additional column is now included that indicates whether the analysis is streamflow-based. The last line, first column has been amended to read 'virtual hydrological evaluation'. Table 1 is reproduced below for convenience.

**Table 1 Comparison of the sources of error for observed-rainfall, observed-streamflow and virtual hydrological evaluation frameworks as well as whether the evaluation is streamflow-based.**

| | Source of error | | | Streamflow-based evaluation |
|---|---|---|---|---|
| | Stochastic rainfall model | Hydrological model | Observed streamflow | |
| **Observed-rainfall evaluation** | Yes | No | No | No |
| **Observed-streamflow evaluation** | Yes | Yes | Yes | Yes |
| **Virtual hydrological evaluation** | Yes | No | No | Yes |
* * *
**Comment 10:**

*P6Table 1 From my opinion the results from the virtual-observed streamflow approach can still be biased by the applied r-r model. For example, rainfall is generated in space and two rainfall generation methods show differences in terms of rainfall characteristics, but not in the simulated streamflow. After what I've read in the introduction and methods section, the conclusion is that the compared rainfall characteristics are then not practicable ("no impact") und useless (for the study region). But this also depends on i) the model choice (including e.g. spatial resolution, model type (fully / semi-distributed), several model approaches) and ii) the parameter identification. In a semi-distributed model differences in spatial rainfall could be dampened, while they are (maybe) not dampened in a fully-distributed model. The parameters have to be chosen a priori – a calibration on one of the rainfall data sets is not possible to avoid biases. Will the parameters be calibrated by an additional rainfall data set (the observed data) and if so, how can be avoided that this calibration introduces a bias (e.g. maybe the observed rainfall data is more similar to rainfall data set A under investigation than to B)? So all of the results depend on the chosen setup for the r-r simulations and drawn conclusions are only valid in context with the model setup and parameter set. This is of course always the case in hydrology, but it becomes more important if a virtual runoff time series is applied, since the "relation" between the model output and reality gets lost. However, the authors point these issues out later in their investigation (p20), but it should be communicated earlier to the reader.*
* * *
**Response 10:**

The reviewer has raised some excellent discussion points. Although we have discussed some of these points later in the investigation (Section 5), we now raise these key issues earlier with the reader (Section 2.1).

We provide specific responses below to the discussion points raised.

1. "*Virtual-observed streamflow approach can still be biased by the applied r-r model*" – Yes, we agree. This is a very important matter to consider. As a result we acknowledge that virtual hydrological evaluation should use an appropriate hydrological model in Section 2 :

    Page 5, Lines 29-30: "*It is also important that the selected hydrological model is fit for purpose so that it can simulate the streamflow characteristics of interest.*"

    The need for an appropriate hydrological model is further discussed later in Section 5.3:

    Page 27, Lines 25-30: "*Hydrological structural errors may potentially skew interpretation of the SRM evaluation if the hydrological model poorly represents the catchment processes. To reduce these impacts the steps taken in this study included (a) using a well-tested hydrological model that has demonstrated good performance on a wide range of catchments (e.g. the GR4J model has been widely tested , see (Perrin et al., 2003, Coron et al., 2012); (b) calibrating and*

*evaluating the hydrological on a catchment close to the observed rainfall sites to ensure it provided sufficiently good performance  (e.g. GR4J was calibrated to the Onkaparinga catchment - see (Westra et al., 2014a, Westra et al., 2014b)."*

Future research on the application of the VHE will consider multiple hydrological models to avoid the reliance on single rainfall-runoff models and/or single calibration schemes. This discussion has been added to Section 5.3:

> Page 27. Lines 30-32: "*Future research will use multiple, well-tested hydrological models with sufficiently good performance to reduce the reliance on a single hydrological model and ensure the identification of SRM deficiencies is not dependent on a single hydrological model.*"

2. "*Differences in terms of rainfall characteristics, but not in the simulated streamflow*" – Yes, there is the potential that a chosen rainfall-runoff model is insensitive to certain important differences in modelled rainfall. While we agree with the reviewer that spatial patterns are important, temporal differences in rainfall patterns, which are capture by a lumped rainfall-runoff model are potentially equally as important.

3. "*But this also depends on i) the model choice … and ii) the parameter identification*" – Yes, as with the comments in (1), all elements of the modelling method can potentially introduce bias. The end-user's impact of interest and associated modelling process can influence an outcome. These observations reinforce the need for care when applying the framework. This includes ensuring that (i) the rainfall-runoff model is 'fit for purpose',

> Page 5, Lines 29-30: "*It is also important that the selected hydrological model is fit for purpose so that it can simulate the streamflow characteristics of interest.*"

(ii) that this rainfall model is well-tested,

> Page 27, Lines 25-30: "*Hydrological structural errors may potentially skew interpretation of the SRM evaluation if the hydrological model poorly represents the catchment processes. To reduce these impacts the steps taken in this study included (a) using a well-tested hydrological model that has demonstrated good performance on a wide range of catchments (e.g. the GR4J model has been widely tested , see (Perrin et al., 2003, Coron et al., 2012)*"

and (iii) well calibrated.

> Page 27, Lines 25-30: "*(b) calibrating and evaluating the hydrological on a catchment close to the observed rainfall sites to ensure it provided sufficiently good performance  (e.g. GR4J was calibrated to the Onkaparinga catchment - see (Westra et al., 2014a, Westra et al., 2014b).*"

A description of the hydrological model calibration  approach is reproduced below:

> Page 14, Lines 1-17: "*The hydrological model GR4J (Perrin et al., 2003) was used to simulate virtual-observed streamflow at a daily time step. GR4J is a daily lumped hydrological model that simulates daily streamflow in a parsimonious manner using four parameters. The GR4J model was calibrated according to the procedure set out in Westra et al. (2014b) for the stationary version of the GR4J hydrological model. The details are provided in (Westra et al., 2014a) and a short summary is provided here. The multi-site rainfall gauges were Thiessen weighted to calculate the catchment average rainfall. The model was calibrated to the streamflow data at Houlgrave Weir (see Figure 4) using model calibration period of 15 years (1985-1999). The parameters were estimated using maximum likelihood estimation procedure with a weighted least squares likelihood function. The model parameters that maximised the likelihood function were found using a multi-start quasi-Newton optimisation procedure with 100 random starts. Overall, the GR4J model was a good fit to the observed streamflow, with a Nash-Sutcliffe efficiency of 0.8. A similar type of hydrological model and calibration approach has been used for other virtual evaluation studies (Li et al. 2014; 2016).*"

4. "*If so, how can be avoided that this calibration introduces a bias*" – This is an excellent question, which we will address with further discussion. While best-practice models and methods are important, this does not necessarily guard against the possibility that a model poorly represents key processes of

interest. A remedy for this limitation would be to use multiple rainfall-runoff models and this is now clearly mentioned as the first step of Future research in new Section 5.3, Limitations and future research.

> Page 27, Line 22-32: "*The formalisation of the virtual hydrological framework for SRM evaluation provides the opportunity for further improvements in the future, including:*
>
> *(i) Using multiple, well-tested hydrological models - a potential limitation of the virtual hydrologic evaluation framework is that it is reliant on the use of a hydrological model. Hydrological structural errors may potentially skew interpretation of the SRM evaluation if the hydrological model poorly represents the catchment processes. To reduce these impacts the steps taken in this study included (a) using a well-tested hydrological model that has demonstrated good performance on a wide range of catchments (e.g. the GR4J model has been widely tested , see (Perrin et al., 2003, Coron et al., 2012); (b) calibrating and evaluating the hydrological on a catchment close to the observed rainfall sites to ensure it provided sufficiently good performance  (e.g. GR4J was calibrated to the Onkaparinga catchment - see (Westra et al., 2014a, Westra et al., 2014b). Future research will use multiple, well-tested hydrological models with sufficiently good performance to reduce the reliance on a single hydrological model and ensure the identification of SRM deficiencies is not dependent on a single hydrological model.*"
* * *
**Comment 11:**

*P9Fig3b Maybe the authors can spend a more detailed explanation of the two different indices k and t. For me the difference was not quite clear at the beginning. Also, it is clear that rainfall in June can affect the runoff in July (or from April by filling storages and hence affecting runoff in July). But how can rainfall in July affect runoff in June, although the months August to January obviously don't? Is the rainfall information transformed into runoff over such a long period in the model? Since there is no rain in the summer half year, shouldn't the storages run empty?*
* * *
**Response 11:**

We have more clearly indicated the meaning of indices *k* and *t* in the descriptive text (see Sections 2.4.1 and 2.4.2), Figure 3b (Figure 4b in the revised manuscript) and its caption to aid the reader.

[revised manuscript text omitted]

A unit test is undertaken by evaluating the ensemble of simulated streamflows from transforming the spliced rainfall from the 12 potential influencing months for an evaluated month, $t$. We believe it is necessary to evaluate all 12 potential influencing months because *a priori* the impacts of 'poor' rainfall can have long-term impacts on streamflow statistics due to catchment storage in the rainfall-runoff model. Some catchment models have short-term stores to represent features such as depressions, basins, and channels, but other catchments models have long-term stores to represent the long-term memory in subsurface catchment storages) that can have memory over multiple months. For the case study catchment the storages do not run empty each year in summer, so there is potential for persistence at longer timescales due to this 'memory' in the catchment. Therefore, it is plausible that rainfall from 12 months prior can influence the current state of a catchment (especially if that month/season was anomalously wet or dry). An additional figure of monthly rainfall and streamflow boxplots is now provided to illustrate the highly seasonal nature of the case study catchment in Section 3 (also see response to comment 14). The new figure (Fig. 6) is reproduced below for convenience.

[Figure]

**Fig. 6 Seasonal variation of catchment average rainfall and flow at Houlgrave Weir. Boxplots show the variation across years. Wettest indicates (high rainfall, high flow), direst indicates (low rainfall, low flow) and wetting-up indicates (high rainfall, low flow).**

**Comment 12:**

*P9Section 2.4 It would be useful for the reader to illustrate the implementation of the framework with a flow chart, since the authors use step 1, step 6 and so on throughout the section (and the manuscript).*

**Response 12:**

We have incorporated a flow chart to illustrate the framework procedure in Section 2.1 (see Fig. 2 in the revised manuscript). For convenience the flow chart is reproduced below.

[Figure]

**Fig. 2 - Virtual hydrological evaluation procedure.**

In the revised manuscript the framework steps have been streamlined (from seven steps down to three) to further improve the presentation of the framework and Section 2 has been restructured to better integrate the framework steps and the specific virtual tests (i.e. integrated and unit tests). The structure of Section 2 is now as follows:

> *2. Virtual hydrological framework*
>> *2.1 Overview*
>> *2.2 Step 1 – Identify poor performing sites*
>>> *2.2.1 Selection of primary streamflow characteristic and relevant hydrological model*
>>> *2.2.2 Integrated test procedure*
>>> *2.2.3 Identify poor performing sites using CASE framework*
>> *2.3 Step 2 – Identify poor performing time periods*
>> *2.4 Step 3 – Identify sources of poor performance*
>>> *2.4.1 Unit test procedure*
>>> *2.4.2 Compare unit tests and integrated tests*
>>> *2.4.3 Identify types of key deficiencies*

Section 4 (Results) has also been restructured to follow the streamlined framework procedure. The structure of Section 4 is now as follows:

> *4. Results*
>> *4.1 Step 1 – Identify poor performing sites*
>> *4.2 Step 2 – Identify poor performing time periods*
>> *4.3 Step 3 – Identify sources of poor performance*
>>> *4.3.1 Streamflow errors mostly originate from rainfall model deficiencies in the evaluated month*
>>> *4.3.2 Streamflow errors originate from rainfall model deficiencies over a contiguous block of months*
>>> *4.3.3 Streamflow errors originate from rainfall model deficiencies in a preceding month more so than evaluated month*
>>> *4.3.4 Influence of monthly rainfall on annual flow volumes*

**Comment 13:**

*P10l14-17 What is the 90 % limit of the simulated statistic? If m=10 mm, everything between 1 mm and 19 mm is considered as good? Here an additional explanation is required.*

**Response 13:**

Thank you. We have added additional information to explain the 90% limit test as requested both in text and graphically (see Section 2.2.3, Table 2 and Fig. 3). The text and figures are reproduced below for convinience:

> Page 7, Line 15-22: " *The integrated test results aim to identify the sites that are poor performing for the primary streamflow characteristic. Model performance is categorised using a CASE framework approach as 'good', 'fair' or 'poor' following Bennett et al. (2018). The quantitative tests for each performance category are provided in Table 2 alongside an illustration of each in Fig. 4. The quantitative tests proceed by comparing the statistics of the virtual-observed streamflow against those calculated from replicates of the simulated streamflow. Performance was categorised as 'good' if the selected statistic for the virtual-observed streamflow fell within the 90% limits of the statistic calculated from the simulated streamflow replicates (Fig. 4, case i), as 'fair' if the virtual-observed statistic fell outside the 90% limits of the simulated streamflow replicates but within the 99.7% limits (Fig. 4, case ii) and otherwise as 'poor' (Fig. 4, case iii).*"

**Table 2 CASE performance classification criteria. Adapted from Bennett et al. (2018).**

| Performance Classification | Test | Key |
|---|---|---|
| **'good'** | Observation lies within the 90% limits (case i) | 🟩 |
| **'fair'** | Observation lies outside the 90% limits but within the 99.7% limits (case ii) | 🟨 |
| **'poor'** | Otherwise (case iii) | 🟥 |

[Figure]

**Fig. 4 Illustration of performance classification, case (i) shows 'good' performance, case (ii) shows 'fair' performance and case (iii) shows 'poor' performance. Adapted from Bennett et al. (2018).**

**Comment 14:**

*P11l9-10 From Table 2 it cannot be seen, how long the time series used for the calibration of the rainfall generator are. It would be useful to the reader to characterize the time series more in detail (wet spell durations and amount, dry spell durations and maybe even on a monthly basis, since further investigations are carried out on a monthly basis). At least a hint to Fig. 6 and Fig. 7, which include some monthly observations, would be useful.*

**Response 14:**

We agree with the reviewer and recognise that the high-level summaries need more tangible details on the rainfall and streamflow statistics on a monthly basis to help the reader understand the seasonal behaviour of the case study catchment. We have revised Table 2 (now Table 3) to characterise the rainfall time series in more detail including the addition of columns that present rainfall statistics (total rainfall, no. of wet days,

average daily rainfall, average wet day length, average dry spell durations) in different seasons – for brevity in this table we show two months: January to represent the dry summer and July to represent the wet winter. Further detail on these statistics for all months at each site is also provided as supplementary material. Table 3 is reproduced below:

**Table 3 Site names, locations and seasonal rainfall characteristic summary. Sites ordered from lowest to highest elevation.**

| | | | | January | | | | | July | | | | |
| Site No | Site Name | Elev (m) | Ann. Av. Rain (mm) | Total (mm) | No. Wet (days) | Daily Av. (mm) | Wet-spell (days) | Dry-spell (days) | Total (mm) | No. Wet (days) | Daily Av. (mm) | Wet-spell (days) | Dry-spell (days) |
|---|---|---|---|---|---|---|---|---|---|---|---|---|---|
| 19 | Old Noarlunga | 7 | 520 | 20 | 4.1 | 0.6 | 1.6 | 9.6 | 72 | 17 | 2.3 | 3.1 | 2.8 |
| 16 | Morphett Vale | 90 | 560 | 20 | 4.1 | 0.6 | 1.5 | 8.9 | 76 | 17 | 2.4 | 3.3 | 2.8 |
| 10 | Happy Valley | 148 | 640 | 22 | 4.8 | 0.7 | 1.7 | 8.2 | 88 | 18 | 2.8 | 3.6 | 2.6 |
| 21 | Willunga | 158 | 640 | 23 | 4 | 0.7 | 1.6 | 10 | 95 | 17 | 3 | 3.2 | 2.7 |
| 5 | Clarendon | 223 | 820 | 25 | 4.7 | 0.8 | 1.7 | 8.9 | 114 | 17 | 3.7 | 3.4 | 2.8 |
| 6 | Coromandel | 234 | 710 | 24 | 4.8 | 0.8 | 1.8 | 9.2 | 102 | 18 | 3.3 | 3.6 | 2.8 |
| 13 | Macclesfield | 302 | 730 | 28 | 5.3 | 0.9 | 1.8 | 7.9 | 99 | 17 | 3.2 | 3 | 2.7 |
| 15 | Cudlee Creek | 311 | 830 | 29 | 5 | 0.9 | 1.8 | 8.4 | 123 | 18 | 3.9 | 3.8 | 2.7 |
| 11 | Harrogate | 335 | 550 | 23 | 3.5 | 0.7 | 1.6 | 12 | 75 | 12 | 2.4 | 2.2 | 3.8 |
| 4 | Cherry gardens | 345 | 920 | 30 | 5.4 | 1 | 1.8 | 7.7 | 134 | 18 | 4.3 | 3.8 | 2.6 |
| 8 | Gumeracha | 346 | 790 | 27 | 5.3 | 0.9 | 1.8 | 7.8 | 108 | 18 | 3.5 | 3.5 | 2.6 |
| 9 | Hahndorf | 347 | 850 | 29 | 5.4 | 0.9 | 1.9 | 8.1 | 123 | 18 | 4 | 3.4 | 2.7 |
| 17 | Mount Barker | 349 | 770 | 28 | 5.9 | 0.9 | 1.9 | 7.2 | 104 | 18 | 3.3 | 3.3 | 2.6 |
| 7 | Echunga | 375 | 805 | 28 | 5 | 0.9 | 1.8 | 8.7 | 110 | 17 | 3.5 | 3.3 | 2.6 |
| 3 | Bridgewater | 376 | 1050 | 32 | 5.2 | 1 | 1.9 | 8.9 | 154 | 18 | 4.9 | 3.6 | 2.7 |
| 14 | Meadows | 384 | 870 | 30 | 4.8 | 1 | 1.7 | 8.5 | 122 | 17 | 3.9 | 3.2 | 2.7 |
| 2 | Birdwood | 385 | 720 | 25 | 4.4 | 0.8 | 1.8 | 9.6 | 104 | 17 | 3.4 | 3.4 | 2.8 |
| 1 | Belair | 386 | 790 | 28 | 4.6 | 0.9 | 1.8 | 9.8 | 111 | 16 | 3.6 | 3.2 | 3 |
| 22 | Woodside | 387 | 800 | 27 | 4.3 | 0.9 | 1.6 | 8.3 | 121 | 16 | 3.9 | 2.9 | 2.7 |
| 18 | Nairne | 403 | 680 | 28 | 4.7 | 0.9 | 1.6 | 8 | 93 | 16 | 3 | 2.8 | 2.8 |
| 12 | Lobethal | 470 | 880 | 28 | 4.9 | 0.9 | 1.8 | 8.4 | 133 | 18 | 4.3 | 3.5 | 2.6 |
| 20 | Uraidla | 499 | 1090 | 35 | 4.7 | 1.1 | 1.8 | 9 | 161 | 17 | 5.2 | 3.4 | 2.7 |

In addition, a new figure that shows seasonal variation of catchment average rainfall and streamflow on a monthly basis has been added to Section 3 (Case Study) of the main paper.

[Figure]

**Fig. 6 - Seasonal variation of catchment average rainfall and flow at Houlgrave Weir. Boxplots show the variation across years. Wettest indicates (high rainfall, high flow), direst indicates (low rainfall, low flow) and wetting-up indicates (high rainfall, low flow).**
* * *
**Comment 15:**

*P11l16 For the calibration of the model the reader is referred to Westra et al. (2014), which is a non-reviewed technical report with 100+ pages, as far as I can see. In context with my former specific comment it is necessary to provide information in the actual manuscript, how the model has been calibrated. Which rainfall data was used for the calibration? If all 22 stations have been applied, how was the areal rainfall estimated as input for the lumped r-r model?*
* * *
**Response 15:**

The calibration of the hydrological model was actually based on the approach used in a water resources research paper (Westra et al., 2014a) is now be cited alongside the report. The paper provides a compact peer-reviewed summary of the model and its calibration – for a neighbouring catchment (Scott Creek). This paper was acknowledged with a Research Spotlight Award from American Geophysical Union (top 5% of papers in

AGU). The reference to the report, Westra et al. (2014b), is also retained since it gives details specific to the Onkaparinga catchment used in this paper and because it is comprehensive. The Scott Creek and Onkaparinga catchments were calibrated as part of the same project using consistent models and techniques.

The relevant aspects of the calibration and model-selection have been added to the manuscript.

Page 13, Lines 1-12:

> *"The hydrological model GR4J (Perrin et al., 2003) was used to simulate virtual-observed streamflow at a daily time step. GR4J is a daily lumped hydrological model that simulates daily streamflow in a parsimonious manner using four parameters. The GR4J model was calibrated according to the procedure set out in Westra et al. (2014b) for the stationary version of the GR4J hydrological model. The details are provided in (Westra et al., 2014a) and a short summary is provided here. The multi-site rainfall gauges were Thiessen weighted to calculate the catchment average rainfall. The model was calibrated to the streamflow data at Houlgrave Weir (see Figure 4) using model calibration period of 15 years (1985-1999). The parameters were estimated using maximum likelihood estimation procedure with a weighted least squares likelihood function. The model parameters that maximised the likelihood function were found using a multi-start quasi-Newton optimisation procedure with 100 random starts. Overall, the GR4J model was a good fit to the observed streamflow, with a Nash-Sutcliffe efficiency of 0.8. A similar type of hydrological model and calibration approach has been used for other virtual evaluation studies (Li et al. 2014; 2016). The same set of hydrological model parameters are used for both the unit and integrated tests so that the same transformation of rainfall to flow is used."*
* * *
**Comment 16:**
**P11Section 3 Although the observed discharge time series is not used in the investigation, it would be useful for the reader to provide some runoff characteristics (e.g. mean discharge) to get a feeling for the catchment.**
* * *
**Response 16:**

Details of the catchment's runoff characteristics at the annual and seasonal level have been added to Table 2 (now Table 3 in the revised manuscript) and a new figure (Fig. 6) showing the catchment's seasonal runoff and rainfall characteristics has also been added (see response to comment 13). Additional descriptions of key portions of the catchment cycle (i.e. "wettest", "driest", "wetting-up") have also been added and used throughout the manuscript. The new figure is reproduced below for convenience.

[Figure]

**Fig. 6 - Seasonal variation of catchment average rainfall and flow at Houlgrave Weir. Boxplots show the variation across years. Wettest indicates (high rainfall, high flow), direst indicates (low rainfall, low flow) and wetting-up indicates (high rainfall, low flow).**

**Comment 17:**
*P11l14-18 On p9l15-17 you mention "The hydrological model should be selected on the basis that it is capable of simulating streamflow for the timescales, magnitudes and physical processes of interest to the intended application." Is the lumped model able to simulate the physical processes of a catchment with a few 100 km2 catchment area (I could not find the catchment area in the manuscript).*

**Response 17:**

Thank you for pointing this out. The catchment area (323 km$^2$) has been included in Section 3 of the revised manuscript.

It is important that the chosen hydrological model is fit for purpose (see also discussion of comment 10). The GR4J model used in this paper is for catchment inflows to the Mount Bold reservoir and is appropriate for analysis of catchment yield (i.e. focussed on means and variances of annual inflow)[2].

However, if we were examining impacts on instantaneous peak flows impacts, this model would not be suitable and if we wanted to look at impacts of distributed rainfall, we would need a distributed rainfall-runoff model. However, for the purpose of this paper, which is to demonstrate the virtual hydrological framework (including the unit test) for evaluating the ability of a stochastic rainfall model to estimate catchment yield, the model is deemed sufficient.

**Comment 18:**
*P12l8-10 Which result is analyzed? Integrated test or unit test?*

**Response 18:**

We agree this is not clear, thank you for pointing it out. The result is from the integrated test (step 4). This is clarified in the revised text.

**Comment 19:**

*P19l19 In Fig. 5 the results for rainfall are worse than for runoff (for mean values).*

**Response 19:**

There are many interesting features of Fig 5 (now Figure 7) like this. The fact that there is not a direct correspondence between 'good' rainfall and 'good' runoff, and/or 'poor' rainfall and 'poor' runoff is one of the motivations for the virtual hydrological evaluation framework in addition to observed rainfall-based evaluation. Figure 7 shows that it is possible for seemingly 'poor' rainfall to yield 'good' runoff (as also 'good' rainfall can yield 'poor' runoff). We note that the discrepancy in Figure 7 is not in terms of mean values, but for the standard deviation of monthly aggregates (see Figure 5, *sd*(total) for rainfall and runoff in Jan, Mar, May, Jun, Oct, Nov, Dec). For drier months (Nov-Mar) the lack of correspondence (i.e. 'poor' rainfall producing 'good' runoff) is due to the low amount of runoff. While in wetter months (May-Oct) the relationship is more complicated as shown in the unit test demonstrations (Section 4.3).

The description of this Figure has been revised to better explain these features in Section 4.2 of the revised manuscript and is reproduced below.

> Page 15, Line 11 – Page 16, line 12: "*The poor performing sites identified in Step 1 were then compared in terms of both an observed-rainfall evaluation and virtual hydrological evaluation via an integrated test. Fig. 7 graphically summarises this comparison, with each row presenting monthly or annual performance of the following statistics:*
> * *simulated daily rainfall statistics (mean (m) daily amounts, standard deviation (sd) of daily amounts, mean number of wet days (nwet) and the standard deviation of the number of wet days);*
> * *aggregate rainfall statistics (mean and standard deviation of total rainfall); and*
* * *
[2] The GR4J model has a calibrated Nash Sutcliffe of 0.8 (reported in the original manuscript). This model (and its non-stationarity variants) were used to project climate change impact on the Onkaparinga catchment (Westra et al. 2014b).

- *aggregate streamflow statistics (mean and standard deviation of total flow).*

*The first to fourth columns of Fig. 7 summarise the observed-rainfall evaluation and the fifth and sixth of Fig. 7 summarise the virtual hydrological evaluation. The first column of Fig. 7 indicates that of the poor performing sites the SRM exhibited 'good' performance in simulating daily rainfall means and standard deviations as well as the mean number of wet days for all sites and months and at an annual level according to the observed-rainfall evaluation. Whereas the second column indicates that there is mixed performance across sites and months in simulating the variability in the number of wet days (sd(nwet)). Likewise, the third and fourth columns indicate overall 'good' performance in simulating mean monthly totals and mixed performance in simulating the monthly or annual total standard deviations (sd(total)). Whereas the virtual hydrological evaluation (fifth and sixth) columns show mostly 'good' performance in all months other than those in the 'wettest' or 'wetting-up' periods.*

*A clear trend, from Fig. 7 is the contrast in performance between the observed-rainfall evaluation and the virtual hydrological evaluation. One constrast is that, in the driest months (Dec, Jan, Feb) 'poor' performance in simulating rainfall (based on observed-rainfall evaluation) did not necessarily translate to 'poor' performance in simulating streamflow (based on virtual hydrological evaluation). For example, examining the first row of Fig. 7, the observed-rainfall evaluation shows that in January the SRM's ability to simulate variability in the number of wet days, sd(nwet), was 'poor' for all sites. However, in contrast the virtual hydrological evaluation shows that most sites had 'good' performance in simulating the January distribution of monthly total flow (i.e. m(total) and sd(total)).*

*A second contrast is that 'good' performance in the observed-rainfall evaluation does not necessarily translate to 'good' performance for the virtual hydrological evaluation, particularly for months in the 'wettest' and 'wetting-up' periods. For example, in Fig. 7 the rows summarising June and August show large percentages of 'poor' sites in the virtual hydrological evaluation of monthly total flow. This deficiency would have been difficult to infer using the observed-rainfall evaluation due to the 100% 'good' performance of m(total) rainfall and 'good/fair' performance of sd(total) rainfall in these months.*

*Likewise, by examining the bottom row of Fig. 7 that summarises annual performance, it can be seen that the observed-rainfall evaluation shows unbiased mean annual total, m(total), rainfall (100% 'good') and yet the mean annual total flows showed only 10% of sites as 'good'. Discussion of the unit tests in the following section will investigate reasons why apparently 'good' rainfall can yield 'poor' flow."*
* * *
**Comment 20:**
**P19 Results-section Before it was mentioned that also the influence of spatial rainfall patterns can be evaluated. Since this is not done in the manuscript, it can be moved to the outlooks of the manuscript. Otherwise a spatial analyses can be implemented in the manuscript (what I would recommend), to show further advantages of the unit test.**
* * *
**Response 20:**
This concept has not been demonstrated in the main paper, therefore it has been deferred to the discussion on outlooks (Section 5.3). We believe the unit test has sufficient novelty to represent a substantial contribution, hence we will the leave spatial rainfall evaluation for future developments.

**Comment 21:**

*P20l20-21 With the introduced framework it is still not possible to identify, which rainfall characteristics are important for streamflow prediction. Based on the high non-linearity of the rainfall-runoff transformation process, a single rainfall characteristic cannot be sufficient to draw conclusions about the impact on the resulting runoff. If this would be the case, r-r models wouldn't have to be used anymore. However, could the authors identify, based on their analysis, which rainfall characteristics are important for the resulting runoff behaviour? (of course, the results depend on the study site, model choice and so on, but nevertheless…)*

**Response 21:**

The comment is correct that the proposed method does not identify the impact of specific singular rainfall characteristics on the resulting runoff. However, the framework does provide a clear approach to isolate which set of components of the rainfall model require further attention. The integrated test focuses attention on hydrological properties, and the unit test can isolate deficiencies in rainfall by month. When applied to the case study in our paper, the limitations of the model are in the variability of the rainfall and not in the rainfall mean. The initial motivation for the approach can be seen in Figure 7 (formerly Figure 5) where the mean of the annual rainfall is 'good', but the mean of the annual runoff is 'poor'. Figure 7 also shows that this is mostly attributed to 'poor' mean runoff in June, July and August. Unit tests were then used to show that the rainfall in the catchment 'wetting-up' period (May-June) is of key importance. This is greater insight than could have been achieved with observed-rainfall evaluation and is greater insight than could be gathered from other virtual-observed streamflow approaches.

However, we agree with the reviewer that the framework cannot currently distinguish between particular features of the rainfall (e.g. "Is it rainfall correlation, magnitude, or intermittency that causes a low standard deviation in monthly streamflow?"). Nonetheless, the framework has significant potential to be extended to diagnose which are rainfall characteristics should be improved. This can be done by comparing multiple rainfall model variants (parametrically, or via bootstrap techniques) which are designed to have contrasting features of a key characteristic (e.g. intermittency, rainfall correlation). Such an approach was undertaken by Evin et al. (2018) using an observed-rainfall evaluation approach to compare model variants. In the revised manuscript, this is identified as a limitation and this extension is highlighted for further research (Section 5.3).

> Page 27, Line 22 - Page 28, Line 7: *"The formalisation of the virtual hydrological framework for SRM evaluation provides the opportunity for further improvements in the future, including: …*
>
> *(ii) Comparison of SRMs – this framework can be extended to provide more direct guidance on which rainfall features (in terms of components of the SRM) should be modified to improve streamflow performance. This can be done by comparing multiple rainfall model variants (parametrically, or via bootstrap techniques) which are designed to have contrasting features of a key characteristic (e.g. intermittency, rainfall correlation). Such an approach was undertaken by Evin et al. (2018) using an observed-rainfall evaluation approach. If the SRMs have monthly/seasonal autocorrelation (these were not significant for the rainfall in the Onkaparinga catchment) the unit testing approach would need to be extended by conditionally sample the simulated rainfall in a manner that preserves monthly correlations."*

**Comment 22:**
*P21l32-p22l1 This example is hard to follow, maybe the authors can extend it. From my understanding it depends on the calibration of the storage coefficients. If storage coefficients are small, the results from the monthly rainfall will be transferred to runoff immediately. This would be possible with the "traditional" approach.*

**Response 22:**
Thank you for pointing out that the example is hard to follow. The example and paragraph framing the example have been revised to better step through the example (i.e. what elements come from the integrated test or unit test). We have also added additional information on catchment seasonality in Section 3 to better explain the importance of the 'wetting-up' months and storage in the catchment (also see responses to comments 14 and 16).

**Comment 23:**
*P23 There is a reference of Li et al. (2015b), but no Li et al. (2015a). Also Li et al. (2016) is mentioned before Li et al. (2015b)*

**Response 23:**
Thank you, this has been corrected.

**Response to Reviewer 2**

**Major Comments**
* * *
**Comment 1:**

*The authors present a streamflow-based evaluation framework to assess the adequacy of hydrologic predictions from stochastic rainfall generators (SRGs). This is a "virtual framework" in that it benchmarks these predictions against streamflows produced by historical continuous simulations, rather than observed streamflow timeseries. The authors point out that this avoids the complicating issues of model structural errors. This is a useful approach to benchmarking SRGs, and could perhaps be applied to other fluxes of interest (not just streamflow) for which long-term observation records aren't just available.*
* * *
**Response 1:**

We are pleased that the reviewer found the approach useful for evaluating stochastic rainfall models and appreciated the wider potential application of the approach for other fluxes of interest.
* * *
**Comment 2:**

*I agree with the first reviewer, who stated that "the theoretical elements of the paper are very long". There seems to be a fair bit of repetition, or at least over-explanation, of the motivation, and I strongly recommend that the authors look closely at how Section 2 can be shortened.*
* * *
**Response 2:**

We have restructured the Introduction and Section 2 to shorten the presentation (and remove repetition) of the theoretical elements by 50 lines while maintaining the key points. Please also see the response to Reviewer 1, comment 7.
* * *
**Comment 3:**

*Generally, I think that the demonstration would be more illuminating if the authors used it to compare two or more SRGs and/or hydrologic models.*
* * *
**Response 3:**

We support this idea. In the future, the framework can be used to compare two or more SRG's for particular hydrological applications. Furthermore, by utilising two or more hydrological models in the virtual evaluation framework, it would reduce the dependence on the choice of hydrological model (which was raised by reviewer #1 see comment 10), because one could look for patterns of errors for a single SRG across two hydrological models.

However, our preference is not to include this in this paper, for the following reasons:

1. To include the details of a second SRG and/or a second hydrological model, as well as providing a complete explanation of the details of the framework, observed-rainfall evaluation, virtual observed streamflow evaluation, two different tests, the integrated and unit tests, would make the paper and/or analysis overly long. Reviewer #1 has asked for extra details and additional figures/tables to explain a wide range of details – including the hydrological model calibration (see reviewer #1 comment 15), extra streamflow analysis (see reviewer #1 comment 3). This is over and above the seven figures already included. If we included another SRG and hydrological model, the number of the figures could increase dramatically making the paper overly long and lose focus on the presentation of the framework.

2. The framework has not been presented before, in particular the unit test. Therefore, in the manuscript we present the evaluation of a single stochastic rainfall model to demonstrate the framework. This application to a single model has demonstrated some new insights; that the errors in streamflow for a

particular month can be affected by errors in the in rainfall from the previous 2 to 3 months. This innovation is something that has not been previously identified in the literature.

Once the framework has been established and explained in this paper, future work will undertake multiple SRG and/or multi-hydrological model comparisons, as suggested by the reviewer. A comment on this has been incorporated into the discussion (Section 5).

> Page 27, Line 22 - Page 29, Line 7: *"The formalisation of the virtual hydrological framework for SRM evaluation provides the opportunity for further improvements in the future, including:*
>
> *(i) Using multiple, well-tested hydrological models - a potential limitation of the virtual hydrologic evaluation framework is that it is reliant on the use of a hydrological model. Hydrological structural errors may potentially skew interpretation of the SRM evaluation if the hydrological model poorly represents the catchment processes. To reduce these impacts the steps taken in this study included (a) using a well-tested hydrological model that has demonstrated good performance on a wide range of catchments (e.g. the GR4J model has been widely tested , see (Perrin et al., 2003, Coron et al., 2012); (b) calibrating and evaluating the hydrological on a catchment close to the observed rainfall sites to ensure it provided sufficiently good performance  (e.g. GR4J was calibrated to the Onkaparinga catchment - see (Westra et al., 2014a, Westra et al., 2014b). Future research will use multiple, well-tested hydrological models with sufficiently good performance to reduce the reliance on a single hydrological model and ensure the identification of SRM deficiencies is not dependent on a single hydrological model.*
>
> *(ii) Comparison of SRMs – this framework can be extended to provide more direct guidance on which rainfall features (in terms of components of the SRM) should be modified to improve streamflow performance. This can be done by comparing multiple rainfall model variants (parametrically, or via bootstrap techniques) which are designed to have contrasting features of a key characteristic (e.g. intermittency, rainfall correlation). Such an approach was undertaken by Evin et al. (2018) using an observed-rainfall evaluation approach. If the SRMs have monthly/seasonal autocorrelation (these were not significant for the rainfall in the Onkaparinga catchment) the unit testing approach would need to be extended by conditionally sample the simulated rainfall in a manner that preserves monthly correlations."*
* * *
**Comment 4:**

***The demonstration of these methods is provided at the monthly timescale. While this timescale might be useful for applications of water supply, it is not meaningful for flood processes in all but the very largest watersheds. It is easy to picture a hydrologic model that produces adequate performance in terms of monthly flows, but not daily or subdaily extremes, while the opposite is also possible. Similarly, it is also probably an easier task to create a stochastic rainfall generator that works well for producing monthly means and associated variability than fine-scale extremes. Thus, the virtual framework in this manuscript may not be as broadly useful for extremes as the authors claim (at least flood extremes, droughts might be a different story). I thus recommend that the authors acknowledge this shortcoming, and "tone down" the framework's purported usefulness for flood risk (e.g. page 9 lines 17), since this remains unproven.***
* * *
**Response 4:**

Thank you. We have toned done the discussion of the framework's usefulness for flood risk applications as this is not demonstrated in the manuscript. For example, we have removed the example that referred to flood risk which appeared on page 9 line 17 of the original manuscript so the sentence now reads:

> *"For example, the distribution of annual total flow would be a suitable characteristic when investigating yield"*

Reviewer 1 (see comment 3) has suggested that we examine additional streamflow characteristics, in particular flow duration curves. We have done so and included the flow duration curves to provide supporting evidence of the identified deficiencies in the simulated rainfall as part of the framework. This provides a broader demonstration of the framework through an application that considers statistics at the finer (daily) scale.
* * *
**Comment 5:**

*It isn't clear how the boxplots (e.g. figure 3) are constructed. Is it the "13 errors" mentioned in page 8 line 24? Or is it somehow derived from the 10,000 synthetic rainfall years? Or the 73 years of observed data with synthetic rainfall "spliced in"? Either way, it isn't clear that the authors have avoided the proliferation of error metrics that they identify as a limitation of previous measures on page 3. If this method is applied to a large number of sites, it still seems like a not-entirely compact evaluation scheme. Perhaps the authors could clarify how this compares to other methods in this respect.*
* * *
**Response 5:**

Thank you for pointing this out. A related response is given to reviewer 1, comment 11. We have revised the example which explains how the unit test figure (those like Figure 4b in the revised manuscript) are constructed and used as a diagnostic (see Section 2.4.2 and 2.4.3). The companion figure (4b) and the caption has also been improved to step the reader through the figure.

The revised text clarifies that each boxplot is a summary of the error (the difference between the simulated and virtual-observed performance statistic, Eq. 6) across all replicates of the simulated time series. The "13 errors" relates to the number of boxplots displayed in figures of this type (the unit test for 12 influencing months and 1 integrated test). The description is reproduced below.

> Section 2.4.1, Pages 9 - 11: "Following Fig. 3(a), consider the time series of observed, $R^{obs}$, and simulated, $R^{sim}$, daily rainfall for each year (and replicate) at a given site. Fig. 3(a) illustrates the embedding of simulated rainfall $R_k^{sim}$ in an influencing month, $k$, within observed rainfall $R_m^{obs}$ for all other months $m \in \{1, \dots, 12 | m \neq k\}$. The resulting spliced rainfall time series $R_{(k)}^{spl}$ is denoted with respect to the influencing month, $k$, and has the same length as the corresponding observed $R^{obs}$ and simulated $R^{sim}$ time series.

$$R_{(k)}^{spl} = \bigcup_{m=1}^{12} \begin{cases} R_m^{sim} ; m = k \\ R_m^{obs} ; m \neq k \end{cases} \tag{3}$$

> For example, if June ($k = 6$) is selected as the influencing month, each year of the spliced time series, $R_{(6)}^{spl}$, would be composed as follows:

$$R_{(6)}^{spl} = \{R_1^{obs}, \dots, R_5^{obs}, R_6^{sim}, R_7^{obs}, \dots, R_{12}^{obs}\} \tag{4}$$

> The ensemble of $k = 1, \dots, 12$ spliced rainfall time series $R_{(k)}^{spl}$ for all influencing months and additional inputs (e.g. potential evapotranspiration) indicated by '…' are transformed according to a hydrological model $g[\,]$ to produce an ensemble of simulated streamflow, $Q_{(k)}^{spl}$. This procedure is repeated for all simulated rainfall replicates.

$$Q_{(k)}^{spl} = g\left[R_{(k)}^{spl}, \dots\right] \tag{5}$$

> By construction, the spliced rainfall is identical to the observed rainfall for all months other than the influencing month, so any errors in streamflow statistics can be attributed to the influencing month free from other factors.

> The full set of spliced rainfall (e.g. spliced rainfall for each month designated as the influencing month $R_{(k)}^{spl}; k = 1, \dots, 12$) is input to the hydrological model. This step is repeated for all available replicates of the spliced time series. The results of the unit test and the integrated test (Steps 1-2) are then investigated and compared selecting each month as the evaluated time period in turn as well as other key time periods (e.g. annual)."

[Figure]

[Figure]

**Fig. 5 Schematic of (a) the method of constructing a unit test by embedding simulated months in an observation time series, and (b) the error profile produced when using the integrated and unit tests for the evaluated time period of June ($t=6$) (box plot whiskers indicate the 90% limits of the simulated streamflow replicates). For the unit test the errors in the evaluated period ($t$) are calculated as the difference between $Q_{(k)}^{spl}$ and $Q_{(t)}^{vo}$. For the integrated test the errors are calculated as difference between $Q^{sim}$ and $Q_{(t)}^{vo}$.**

Section 2.4.2, Page 11, Lines 11-20: "Using the function $h[\ ]$ to denote a calculated statistic of interest (e.g. mean or standard deviation), the relative error in an evaluated time period $t$ (e.g. annual or particular month) is given by

$$\%Err_{(t)} = \frac{h\left[Q_{(t)}^{eval}\right] - h\left[Q_{(t)}^{vo}\right]}{h\left[Q_{(t)}^{vo}\right]} \times 100$$
$$(6)$$

where $Q_{(t)}^{vo}$ is the virtual-observed streamflow and $Q_{(t)}^{eval}$ is the simulated streamflow from the selected virtual hydrologic test (i.e. $Q^{sim}$ if integrated test or $Q_{(k)}^{spl}$ if unit test selected) in the evaluated time period $t$. This procedure is repeated for all replicates of the simulated streamflow such that a range of errors is reported for each test for the target time period.

Following the calculation of this error metric for all replicates of the integrated test and ensemble of unit tests ($k = 1, …, 12$) it is possible to investigate deficiencies in the simulated streamflow in terms of which influencing month(s) contribute more to the deficiencies in streamflow for the target time

period based on that statistic of interest. Thus, for each site, statistic and evaluated time period there are 13 sets of errors to compare."

The integrated test is designed to be a compact evaluation that includes multiple sites and statistics. Once the integrated test is completed and problems identified in the simulated streamflow, the more detailed unit test is applied to sites of interest (Step 3 of the framework in the revised manuscript). Figure 4 (formerly Figure 3) describes the unit tests, which are not designed to be undertaken on a large number of sites – they are designed to be more probing and are only undertaken on certain sites with problems, such as those identified by the integrated test.

The reviewer is right to point out that the example does make it look like the proliferation of error metrics is identified as a limitation. We can see that the accompanying example emphasises the large number of statistics rather than our intended key point: that there are difficulties in assessing trade-offs or the relative importance of statistics. The introduction has been revised to clarify that the key issue for observed-rainfall evaluation is the difficulty in understanding the relative importance of rainfall features in terms of streamflow generation and what to do when performance is 'mixed'.

> Page 2, Lines 3-9: "*Observed-rainfall evaluation is the most common method for SRM evaluation (Rasmussen, 2013, Wilks, 2008, Baxevani and Lennartsson, 2015, Srikanthan and Pegram, 2009, Evin et al., 2018, Bennett et al., 2018). As shown in Fig. 1(a) it involves comparisons between observed and simulated rainfall typically using a large number of evaluation statistics. Often, this method shows 'mixed' performance where many statistics are reproduced well, but some are poor. While these assessments are useful, a drawback is that it is difficult to ascertain if the rainfall model's performance is sufficient in terms of predictions of practical interest, which are typically streamflow-based. This means it is unclear if it is necessary to invest time and effort to address instances of poor performance, when the majority of statistics are well reproduced (Bennett et al., 2018, Evin et al., 2018).*"

**Comment 6:**

***Relatedly, in Figure 6 and 7, shouldn't the "obs rain" and "virtual obs. Flow" be a range, rather than a single value? There are 73 years of monthly rainfall and simulated flows… this variability would be valuable context for evaluating the variability of the stochastic realizations.***

**Response 6:**

Yes, the variability of the rainfall and simulated flows is valuable context.

In Figures 6 and 7 (now Figures 8 and 11) we present a higher-level summary to evaluate the model performance considering both mean conditions and their variability (i.e. standard deviations). The statistics presented in these figures are the observed rain and the virtual-observed flow means (left column) and standard deviations (right column) over the full 73 years, calculated for the 12 months respectively. There are 12 monthly means and 12 monthly standard deviations per realisation. We are not calculating a separate statistic for each year of the timeseries. The boxplots show the range of these monthly statistics for the 10,000 stochastic rainfall model replicates. This convention is common to other papers in the field (e.g. Bennett et al. 2018, Khedhaouiria et al. 2018, Evin et al. 2018, Frost et al. 2011, Frost et al. 2004, Srikanthan et al. 2004, etc.)

**Comment 7:**

*It is unclear how other meteorological forcings (temperature, etc.) are handled in this framework. The authors focus on stochastic rainfall generators, as opposed to stochastic weather generators, meaning that the other forcings must be supplied independently of the rainfall. I would imagine that this could create some serious issues in some cases if synthetic rainfall is spliced together with inappropriate series of temperature or other forcings; one can imagine getting strange results in terms of precipitation vs. ET balances, with unclear consequences for the evaluation results.*

**Response 7:**

The reviewer is correct, as the focus is on evaluating stochastic rainfall generators, the other forcings are supplied independently. In our case study, the potential evapotranspiration (PET) time series (our only other meteorological forcing) is unchanged from the observed values in all hydrological simulations (i.e. the same PET time series is used in the simulation of the virtual-observed streamflow, integrated tests and unit tests). This is important as the hydrological evaluation is a relative comparison of the observed and simulated rainfall, hence all other time series and parameters relating to the hydrological model are kept the same in all tests. This approach was also taken in Sikorska et al. (2018), where the impact of using different rainfall disaggregation schemes on resultant flow was tested using a hydrological model. For all these tests the historical observed temperature time series was used to enable a comparison between the rainfall elements only.

To assess this assumption for the Onkaparinga case study we have evaluated the rainfall-PET correlation in all months. There is a negative relationship, which accounts for a small portion of the variance, up to $R^2 = 0.11$ in drier summer months. shows the rainfall-PET correlations for a drier summer month (January) and a wet winter month (June).

[Figure]

**Figure 1: Rainfall-PET correlation (left) January and (right) June.**

While there is some non-zero relationship, we do not consider it to undermine the case study (since all other statistics of PET are reproduced and the relationship is mild). However, this may not be the case for other locations where the model is applied.

We have included the reviewer's recommendation to apply the framework more generally to stochastic weather generators in Section 5.3. The application would require care to ensure that the PET (or other weather variable) generator does not introduce other deficiencies.

> Page 27, Line 22 - Page 28, Line 19: *"The formalisation of the virtual hydrological framework for SRM evaluation provides the opportunity for further improvements in the future, including: …*
>
> *(iv) Evaluation of spatial performance – there are multiple opportunities to develop tests for spatial performance including (a) repeating the integrated test for all sites and for catchment average rainfall means it would be possible to diagnose whether specific locations or the spatial dependence causes*

*poor reproduction of streamflow statistics, (b) developing a spatial unit test (which is analogous to the temporal unit test but extended to space) where different combinations of sites are 'spliced' in the construction of catchment average rainfall – to evaluate the impact of 'mixed' performance in the SRMs between sites on the catchment average rainfall, and (c) these spatial unit tests could be used to evaluate stochastic weather generators (SWG) more generally as well as spatially distributed SRGs – though these would require a spatially distributed hydrological model."*
* * *
**Comment 8:**

*I have never developed my own SRG, but I imagine that it might be hard to know exactly how to use the results from this analysis to refine that generator, despite the authors' claim that this is a valuable use of the framework. It identifies performance by month, rather than by "rainfall characteristics" (pg. 21 line 9). It is useful to know whether the SRG performs well for some months than others, but what next? If the authors plan to continue research on this topic, I would suggest that a method that "tracks" the propagation of rainfall through the model might be more effective. To me, the most clear way of doing this is to track how different rainfall statistical moments translate to different statistical moments in the streamflow, using both historical and synthetic rainfall. Such an approach would be amenable to changing the evaluation timescale. For these reasons, I recommend that the authors delete the statement that this framework "should be an essential step in the development and application of stochastic rainfall models" (page 21 line 22-23). On a related note, the authors should comment on how this technique would apply to distributed (i.e. high-resolution gridded) SRGs and hydrologic models.*
* * *
**Response 8:**

The reviewer has raised some excellent discussion points. We provide specific responses below to the discussion points raised.

1. "*It is useful to know whether the SRG performs well for some months than others, but what next?*" – We agree that it is a useful feature and it provides much more information than observed-rainfall evaluation alone. This is a key innovation of the paper. Following an observed-rainfall evaluation the focus would have been on the months Jan, Feb, Nov, Dec, May and June (Bennett et al., 2018). However, based on the results of the virtual hydrologic framework, we now know that May-July are the key months when considering the hydrology and that the problems with modelled rainfall in Jan-Feb, Nov-Dec are less important. Also, we now know that rainfall in preceding months is important and not just the month in which the flow is evaluated, which is more information than before.

   We agree that it does not tell us exactly which rainfall characteristics to focus on. However, it is unlikely to be that simple – a single rainfall statistic is unlikely to translate into a single runoff statistic because streamflow integrates a range of rainfall processes (see also reviewer #1, comment 21).

   Now that we know in which months deficiencies originate, we can focus on those months and trial various alternatives to the rainfall model to address the problem. This is left for future research, as mentioned in Section 5.3.

   > Page 27, Line 22 - Page 28, Line 7: *"The formalisation of the virtual hydrological framework for SRM evaluation provides the opportunity for further improvements in the future, including: …*
   >
   > *(ii) Comparison of SRMs – this framework can be extended to provide more direct guidance on which rainfall features (in terms of components of the SRM) should be modified to improve streamflow performance. This can be done by comparing multiple rainfall model variants (parametrically, or via bootstrap techniques) which are designed to have contrasting features of a key characteristic (e.g. intermittency, rainfall correlation). Such an approach was undertaken by Evin et al. (2018) using an observed-rainfall evaluation approach. If the SRMs have monthly/seasonal autocorrelation (these were not significant for the rainfall in the Onkaparinga catchment) the unit testing approach would need to be extended by conditionally sample the simulated rainfall in a manner that preserves monthly correlations."*

2. "*I would suggest that a method that "tracks" the propagation of rainfall through the model might be more effective … most clear way of doing this is to track how different rainfall statistical moments translate to different statistical moments in the streamflow, using both historical and synthetic rainfall*" – This idea offers scope to extend the framework, and is something to consider in the future. In this paper, we describe the integrated test and then introduce the unit test in terms of 'splicing' monthly blocks of rainfall. However, the reviewer is right that the approach could be formulated differently to use different 'splicing' approaches. For example, to examine the percentage changes in resultant streamflow as a function of a particular change in the inputted rainfall. To allow for alternative approaches that 'track' the propagation of error we have softened our concluding remarks.

3. "*I recommend that the authors delete the statement that this framework "should be an essential step in the development and application of stochastic rainfall models"*" – We have softened the wording of this statement. It now reads:

> Page 29, Lines 2-4: "*The virtual hydrologic evaluation framework provides insights not available through conventional approaches and provides useful diagnostic ability for the development and application of SRMs.*"

4. "*On a related note, the authors should comment on how this technique would apply to distributed (i.e. high-resolution gridded) SRGs and hydrologic models.*" – This is an important topic. We have commented that this technique could be extended to distributed rainfall and hydrologic models in Section 5.3.
* * *
**Comment 9:**

*I wonder if this framework should consider the autocorrelation in monthly rainfalls when doing this splicing. I don't know too much about the climate of South Australia, but I can imagine that autocorrelation at least in dry periods can be quite important, and this is likely not preserved during the splicing. It's not clear what the implications would be for the resulting evaluation.*
* * *
**Response 9:**

This is a valid and interesting point. Bennett et al. (2018) demonstrated that the monthly autocorrelations are small for the Onkaparinga catchment (from -0.2 in drier summer months to 0.3 in the wetter winter months), and as a result this issue was not considered in the presentation of the framework.[3] We can appreciate that monthly/seasonal autocorrelation is a significant feature of other locations and that could be a limitation when applying this method. We briefly suggested in Section 5.3 that the issue of monthly autocorrelation could be explored as an extension of the model and have provided further details to explain how this might be achieved in the revised manuscript. For example, rather than naïvely splicing rainfall it might be possible to conditionally sample the simulated rainfall in a manner that preserves monthly correlations. The efficacy of this technique would require some exploration since there may be limitations arising from the conditional sampling.
* * *
**Comment 10:**

*Figure 3 and elsewhere: I don't understand what "(90% limits shown)" means.*
* * *
**Response 10:**

The reference to 90% limits indicates that the boxplot whiskers extend to from the 5th to 95th percentile values of the metrics based on the 10,000 replicates. The initial description of these figures (see Section 2.4.2 in the revised manuscript) and all figure captions has been revised to clarify that the "(90% limits shown)" indicates that the boxplot whiskers extend to the 90% limits of the 10,000 simulations for the presented statistic.
* * *
[3] Bennett at al. (2018) also demonstrated that the model sufficiently reproduced these small monthly autocorrelations.
* * *
**Comment 11:**

*Section 3: Mention basin size. Also, why are stations outside the watershed used?*
* * *
**Response 11:**

The basin size is included in the revised manuscript (323 km$^2$). All the sites identified in Figure 2 (now Figure 5) were used to estimate the catchment average rainfall for the rainfall-runoff modelling calibration. This is stated in the revised Section 3 of the manuscript.

> Page 13, Lines 1-12: *"The hydrological model GR4J (Perrin et al., 2003) was used to simulate virtual-observed streamflow at a daily time step. GR4J is a daily lumped hydrological model that simulates daily streamflow in a parsimonious manner using four parameters. The GR4J model was calibrated according to the procedure set out in Westra et al. (2014b) for the stationary version of the GR4J hydrological model. The details are provided in (Westra et al., 2014a) and a short summary is provided here. The multi-site rainfall gauges were Thiessen weighted to calculate the catchment average rainfall. The model was calibrated to the streamflow data at Houlgrave Weir (see Figure 4) using model calibration period of 15 years (1985-1999). The parameters were estimated using maximum likelihood estimation procedure with a weighted least squares likelihood function. The model parameters that maximised the likelihood function were found using a multi-start quasi-Newton optimisation procedure with 100 random starts. Overall, the GR4J model was a good fit to the observed streamflow, with a Nash-Sutcliffe efficiency of 0.8. A similar type of hydrological model and calibration approach has been used for other virtual evaluation studies (Li et al. 2014; 2016). The same set of hydrological model parameters are used for both the unit and integrated tests so that the same transformation of rainfall to flow is used."*

When estimating catchment average rainfall it is fairly common to use sites outside the catchment, to better represent the spatial variability and to avoid boundary effects. It is therefore important that a stochastic rainfall model is able to reproduce the rainfall statistics at all of the sites outlined in Figure 5. This is why we evaluated the stochastic rainfall model at all the sites indicated in Figure 5. There are further reasons why this is valid for a virtual approach. Most notably, because there is no comparison made with observed streamflow. The virtual hydrological evaluation uses the calibrated hydrological model as a tool to process the observed and simulated rainfall for comparison. The virtual-observed streamflow can be thought of as a virtual stream flow gauge. The virtual stream gauges have no physical location that they are trying to replicate. Instead the virtual stream gauges enable a synthetic test of the simulated rainfall.

Virtual hydrological evaluation of a single rainfall site is analogous to treating the information at the selected rainfall gauge (observed and simulated) as being representative of the catchment rainfall. This 'catchment rainfall' is then routed through the chosen hydrological model to produce simulated and virtual-observed streamflow at the 'virtual catchment outlet'. This type of virtual approach was used in a different context (the development of new techniques for flood frequency estimation) in which a calibrated hydrological model was 'moved' all over Australia (Li et al., 2016).

**Comment 12:**

***More importantly, is the rainfall hydroclimate stationary? If not, then it seems as though this whole issue of stochastic generation and comparison of resulting streamflows against a nonstationary continuous simulation would be more complicated. Please comment on this.***

**Response 12:**

We have now commented on this issue in Section 5.3.

> Page 27, Line 22 -  Page 28, Line 12: *"The formalisation of the virtual hydrological framework for SRM evaluation provides the opportunity for further improvements in the future, including: …*
>
> *(iii) Evaluation of temporal non-stationarity – this framework can be extended to evaluate the impact of non-stationarity on SRM model performance by applying it on a selected non-stationary period. Care would be needed in the selection of statistics to identify model performance (since the performance in different sub-periods could be masked when evaluating an overall period). A related issue is that the hydrological model should provide adequate performance across the range of non-stationary climate forcings to which it is subjected."*

The reviewer is right to point out the complicated nature of comparing stochastically generated rainfall against a nonstationary continuous simulation. In this paper we took steps to minimise this impact by careful selection of the observed rainfall period.[4]

**Minor Comments**

**Comment 13:**

***Page 1 line 15: change "months" to "seasons"- that is a more broadly relevant term. Hydrology varies seasonally, months are an arbitrary construct (this comment applies elsewhere in the paper, such as page 2 line 31)***

**Response 13:**

Thank you for pointing this out. Where possible the terms 'seasons' and 'time periods' are used in preferred to 'months'. Where 'months' form part of the specific analysis (i.e. influencing months) the term has been retained.

**Comment 14:**

***Throughout paper: I recommend introducing an acronym for stochastic rainfall models and using it throughout.***

**Response 14:**

Thank you, we have introduced an acronym for stochastic rainfall models (SRMs) and used it throughout the revised manuscript.
* * *
[4] The catchment experiences a significant rainfall decline in the early 2000's (see Westra et al 2014a and 2014b) due to the 'millennium drought'. This is why we choose an earlier rainfall period that finishes in 1986. Although this does not mean we have eliminated the impact of non-stationarity it has been reduced by taking this step.

**Comment 15:** *Page 1 line 9: change "is" to "has been"*

**Comment 16:** *Page 1 line 10: change "is given" to "has been paid"*

**Comment 17:** *Page 1 line 12: delete "whenever the simulated rainfall are poor"*

**Comment 18:** *Page 1 line 19: change "catchment cycle" to "annual hydrologic cycle"*

**Comment 19:** *Page 1 line 28: delete comma after "targeted"*

**Comment 20:** *Page 2 line 10: "and/or" is not appropriate in technical writing. Use "or"*

**Comment 21:** *Page 2 line 12: put "virtual experiments" in quotations when mentioned for the first time, for emphasis*

**Comment 22:** *Page 3 line 16: add comma after "poor"*

**Comment 23:** *Page 5 line 10: The goal is not to match streamflow observations. It is to match the statistics of streamflows*

**Comment 24:** *Page 3 line 12: add "model" before parameters*

**Comment 25:** *Page 11 line 16: grammar problem "was fit good"*

**Comment 26:** *Page 20 line 18: put "memory" in quotations.*

**Comment 27:** *Page 10 line 15: I think that "observed/virtual" is a strange term. Observations have very little usage in this study…*

**Comment 28:** *Page 14 line 14: This sentence is a bit awkward. It isn't perhaps so "common and obvious" to the reader.*

**Comment 29:** *Page 20 line 13: good place to mention that multiple SRGs could be used too, not just multiple hydrologic models.*

**Response 15 – 29:**

Fixed.

**Comment 30:** *Page 1 line 24-25: I recommend deleting "risks" after "floods" and "droughts" and changing it to "hazards" after "hydrologic"*

**Response 30:**

'Risks' has been deleted after 'floods' and 'droughts'. 'Hydrologic risks' has been retained in preference to 'hydrologic hazards'.

**Comment 31:** *Page 3 line 13: Why would you call ET "extraneous"? It is generally very very important.*

**Response 31:**

We agree ET is important it was referred to as extraneous only to evaluations of rainfall.

**Comment 32:** *Page 19 line 3-4: streamflow arises from more than just rainfall integration over a catchment area-what about ET, etc.?*

**Response  32:**

Thank you, we agree. The impact of ET and other catchment properties related to the production of streamflow are mentioned later in Section 5.1.

**Comment 33:** *Page 20 line 11-13: I don't understand this sentence. Certainly model performance depends on the chosen model.*

**Response 33:**

Due to a revision of the text the sentence no longer appears.

**Response to Reviewer 3**

**Comment 1:**

*The manuscript proposes a "virtual hydrological" framework useful for the performance evaluation of stochastic rainfall generators (SRGs). Differently from other studies involved on this topic, this work proposes 1) to evaluate the rainfall performances directly in terms of discharge by considering as benchmark the "virtual observed streamflow", i.e. the streamflow obtained by running the observed rainfall into the hydrological model, 2) to use two different tests to highlight discrepancies between observed and simulated rainfall for a specific site or month.*

*Although the topic is surely of interest for the readership of HESS, a major revision is required before to consider the manuscript suitable for the publication. Indeed, throughout the manuscript some important information are missing (for details see specific comments below) whereas the section 2 and details about the virtual hydrological framework should be shortened. Moreover, the outcomes of this study seem linked to the specific case study and the authors should discuss how the results could be generalized for different SRGs and hydrological models.*

**Response 1:**

Thank you for your comments. We have revised the manuscript to address the matters raised in particular we have:

- Shortened the Section 2 and other sections describing the virtual hydrological evaluation framework.
- Provided further details on the stochastic rainfall and hydrological models in Section 3.
- Outlined and discussed aspects of the virtual hydrological framework implementation that need further work (Section 5.3).
- Added better signposting in text to make clear the outcomes that are generic and those that are specific to the case study.

Greater detail is given in response to specific comments below.

**Comment 2:**

*Abstract section.*

*This section should be made clearer concerning both the explanation of the virtual hydrological framework features and the results obtained in the work. Specifically, lines 12-15 and 18-20 in page 1 are not clear without reading the paper.*

**Response 2:**

The abstract has been revised to provide more detail on the framework as well as improving signposting (within the text to better distinguish the framework outcomes and case study specific outcomes.

The revised abstract text is reproduced below:

> Page 1, Lines 8-24: "*Stochastic rainfall modelling is a commonly used technique for evaluating the impact of flooding, drought or climate change in a catchment. While considerable attention has been given to the development of stochastic rainfall models (SRMs), significantly less attention has been paid to performance evaluation methods. Typical evaluation methods employ a wide range of rainfall statistics. However, they give limited understanding about which rainfall statistical characteristics are most important for reliable streamflow prediction. To address this issue a formal evaluation framework is introduced, with three key features: (i) streamflow-based — to give a direct evaluation of modelled streamflow performance, (ii) virtual — to avoid the issue of confounding errors in hydrological models or data, and (iii) targeted — to isolate the source of errors according to specific sites and seasons. The*

*virtual hydrologic evaluation framework uses two types of tests, integrated tests and unit tests, to attribute deficiencies that impact on streamflow to their original source in the SRM according to site and season. The framework is applied to a case study of 22 sites in South Australia with a strong seasonal cycle. In this case study, the framework demonstrated the surprising result that apparently 'good' modelled rainfall can produce 'poor' streamflow predictions, whilst 'poor' modelled rainfall may lead to 'good' streamflow predictions. This is due to the representation of highly seasonally catchment processes within the hydrological model that can dampen or amplify rainfall errors when converted to streamflow. The framework identified the importance of rainfall in the 'wetting-up' months (months where the rainfall is higher but streamflow lower) of the annual hydrologic cycle (May and June in this case study) for providing reliable predictions of streamflow over the entire year despite their low monthly flow volume. This insight would not have been found using existing methods and highlights the importance of the virtual hydrological evaluation framework for SRM evaluation.*"
* * *
**Comment 3:**

*Section 2.*

**This section should be shortened deleting multiple repetitions about the framework description in subsections 2.2 and 2.3. Moreover, Figure 1 and 2 could be merged into one figure. Conversely, the section 2.4 should be improved (also adding a flowchart) to allow the readers to easily follow the section "results".**
* * *
**Response 3:**

This comment aligns with feedback from other reviewers. We have shortened the introduction and Section 2 (from 167 lines down to 127 lines) and merged Figures 1 and 2 as suggested.

Please also see the response to Reviewer 1 comment 12. We have significantly improved the explanation of the virtual hydrological framework procedure. The procedure has been simplified from seven steps down to three steps and the presentation of the framework in Section 2 has been restructured to better integrate the steps and specific virtual tests (i.e. integrated and unit tests). The structure of Section 2 is now as follows:

> *2. Virtual hydrological framework*
>> *2.1 Overview*
>> *2.2 Step 1 – Identify poor performing sites*
>>> *2.2.1 Selection of primary streamflow characteristic and relevant hydrological model*
>>> *2.2.2 Integrated test procedure*
>>> *2.2.3 Identify poor performing sites using CASE framework*
>> *2.3 Step 2 – Identify poor performing time periods*
>> *2.4 Step 3 – Identify sources of poor performance*
>>> *2.4.1 Unit test procedure*
>>> *2.4.2 Compare unit tests and integrated tests*
>>> *2.4.3 Identify types of key deficiencies*

A flow chart (figure 2) has been added to graphically summarise the process to make it easier to follow in terms of the method and results. The figure is reproduced below.

[Figure]

**Fig. 2 - Virtual hydrological evaluation procedure.**
* * *
**Comment 4:**

*Section 3. Some important details are missing in this section. In addition to the area of the catchment and the temporal resolution of the simulated rainfall, the authors should specify how the GR4J model is forced by observed rainfall and how it is calibrated. Is the observed catchment average rainfall used to force and calibrate the hydrological model? How many years of observed discharge data are used for calibration? Is this set of parameters used to simulate streamflow within the unit and integrated test?*
* * *
**Response 4:**

Other reviewers have made similar comments (reviewer #1 comments 14-17, reviewer # 2 comments 7 and 11). The following details have been included in the revised Section 3:

- Catchment area (323 km$^2$)
- Rainfall model resolution (daily)
- Hydrological model resolution (daily)
- Hydrological model calibration details such as the number of observed years (model calibration and selection: 1985-1999, model evaluation: 2000-2009), Thiessen weighting of rainfall gauges was used to calculate catchment average rainfall, and the impact of rainfall errors was considered in detail (see Westra et al. 2014a, Westra et al. 2014b).
- The same set of hydrological model parameters are used for the unit and integrated tests so that the same transformation of rainfall to flow is used.

Section 3 is reproduced below for convenience.

> Page 12, Line 10 – Page 13, Line 12: "*The Onkaparinga catchment in South Australia is used as a case study (Fig. 6). The 323 km$^2$ catchment lies 25 km south of the Adelaide metropolitan area and contains the largest reservoir in the Adelaide Hills supplying the region (Mount Bold Reservoir). The catchment has a strong seasonal cycle (shown in Fig. 6) where the driest months (December, January and February) exhibit low rainfall and low streamflow, the wettest months (July, August and September) have high rainfall and high streamflow and the 'wetting-up' period (April, May and June) has high rainfall and lower streamflow.*
>
> *There is a strong rainfall gradient (Table 3), with average annual rainfall ranging from approximately 500 mm on the coast (Site No. 19) to over 1000 mm in the region of highest elevations (Site No. 20). A breakdown of the rainfall characteristics (annual total, number of wet days, daily average amounts, wet-spell and dry spell durations) at each site on a monthly basis is provided in Supplementary Material A.*
>
> *The simulated daily rainfall was determined from the latent variable autoregressive daily rainfall model of Bennett et al. (2018) using at-site calibrated parameters. This rainfall model uses a latent variable concept, which relies on sampling from a normally distributed 'hidden' variable. The latent variable can then be transformed to a rainfall amount by truncating values below zero and by rescaling values above zero to match the observed rainfall's distribution. Here, the rainfall is rescaled using a power transformation.*
>
> *To calibrate the model the rainfall data at a given site is partitioned on a monthly basis and separate parameters are fit for each month. The mean and standard deviation of rainfall amounts, as well as the proportion of dry days is calculated. These statistics are matched to the corresponding properties of the truncated power transformed normal distribution. The at-site lag-1 temporal correlation is then calculated based on the observed wet day periods for a given month. This statistic is transformed to the equivalent correlation of the underlying latent variable by accounting for the effects of truncation to determine the autocorrelation parameter. Full details of the calibration procedure are provided in Bennett et al. (2018).*

*In this study the daily rainfall model was calibrated and simulated at 22 locations throughout the catchment that have long, high-quality records (Table 3). 10,000 replicates of simulated rainfall covering a 73 year period (1914-1986) were used.*

*The hydrological model GR4J (Perrin et al., 2003) was used to simulate virtual-observed streamflow at a daily time step. GR4J is a daily lumped hydrological model that simulates daily streamflow in a parsimonious manner using four parameters. The GR4J model was calibrated according to the procedure set out in Westra et al. (2014b) for the stationary version of the GR4J hydrological model. The details are provided in (Westra et al., 2014a) and a short summary is provided here. The multi-site rainfall gauges were Thiessen weighted to calculate the catchment average rainfall. The model was calibrated to the streamflow data at Houlgrave Weir (see Figure 4) using model calibration period of 15 years (1985-1999). The parameters were estimated using maximum likelihood estimation procedure with a weighted least squares likelihood function. The model parameters that maximised the likelihood function were found using a multi-start quasi-Newton optimisation procedure with 100 random starts. Overall, the GR4J model was a good fit to the observed streamflow, with a Nash-Sutcliffe efficiency of 0.8. A similar type of hydrological model and calibration approach has been used for other virtual evaluation studies (Li et al. 2014; 2016). The same set of hydrological model parameters are used for both the unit and integrated tests so that the same transformation of rainfall to flow is used."*
* * *
**Comment 5:**

***Finally, major details should be added to this section about the rainfall statistics used for the calibration of the SRG of Bennett et. al. (2018).***
* * *
**Response 5:**

In the revised manuscript a summary is provided of the calibration approach for the rainfall model (see Section 3) so that it is easier for the reader to understand the model without needing to also read Bennett et al (2018).

*Page 12, Lines 19-32: "The simulated daily rainfall was determined from the latent variable autoregressive daily rainfall model of Bennett et al. (2018) using at-site calibrated parameters. This rainfall model uses a latent variable concept, which relies on sampling from a normally distributed 'hidden' variable. The latent variable can then be transformed to a rainfall amount by truncating values below zero and by rescaling values above zero to match the observed rainfall's distribution. Here, the rainfall is rescaled using a power transformation.*

*To calibrate the model the rainfall data at a given site is partitioned on a monthly basis and separate parameters are fit for each month. The mean and standard deviation of rainfall amounts, as well as the proportion of dry days is calculated. These statistics are matched to the corresponding properties of the truncated power transformed normal distribution. The at-site lag-1 temporal correlation is then calculated based on the observed wet day periods for a given month. This statistic is transformed to the equivalent correlation of the underlying latent variable by accounting for the effects of truncation to determine the autocorrelation parameter. Full details of the calibration procedure are provided in Bennett et al. (2018).*

*In this study the daily rainfall model was calibrated and simulated at 22 locations throughout the catchment that have long, high-quality records (Table 3). 10,000 replicates of simulated rainfall covering a 73 year period (1914-1986) were used."*

**Comment 6.1:**

*Section 4. In this section the authors should address the following points:*

*1) as the authors know, the rainfall simulated by the SRG are function of the rainfall statistic properties used to estimate the model parameters. According to the authors, in which way the rainfall statistical properties and the results obtained by the unit and integrated test are linked? If different rainfall statistical properties are used for the SRG calibration, are the results different? For instance, is the identification of the 10 "poor" sites sensitive a variation of rainfall statistics? If different statistics are used for SRG calibration, is it possible to reduce the streamflow errors? These aspects should be demonstrated/discussed by the authors in order to provide to the readers a general framework not tailored for a specific case study.*

**Response 6.1:**

It is important to clarify, that the goal of this paper is introduce a generic virtual-observed streamflow framework, and two tests (integrated and unit) that provide greater insight than traditional observed-rainfall evaluation approaches. This evaluation framework was demonstrated using a case study that included the rainfall model of Bennett et al. (2018) and a hydrological model for the Onkaparinga catchment. We demonstrated new insights/outcomes that are necessarily case study specific – but, they could not be derived, using traditional observed-rainfall evaluation techniques, and hence are a demonstration of the framework. As requested by the reviewer, we have more clearly signposted the generic components of the framework and the case study specific outcomes in the revised paper.

The selection of a rainfall model and its calibration approach, are independent of the generic evaluation framework. We agree with the reviewer, that if a different calibration approach (i.e. different rainfall statistics) were used, then the results may change, and the streamflow errors may reduce. Indeed, evaluating how different calibration approaches can influence the streamflow would be an excellent future use of this evaluation framework. As the paper has increased substantially to address the issues raised by the reviewers (13 figures, ~10,000 words), examining this issue is outside of scope. We have briefly touched on this idea in the revised discussion section (Section 5.3) as a future research application of the framework.

**Comment 6.2:**

*2) as the streamflow generation is a results of the mean areal (rather than single-site) rainfall over the basin, before to apply the integrated test to identify sites for which the rainfall simulation is not good, it could be interesting to estimate the streamflow errors coming from the mean areal rainfall, evaluated as average over the 22 sites. How good are these streamflow time series with respect to the "virtual-observed" ones (obtained by the rainfall observed over the 22 sites and averaged over the catchment)? More interesting, the authors should highlight the benefits deriving from the use of integrated test. For instance, they should show what are the streamflow errors if only the 12 "good" sites are retained to evaluate the mean areal rainfall. Is it better than using all 22 sites?*

**Response 6.2:**

The reviewer is right – in a 'real-world' catchment, streamflow is the result of rainfall over a basin rather than a single site. Our framework does enable spatial characteristics to be tested and in this respect we agree with the reviewer about the potential utility of the proposed test. By using the framework it would be possible to undertake evaluations with catchment average rainfall.

In this paper we have not evaluated how the observed and simulated catchment average rainfall compare in terms of the resultant streamflow. This is because, as a matter of first priority, our approach focuses on identifying issues with rainfall at each site and getting this right before moving on to assess deficiencies in spatial properties. We therefore prefer to assess the at-site performance prior to the catchment average performance. Future work will demonstrate and apply the framework to catchment average rainfall.

The reviewer makes an excellent suggestion regarding future investigations into the impact of 'mixed' performance in the rainfall model between sites. We have added a discussion on how this could be explored

in future work and indicated that the proposed investigation of resultant streamflow (where different combinations of sites are 'spliced' in the construction of catchment average rainfall) is analogous to our temporal unit test but extended to space.

> Page 27, Line 22 – Page 28, Line 19: *"The formalisation of the virtual hydrological framework for SRM evaluation provides the opportunity for further improvements in the future, including: …*
>
> *(iv) Evaluation of spatial performance – there are multiple opportunities to develop tests for spatial performance including (a) repeating the integrated test for all sites and for catchment average rainfall means it would be possible to diagnose whether specific locations or the spatial dependence causes poor reproduction of streamflow statistics, (b) developing a spatial unit test (which is analogous to the temporal unit test but extended to space) where different combinations of sites are 'spliced' in the construction of catchment average rainfall – to evaluate the impact of 'mixed' performance in the SRMs between sites on the catchment average rainfall, and (c) these spatial unit tests could be used to evaluate stochastic weather generators (SWG) more generally as well as spatially distributed SRGs – though these would require a spatially distributed hydrological model."*
* * *
**Comment 7:**

***Page 2, lines 19-21: the example of Bennet et al. (2018) is not clear without reading the paper.***
* * *
**Response 7:**

Thank you for pointing this out. Due to a restructure of the introduction this example no longer appears.
* * *
**Comment 8:**

***Page 7, lines 14-15: This sentence is not clear. Please rephrase it.***
* * *
**Response 8:**

Due to a restructure the sentence no longer appears.
* * *
**Comment 9:**

***Page 8, line 24: Why the authors write "13 errors to compare"? Are the authors considering also the integrated test? It is not clear.***
* * *
**Response 9:**

Yes, the 13 errors compared arise from unit tests conducted for each of the 12 influencing months as well as an integrated test. We have restructured the presentation of methodology and revised the text explaining the calculation and comparison or errors to clarify this point. Please also see the response to reviewer 2, comment 5.
* * *
**Comment 10:**

***Page 8, lines 27-28. This example related to the integrated test is difficult to understand in this section. The reason is that in the previous section, where the authors describe the test there is no mention to the fact that the evaluation of the integrated test is carried out also at the monthly scale.***
* * *
**Response 10:**

We have improved the explanation of the framework to make clear that at Step 2 of the framework the integrated test results are evaluated at the monthly scale to allow a comparison with observed-rainfall evaluation and the unit test (see Section 2.3).

**Comment 11:**

*Figure 4: the position of the streamflow gauging station should be added in the figure.*

**Response 11:**

The streamflow gauge position has been added to the revised figure. It is indicated by the purple square.

[Figure]

**Fig. 6 Onkaparinga catchment, South Australia. Sites indicated by blue triangles are explored in greater detail in this paper due to the relatively poorer ability of simulated rainfall to reproduce annual streamflow totals at these sites.**

**Comment 12:**

*Page 15, lines 19-20: the conclusion about the transitional months should be drawn carefully. Indeed, moving from dry to wet conditions the process of formation of flood is very sensitive to the antecedent soil moisture conditions. Is the hydrological model able to reproduce observed streamflow in the transition period?*

**Response 12:**

Thank you, this is a fair point. Our application has focused on streamflow characteristics relevant to yield. In the framework we stress the importance of choosing a 'fit for purpose' hydrological model in terms of reproducing the streamflow characteristic of interest (Step 2). In our investigation we have chosen a widely applied model (GR4J) that has been calibrated using a rigorous approach (see Westra et al 2014a and 2014b).

The focus on rainfall in the wetting-up period was identified after we applied the new virtual framework. This key period would not have been identified using an observed-rainfall evaluation approach. Therefore, it was not possible to set hydrological model performance throughout this period as a criterion for hydrological model selection at the outset.

We agree with the reviewer that this is an important period of simulation to get right, and as a result of this analysis, we will examine hydrological models that have the potential to better simulate this period.[5] The
* * *
[5] Such models may include non-stationary variants of GR4J (Westra et al. 2014a and 2014b) or SUPERFLEX (Fenicia et al. 2011, Kavetski and Fenicia, 2011).

evaluation using different models is unlikely to change the conclusion that the rainfall in the wetting-up months are important, as the GR4J model already performs reasonably well. However, it may change the magnitude of the rainfall's impact on the hydrological performance – especially for future comparisons to observed streamflow. Further research will investigate this issue and has been added to the discussion Section.

> Page 28, Lines 22-302: *"The formalisation of the virtual hydrological framework for SRM evaluation provides the opportunity for further improvements in the future, including:*
>
> *(i) Using multiple, well-tested hydrological models - a potential limitation of the virtual hydrologic evaluation framework is that it is reliant on the use of a hydrological model. Hydrological structural errors may potentially skew interpretation of the SRM evaluation if the hydrological model poorly represents the catchment processes. To reduce these impacts the steps taken in this study included (a) using a well-tested hydrological model that has demonstrated good performance on a wide range of catchments (e.g. the GR4J model has been widely tested , see (Perrin et al., 2003, Coron et al., 2012); (b) calibrating and evaluating the hydrological on a catchment close to the observed rainfall sites to ensure it provided sufficiently good performance  (e.g. GR4J was calibrated to the Onkaparinga catchment - see (Westra et al., 2014a, Westra et al., 2014b). Future research will use multiple, well-tested hydrological models with sufficiently good performance to reduce the reliance on a single hydrological model and ensure the identification of SRM deficiencies is not dependent on a single hydrological model."*

---

## Author Response (AR2)

**Response to Editor**

*I have now received the reports from the referees. While the first reviewer mentioned several minor issues to be corrected, the second referee has more critical comments and suggestions that I would like you to consider. I kindly ask you to revise the manuscript once again. The revised manuscript will then be re-evaluated.*

*Looking forward to reading the revised text and best regards.*

This document describes our response to the reviewers' comments and our revision of the manuscript entitled 'A virtual hydrological framework for evaluation of stochastic rainfall models' (HESS 2018-489). We agreed with the majority of the issues raised by the two reviewers and editor. The key modifications we have made in the revised paper include:

- Improved the description of 'virtual experiments' in Section 1 (see Response 5).
- Revised Figure 1 to indicate the 'special' status of virtual-observed streamflow within the figure (see Response 4).
- Added clarification surrounding the differences between the CASE framework presented in Bennett et al. (2018) and the virtual hydrological evaluation framework presented in this manuscript (see responses to comments 15, 20, 22 and 45).
- Rephrased the first paper objective to clarify that the virtual hydrological evaluation framework is a novelty but draws on the systematic evaluation and tests introduced by the CASE framework.
- Incorporated additional details and clarified phrasings where suggested to aid reader understanding (see responses to comments 2-3, 4-8, 10, 15, 18-20, 22-23, 30-33, 36, 38, 41-44, 46-52).
- Revised Figures 3, 5, 9-10 and 12-13 as well as Table 3 to improve reader comprehension.

In particular we would like to thank Reviewer #2 for their thoroughness in raising over 40 issues and suggestions in this second revision of the manuscript. Their attention to detail has greatly improved this manuscript. We agree and addressed 90% of this reviewer's comments. The issues we have not addressed were either (i) issues previously raised in the 1st round of reviews and responded to in the 1st response (we have included the relevant text in this 2nd response for completeness); (ii) Deemed beyond the scope of the current paper, which already quite lengthy (revised paper ~10,300 words, 13 Figures, 3 Tables). Responding to these issues thoroughly would require substantial analysis better suited to a dedicated separate future paper to ensure they are addressed and presented in a meaningful way.

Our item-by-item responses to the specific comments appear below.

**Response to Reviewer 1**

**Comment 1:**

*Second Review of HESS-2018-489: "A virtual hydrological framework for evaluation of stochastic rainfall models" by Bennett et al.*

*The authors have by and large addressed my concerns, and I think the manuscript is almost ready for publication. I have a few minor recommendations:*

**Response 1:**

Thank you for your comments. We have revised the manuscript to address the matters raised in particular we have:

-   Improved the description of 'virtual experiments' in Section 1 (see Response 5).
-   Revised Figure 1 to indicate the 'special' status of virtual-observed streamflow within the figure (see Response 4).
-   Incorporated the recommended phrasing changes to improve the clarity of explanations (see Responses 2-3, 6-8).

Greater detail is given in response to specific comments below.

**Comment 2:**

*Pg 1 Line 10: I recommend changing "performance evaluation methods" to "developing methods to evaluate their performance". It isn't actually clear what you're referring to as it is written now.*

**Response 2:**

Done. The sentence has be re-written as suggested.

**Comment 3:**

*Pg 1 Line 22: change "higher" and "lower" to "high" and "low", since the current wording is ambiguous- higher or lower than what?*

**Response 3:**

Fixed. The sentence has be re-written as suggested.

**Comment 4:**

*Figure 1: The "virtual observed streamflow" box should definitely not be yellow, since it isn't observed. It should either be blue, or, I would suggest a diagonal line running through the box, with half being blue and half being yellow to indicate its "special" status.*

**Response 4:**

Thank you for pointing this out. Figure 1 has been amended to that the 'special' status of virtual-observed streamflow is clearly indicated.  Virtual-observed streamflow is denoted via the use of a diagonal line running through the box with half shaded blue and half shaded yellow. Figure 1(c) and the key are reproduced below:

[Figure]

**Figure 1 - Excerpt of revised Figure 1(c) in manuscript now indicating 'special' status of virtual-observed streamflow.**

**Comment 5:**

*Page 4 line 21: carefully define what is meant by virtual experiment.*

**Response 5:**

The paragraph has been updated to define 'virtual experiments' as experiments that focus on comparisons between streamflow simulated under different conditions or inputs without relying on comparisons to observed streamflow. The paragraph is reproduced below:

> Page 4, Line 21: '*To date, 'virtual experiments', that is, experiments that focus on comparisons between streamflow simulated under different conditions or inputs (i.e. virtual streamflow) without relying on comparisons to observed streamflow, have been used in a variety of contexts. Examples include (i) the evaluation of hydrological model sensitivity (e.g. Ball, 1994, Nicótina et al., 2008, Paschalis et al., 2013, Shah et al., 1996, Wilson et al., 1979) including the identification of rainfall features of interest in terms of hydrological behaviour (e.g. Sikorska et al., 2018), (ii) the development of new techniques for flood frequency analysis (e.g. Li et al., 2014, 2016), and (iii) the calibration, validation and selection of SRMs (e.g. Müller and Haberlandt, 2018, Kim and Olivera, 2011).*'

**Comment 6:**

*Page 4 line 21: Also, in this paragraph, references should be of the style "e.g." since these are likely just a few examples. The general idea behind the integrated test gets used over and over in hydrologic subdisciplines, so the referencing in this paragraph is in no way comprehensive.*

**Response 6:**

Fixed. All referenced examples are prefaced with 'e.g.' to clarify that the lists are not exhaustive.

**Comment 7:**

*Pg 6 line 15: replace "fit for purpose" with "suitable"*

**Response 7:**

Done. 'Suitable' has been inserted in place of 'fit for purpose'.

**Comment 8:**

*Section 5.3: There are a number of small issues with parentheses in the wrong place on references. Also pg 27 line 29 appears to be missing the word "model" and pg 28 line 6 should say "sampling" rather than "sample"*

**Response 8:**

Fixed.

**Response to Reviewer 2**

**Comment 9:**

*The manuscript introduces a new framework for the evaluation of generated rainfall time series in terms of their ability to reproduce runoff time series characteristics. This is done by two tests, an integrated test and a unit test. This topic is of broad interest for the hydrological scientific community and suitable for a publication in HESS.*

*I'm involved as reviewer for the second time. The revised manuscript has been significantly improved in response to the review comments and most of the issues raised in the first round of reviews have been addressed in a satisfactory way. The presentation and discussion of the method and the results is now clearer and more robust with the FDCs. However, there is still the need of improving some presentational aspects, especially in terms of method description. I have some follow-up questions on the responses and changes, plus some new suggestions, which are necessary to consider for the readability of the manuscript. Thus I would recommend moderate revisions for the manuscript.*

**Response 9:**

Thank you. We are pleased that the reviewer notes the significant improvement as a result of the first round of reviewer comments and that the topic is broad interest for the hydrological scientific community. We thank the reviewer for this thorough approach and attend to detail. Over 90% of the 44 comments raised by the reviewer have been addressed.

The suggestions and follow-up questions provided by the reviewer in this second round of comments have again led to an improved version of the manuscript. Regarding the specific matters raised:

- Added clarification surrounding the differences between the CASE framework presented in Bennett et al. (2018) and the virtual hydrological evaluation framework presented in this manuscript (see responses to comments 15, 20, 22 and 45). We also rephrased the first paper objective to clarify that the virtual hydrological evaluation framework is a novelty but draws on the systematic evaluation and tests introduced by the CASE framework.
- Incorporated additional details and clarified phrasings where suggested to aid reader understanding (see comments 10, 15, 18-20, 22-23, 30-33, 36, 38, 41-44, 46-52).
- Revised Figures 3, 5, 9-10 and 12-13 as well as Table 3 to improve reader comprehension.

Greater detail is given in response to specific comments below.

**Comment 10:**

*P1l29 „….evaluating…the efficacy of SRM's…comparisons to observed rainfall or streamflow are limited." Maybe the authors want to replace the „or" with „and" to emphasize the streamflow comparison. Comparisons of generated rainfall time series with observed ones are state of the art and applied in the majority of rainfall generation manuscripts (as the authors point out later), so I would not consider the body of literature as „limited".*

**Response 10:**

Thank you for pointing out that the sentence is ambiguous. The intention of this sentence was to point of that current approaches, while state-of-the-art, have limitations in their ability to evaluate the efficacy of SRM's in terms of the resulting streamflow. The sentence has been reworded as follows to avoid ambiguity:

> Page 1, Line 28: '*When evaluating the efficacy of SRM's, current approaches that make comparisons to observed rainfall or streamflow have limited diagnostic ability. They are unable to make a targeted evaluation of the SRM's ability to reproduce streamflow characteristics of practical interest.*'

**Comment 11:**

*P2l16 „poor predictive performance" In general, the rainfall-runoff (r-r) models are calibrated in before with observed time series. This pre-calibration and a subsequent comparison of simulated and observed runoff characteristics enable conclusions, if the hydrological model can be used for the estimation of the runoff characteristics. Hence, for the following „observed-streamflow evaluation" it is from my point of view not challenging to ascertain the „poor performance" as the authors point out later (p2l17). Maybe the authors can implement a discussion on that issue in the manuscript. In general, for the calibration of the r-r models several possibilities exist (using observed time series with a shorter time length than analysed later in the comparison or with lower network density or with coarser temporal resolution or from different data sources (satellite or radar data instead of station-data),….) and can be discussed in the manuscript.*

**Comment 12:**

*P2l10 Data errors do not necessarily occur only for single catchments and not for others, they can also appear in one catchment, but for a limited time period only. In combination with my previous comment, data errors should be identified before or during the calibration process. In general, I would exclude „data errors" and "r-r model structural errors" from the motivation for the introduced framework.*

**Responses 11–12:**

The reviewer raises issues around the difficulty of managing 'r-r model structural errors' and the presence of 'data errors'.

In the authors' experience the 'r-r model structural errors' are ubiquitous. This is often because simplified models are used to represent complex catchment processes. Therefore, these structural errors can have the error of masking the impact of errors in the SRM.

We agree with reviewer's comment that data errors can occur at any time. We also agree that ideally data errors would be identified before and during calibration. However, this is not routinely done, because typically it is quite hard to do and requires a great deal of both effort and information that is not routinely available. For example, data errors include runoff errors, which can be estimated if the rating curve (and any changes) is available, using Bayesian rating curve analysis (e.g. BaRatin; Le Coz et al. (2014)). Data errors also include errors in estimating catchment average rainfall from a limited number of rainfall gauges. These are quite challenging to estimate because the 'true' catchment rainfall is typically unknown (Renard et al., 2010), has high uncertainty often with standard errors exceeding 20%-30%, especially if the gauge network is sparse (Thyer et al., 2009, Linsley et al., 1982), and requires advanced hierarchical Bayesian techniques to integrate as part of the hydrological model calibration process (Renard et al., 2011). Typically, modellers take the best available runoff data, and catchment average rainfall data based on available data (with associated errors). Hence, for the majority of practical catchments, data errors are common, and cannot be excluded from model calibration process.

Both of these sources of error contribute to errors in the observed streamflow evaluation and mask the impact of the errors in the SRM. Based on this, we are of the opinion that they should be included as motivation for the framework.

**Comment 13:**

*P4l15-17 Observed runoff is not required for the virtual hydrological evaluation, but it is for the calibration of the model in before.*

**Response 13:**

Yes, in the calibration of the hydrological model observed streamflow would be used. However, it is not the basis of the streamflow evaluation for the virtual hydrological framework. The virtual hydrological evaluation framework is based on a relative comparison between the virtual-observed streamflow and the simulated streamflow–as both simulated and observed rainfall undergo transformation by the same process representation (i.e. the hydrological model). Provided that the hydrological model is calibrated such that the

necessary catchment streamflow features can be simulated (perhaps even via calibration to a neighbouring catchment with similar catchment properties) insight can be gained from this relative comparison.
* * *
**Comment 14:**

*P5l1 The authors state that the framework "categorises performance at multiple spatial and temporal scales". The case study includes one catchment, simulated with a lumped r-r model lumped and with daily resolution. What are the multiple spatial and temporal scales?*
* * *
**Response 14:**

The virtual hydrological evaluation framework is designed to evaluate the performance of SRMs (and not hydrological models). Hence the reference to multiple temporal scales refers to the evaluation of the SRM at the daily, monthly and annual scale in this manuscript and the ability to test the simulated rainfall at multiple sites or in aggregate across the catchment.
* * *
**Comment 15:**

*P5l5 The authors state that one key objective of the manuscript is the "introduction of a formalised framework for the virtual hydrological evaluation of SRMs", but the authors state before (p4l28) that this framework was developed before by Bennett et al. (2018, reference is missing in the reference list). What is the difference between both frameworks? Is the "introduction of the framework" still a novelty or is it the application of the existing framework introduced by Bennett et al. (2018)? Throughout the manuscript Bennett et al. (2018) is referenced. It is necessary to explain the differences between both studies and to enable a full understanding of the applied method/framework without reading the reference of Bennett et al. (2018).*
* * *
**Response 15:**

The Comprehensive And Systematic Evaluation (CASE) framework presented in Bennett et al. (2018) is an observed-rainfall evaluation framework. The CASE framework enables the systematic comparison of SRM performance through the use of a quantitative classification system that is applied at multiple spatial and temporal scales. The CASE framework, as presented in Bennett et al. (2018), is an observed-rainfall evaluation framework and therefore it does not consider the simulation of streamflow by the SRM as part of its evaluation.

In this manuscript the underlying principles presented from the CASE framework (based on observed-rainfall evaluation) are adopted and as extended to enable a systematic and comprehensive evaluation within the virtual hydrological evaluation framework (based on virtual-streamflow evaluation). The virtual hydrological evaluation framework also introduces two types of virtual experiments. None of these advances were part of the Bennett et al. (2018) paper. Hence, we believe this study represents a substantial advance on Bennett et al. (2018).

To help the reader understand this, clarification of the differences between the two frameworks has been added to the revised manuscript:

> Page 5, Line 1: '*The framework presented in this paper is a significant advance from previously reported virtual experiments because it presents a formal framework to identify key deficiencies in the SRM by (1) extending the comprehensive and systematic evaluation (CASE) framework (developed by Bennett et al., 2018 for observed-rainfall evaluation and used by Evin et al., 2018, Khedhaouiria et al., 2018) that systematically categorises performance at multiple spatial and temporal scales using quantitative criteria for each statistic for use in virtual hydrological evaluations, and (2) utilising two types of virtual experiments that are able to identify the source of key deficiencies in SRM at specific locations and time periods.*'

Additionally, the first paper objective has been re-worded to clarify that the virtual hydrological evaluation framework is a novelty but draws on the systematic evaluation and tests introduced by the CASE framework.

> Page 5, Line 8: '*1. To introduce a formalised framework for the virtual hydrological evaluation of SRMs: the new framework is a stepwise procedure that enables the identification poor performing sites, then*

*poor performing time periods and then the key deficiencies in the SRM for those sites and time periods by drawing on the systematic application of quantitative performance criteria.'*

We do not believe the reference to Bennett et al. (2018) was missing and is given on Page 29, Line 27 of the manuscript and is also reproduced below. This reference was a part of the original submission to HESS and has been included in all revisions.

Bennett, B., Thyer, M., Leonard, M., Lambert, M., and Bates, B. (2018). A comprehensive and systematic evaluation framework for a parsimonious daily rainfall field model, Journal of Hydrology, 556, 1123-1138.
* * *
**Comment 16:**

*P5l23-26 For the comparison of different rainfall data sets (observed vs. generated time series) the extraneous variables are kept. This introduces a new bias, since extraneous variables from a rainy day can occur with a dry day from the generated rainfall time series (and vice versa), while for the observed time series always a "perfect match" occurs. Depending on the used equation for the calculation of the potential evapotranspiration (not mentioned so far) for example very sunny days with high radiation can occur simultaneously with rainfall the whole day. This bias has to be quantified by a sensitivity study, because it is not related to the SRM generation itself. A possible solution would be to use a weather generator (to have consistent weather data set as input for the r-r model) instead of only a rainfall generator (but of course, that introduces biases as well from the other climate time series), but this is maybe beyond the scope of the study. A discussion on the new introduced bias and its quantification would solve this issue.*
* * *
**Response 16:**

This issue was already raised in the previous round of reviewer comments and has been addressed through additions in the discussion section. We have copied the text from the previous response below for convenience that provided additional explanation and outlines the changes made. Further, as the reviewer concedes we consider a more extended discussion to be out of scope for this study.

***Excerpt from Response to Reviewer # 2 Comment 7, Revision Round 1:***

**[***'The reviewer is correct, as the focus is on evaluating stochastic rainfall generators, the other forcings are supplied independently. In our case study, the potential evapotranspiration (PET) time series (our only other meteorological forcing) is unchanged from the observed values in all hydrological simulations (i.e. the same PET time series is used in the simulation of the virtual-observed streamflow, integrated tests and unit tests). This is important as the hydrological evaluation is a relative comparison of the observed and simulated rainfall, hence all other time series and parameters relating to the hydrological model are kept the same in all tests. This approach was also taken in Sikorska et al. (2018), where the impact of using different rainfall disaggregation schemes on resultant flow was tested using a hydrological model. For all these tests the historical observed temperature time series was used to enable a comparison between the rainfall elements only.***

*To assess this assumption for the Onkaparinga case study we have evaluated the rainfall-PET correlation in all months. There is a negative relationship, which accounts for a small portion of the variance, up to $R^2 = 0.11$ in drier summer months, shows the rainfall-PET correlations for a drier summer month (January) and a wet winter month (June).*

[Figure]

*Figure 2- Rainfall-PET correlation (left) January and (right) June.*

*While there is some non-zero relationship, we do not consider it to undermine the case study (since all other statistics of PET are reproduced and the relationship is mild). However, this may not be the case for other locations where the model is applied.*

*We have included the reviewer's recommendation to apply the framework more generally to stochastic weather generators in Section 5.3. The application would require care to ensure that the PET (or other weather variable) generator does not introduce other deficiencies.*

> *Page 27, Line 22 - Page 28, Line 19: "The formalisation of the virtual hydrological framework for SRM evaluation provides the opportunity for further improvements in the future, including: ...*
>
> *(iv) Evaluation of spatial performance – there are multiple opportunities to develop tests for spatial performance including (a) repeating the integrated test for all sites and for catchment average rainfall means it would be possible to diagnose whether specific locations or the spatial dependence causes poor reproduction of streamflow statistics, (b) developing a spatial unit test (which is analogous to the temporal unit test but extended to space) where different combinations of sites are 'spliced' in the construction of catchment average rainfall – to evaluate the impact of 'mixed' performance in the SRMs between sites on the catchment average rainfall, and (c) these spatial unit tests could be used to evaluate stochastic weather generators (SWG) more generally as well as spatially distributed SRGs – though these would require a spatially distributed hydrological model."* **] end of excerpt.**
* * *
**Comment 17:**

*P5l23-26 Also, one aim of the rainfall generation is to provide longer input time series than the existing observed ones. How can this be handled by the framework? For example, would a generated 600 year time series split into "30 realisations" because only 20 years of observations exist? How are completely unobserved catchments (no climate data) validated with this method. Some information on these issues should be provided in the manuscript.*
* * *
**Response 17:**

In this paper 10,000 replicates of 73 year timeseries generated by an SRM are evaluated. In this case the generated SRM timeseries is of equal length to the observed rainfall time series. As the comparisons made between the virtual-observed (or observed) timeseries are performed through the comparison of summary statistics it is possible to compare generated timeseries that are longer than the virtual-observed (or observed) in terms of how well the reproduce the statistics of interest. The evaluation of a longer generated timeseries against a shorter observed period is also reported in observed-rainfall evaluation approaches (Rasmussen, 2013).

The virtual hydrological evaluation framework is based on a relative comparison between the virtual-observed streamflow and the simulated streamflow–as both simulated and observed rainfall undergo transformation by the same process representation (i.e. the hydrological model). Therefore, in instances where the streamflow of a catchment is unobserved the comparison between virtual-observed and simulated streamflow is still possible for the purpose of identifying deficiencies in the SRM. In such cases it is recommended that the hydrological model is calibrated such that the necessary catchment streamflow features can be simulated (e.g. via calibration to a neighbouring catchment with similar catchment properties).
* * *
**Comment 18:**

*P6l7 Maybe the authors consider to replace "compare" by "combine"? If a comparison is the intention behind Fig. 2, I cannot understand how the comparison is carried out.*
* * *
**Response 18:**

Done. The sentence has been rephrased as: '*The formal implementation of the virtual hydrological evaluation framework is summarised in Fig.2 … It combines both observed rainfall-evaluation and virtual hydrological evaluation*.'
* * *
**Comment 19:**

*P6l14 The authors mention integrated tests here, but introduce them a few sections later. This is a bit confusing for the reader. Either the reader is referred to the subsection where the test is introduced or (what I would prefer) the test is introduced before mentioning it.*
* * *
**Response 19:**

A reference to the subsequent subsection where the test is introduced has been inserted (as this section simply provides a brief overview of Step 1).

> Page 6, Line 20: '*Following the selection of a primary streamflow characteristic of interest and a suitable hydrological model, integrated tests are conducted for each rainfall site (described below in Section 2.2.2).*'
* * *
**Comment 20:**

*P6l17 & Fig. 2 The CASE framework was mentioned only in the introduction (p4l28), the provided reference is not included in the reference list. (How) Does the CASE framework differ from the framework applied in this study? If the CASE framework is important for the reader to follow the investigation, it should be (briefly) explained in the current manuscript (the reader should not be forced to read other manuscripts to be able to follow the current investigation).*
* * *
**Response 20:**

Please also see the response to comment 15. At this point in the manuscript (introductory text setting out the elements of Step 1) the important factor is that the systematic application of quantitative performance criteria is used to identify poor performing sites. The text has been revised to read:

> Page 6, Line 20: '*The first step focuses on using integrated tests to identify poor performing sites for further evaluation. Following the selection of a primary streamflow characteristic of interest and a suitable hydrological model, integrated tests are conducted for each rainfall site (described below in Section 2.2.2). The results of the integrated tests are then used to identify sites that are poor performing, according to the systematic application of quantitative performance criteria (see Section 2.2.3), for the primary streamflow characteristic.*'

We do not believe the reference to Bennett et al. (2018) was missing. The reference appears on Page 29, Line 27 of the manuscript and is also reproduced below:

> Bennett, B., Thyer, M., Leonard, M., Lambert, M., and Bates, B. (2018). A comprehensive and systematic evaluation framework for a parsimonious daily rainfall field model, Journal of Hydrology, 556, 1123-1138.
* * *
**Comment 21:**

*P6l22 Is it also possible to take into account more than one primary streamflow characteristic?*
* * *
**Response 21:**

Yes, where a combination of streamflow features are important the 'primary streamflow characteristic' may take into account multiple factors. The purpose of the primary streamflow characteristic is to focus the investigation of SRM deficiencies on to those that impact on streamflow features that are important for the studied catchment.
* * *
**Comment 22:**

*P7l16 Again, the CASE framework should be explained before.*
* * *
**Response 22:**

Thank you. The CASE framework is now described earlier in the manuscript. Please also see responses to comments 15, 20 and 45.
* * *
**Comment 23:**

*Fig. 3 The location of the elements case (i-iii) is hard to identify since it remains unclear, which element (the mean or the range) determines the position on the y-axis. I suggest to put both elements on the same level in direction of y-axis instead of having two different elements. Also, does the range result from different months or from the 10,000 realisations of the SRM or from both? Would it be worth to distinguish between both ranges?*
* * *
**Response 23:** We have amended the Figure so that the mean and range are aligned. Thank you for pointing this out. The mean (indicated by the '+' symbol) should be on the same level as the y-axis (it was originally separated out so that they element could be seen better, but we agree this is confusing).

[Figure]

**Figure 3 - Illustration of performance classification, case (i) shows 'good' performance, case (ii) shows 'fair' performance and case (iii) shows 'poor' performance. Adapted from Bennett et al. (2018).**

Yes, the range is the result of the 10,000 realisations of the SRM. As the figure is statistic based, the statistics for the individual months can be separated out. For example, the performance of the SRM's simulation of mean flow for three separate months would look like Figure 3 where 3 ranges (made up of 10,000) are shown.
* * *
**Comment 24:**

*P7l20-22 How have the threshold of 90 % and 99.7 % been determined? Especially the latter one seems to have a certain origin and was not been chosen arbitrarily…The threshold should depend on the criterion used for the validation, right? So the threshold would be different for a derived flood with a 100 year return period in comparison to the mean discharge or some drought indices.*

**Response 24:**

In this manuscript the thresholds of 90% and 99.7% applied are taken from Bennett et al. (2018) with the 99.7% threshold being used to apply a threshold that is 3 standard deviations from the mean.

Yes, the thresholds should depend on the criterion used for the validation and can be adjusted as needed to suit the criterion. For example, a relative difference metric that provides an indication of the practical tolerance for the particular streamflow or rainfall feature could be adopted instead of a metric that is a function of the simulated range of statistics.
* * *
**Comment 25:**

*P8l5-17 How is snowfall handled in this context? If snow falls and accumulates over the winter period, it depends on the temperature time series when it starts to melt and to contribute to the total runoff. Again, in this context, how is the bias quantified? I can imagine that the differences are quite high depending on the interplay between rainfall and temperature.*

*For example, if the SRM generates "precipitation" and due to the temperature time series it falls as snow and accumulates over days/weeks until it contributes to runoff, it causes a high difference to the observed time series, which had no precipitation in that cold period. This is from my understanding not covered by the virtual hydrological framework, and has to be quantified as the bias mentioned before.*
* * *
**Response 25:**

The case study catchment (Onkaparinga catchment, South Australia) has a Mediterranean climate and an ephemeral streamflow response. It does not experience snowfall or any build-up of snow over the winter period. This a common catchment type for Australia (and in similar climates internationally) and represents different challenges to ones where there is snowfall. The reviewer raises a valid point regarding the impact of snow, but this is not applicable to this catchment. We look forward to addressing this issue in future research when we apply this framework to a catchment with snow.
* * *
**Comment 26:**

*P12l19-29 The description of the rainfall generator is not sufficient to understand it. For further details the reader is referred to a reference which is not provided in the reference list. This section has to be clear to the reader, since it is an essential basic for the evaluation of the introduced framework.*
* * *
**Response 26:**

Please see the two responses from the first round of reviewer comments below in which additional details on the rainfall generator have already been provided. We also do not believe that the reference was missing.

The focus of this manuscript is the virtual hydrological evaluation framework, which is independent of the rainfall generator tested (i.e. essentially any rainfall generator could be tested in its place).

Note that the rainfall generator is clearly explained in Bennett et al. (2018) which was provided in the reference list and applied to the same catchment. We do not think it is necessary to add extra length (to an already long paper) explaining a rainfall generator that is not part of the key innovations and is explained in an easily accessible paper.

**Excerpt from Response to Reviewer #3 Comment 5, Revision Round 1:**

> **[**'*In the revised manuscript a summary is provided of the calibration approach for the rainfall model (see Section 3) so that it is easier for the reader to understand the model without needing to also read Bennett et al (2018).*
>
> > *Page 12, Lines 19-32: "The simulated daily rainfall was determined from the latent variable autoregressive daily rainfall model of Bennett et al. (2018) using at-site calibrated parameters. This rainfall model uses a latent variable concept, which relies on sampling from a normally distributed 'hidden' variable. The latent variable can then be transformed to a rainfall amount*

*by truncating values below zero and by rescaling values above zero to match the observed rainfall's distribution. Here, the rainfall is rescaled using a power transformation.*

*To calibrate the model the rainfall data at a given site is partitioned on a monthly basis and separate parameters are fit for each month. The mean and standard deviation of rainfall amounts, as well as the proportion of dry days is calculated. These statistics are matched to the corresponding properties of the truncated power transformed normal distribution. The at-site lag-1 temporal correlation is then calculated based on the observed wet day periods for a given month. This statistic is transformed to the equivalent correlation of the underlying latent variable by accounting for the effects of truncation to determine the autocorrelation parameter. Full details of the calibration procedure are provided in Bennett et al. (2018).*

*In this study the daily rainfall model was calibrated and simulated at 22 locations throughout the catchment that have long, high-quality records (Table 3). 10,000 replicates of simulated rainfall covering a 73 year period (1914-1986) were used."]* **End of excerpt.**

**Excerpt from Response to Reviewer #3 Comment 4, Revision Round 1:**

**['***Other reviewers have made similar comments (reviewer #1 comments 14-17, reviewer # 2 comments 7 and 11). The following details have been included in the revised Section 3:*

- *Catchment area (323 km$^2$)*
- *Rainfall model resolution (daily)*
- *Hydrological model resolution (daily)*
- *Hydrological model calibration details such as the number of observed years (model calibration and selection: 1985-1999, model evaluation: 2000-2009), Thiessen weighting of rainfall gauges was used to calculate catchment average rainfall, and the impact of rainfall errors was considered in detail (see Westra et al. 2014a, Westra et al. 2014b).*
- *The same set of hydrological model parameters are used for the unit and integrated tests so that the same transformation of rainfall to flow is used.*

*Section 3 is reproduced below for convenience.*

*Page 12, Line 10 – Page 13, Line 12: "The Onkaparinga catchment in South Australia is used as a case study (Fig. 5). The 323 km$^2$ catchment lies 25 km south of the Adelaide metropolitan area and contains the largest reservoir in the Adelaide Hills supplying the region (Mount Bold Reservoir). The catchment has a strong seasonal cycle (shown in Fig. 6) where the driest months (December, January and February) exhibit low rainfall and low streamflow, the wettest months (July, August and September) have high rainfall and high streamflow and the 'wetting-up' period (April, May and June) has high rainfall and lower streamflow. There is a strong rainfall gradient (Table 3), with average annual rainfall ranging from approximately 500 mm on the coast (Site No. 19) to over 1000 mm in the region of highest elevations (Site No. 20). A breakdown of the rainfall characteristics (annual total, number of wet days, daily average amounts, wet-spell and dry spell durations) at each site on a monthly basis is provided in Supplementary Material A.*

*The simulated daily rainfall was determined from the latent variable autoregressive daily rainfall model of Bennett et al. (2018) using at-site calibrated parameters. This rainfall model uses a latent variable concept, which relies on sampling from a normally distributed 'hidden' variable. The latent variable can then be transformed to a rainfall amount by truncating values below zero and by rescaling values above zero to match the observed rainfall's distribution. Here, the rainfall is rescaled using a power transformation.*

*To calibrate the model the rainfall data at a given site is partitioned on a monthly basis and separate parameters are fit for each month. The mean and standard deviation of rainfall amounts, as well as the proportion of dry days is calculated. These statistics are matched to the corresponding properties of the truncated power transformed normal distribution. The at-site lag-1 temporal correlation is then calculated based on the observed wet day periods for a given month. This statistic is transformed to the equivalent correlation of the underlying latent variable by accounting*

> *for the effects of truncation to determine the autocorrelation parameter. Full details of the calibration procedure are provided in Bennett et al. (2018).*
>
> *In this study the daily rainfall model was calibrated and simulated at 22 locations throughout the catchment that have long, high-quality records (Table 3). 10,000 replicates of simulated rainfall covering a 73 year period (1914-1986) were used...")* **] End of excerpt.**
* * *
**Comment 27:**

*P12l30-31 Regarding the parameter estimation: Parameters are estimated from 1914-1986 and have almost no overlay with the calibration period of the hydrological model. Which time series period of the other climate variables is used for the simulations, the period of the calibration period of the r-r model or the period which was also used for the estimation of the SRM parameters?*
* * *
**Response 27:**

Yes, the SRM model parameters are estimated on the 1914-1986 which does not have a large overlay with the hydrological model calibration period and associated climate variable time series (in this case PET). While these differences would present an impediment for observed streamflow evaluation they are not critical for the use of a virtual hydrological evaluation which uses a relative comparison of simulated and virtual-observed streamflow to interrogate the simulated rainfall. In a virtual hydrological evaluation the same hydrological model parameters are used in the production of the virtual-observed streamflow and simulated streamflow. Likewise, the same related climate variables (in this case PET) are used in all simulations. Therefore, the relative comparison between the virtual-observed and simulated streamflow are not impacted by the different calibration period.
* * *
**Comment 28:**

*P13l2-4 Was the calibration carried out for the same period and with the same station density as in this study? If not, does it has an impact on the simulated runoff (e.g. if one station was missing in the calibration period, which has a high influence (weight) on the areal rainfall, but the rainfall generation for that station is different due to its altitude or snow/rainfall occurrences)?*
* * *
**Response 28:**

Similarly to the response to comment 27, while these differences are not as important when using a virtual hydrological evaluation approach which uses a relative comparison of the simulated and virtual-observed streamflow to interrogate the simulated rainfall, as the same hydrological model parameters are used in the production of both the virtual-observed streamflow and simulated streamflow.

In terms of the issue surrounding station density, in this paper we have not evaluated how simulated catchment average rainfall performs in terms of the resultant streamflow. This is because, as a matter of first priority, our approach focuses on identifying issues with rainfall at each site and getting this right before moving on to assess deficiencies in spatial properties. We therefore prefer to assess the at-site performance prior to the catchment average performance. Future work will demonstrate and apply the framework to catchment average rainfall.

**Comment 29:**

*Fig. 5: 7 out of 22 rain gauges have no influence on areal rainfall using Thiessen polygons: station 21, 13, 19, 8, 2, 10, 8. In total, 3 out of them are explored in detail due to "the relatively poorer ability of simulated rainfall to reproduce annual streamflow totals at these sites". How can these points have a negative impact on the simulated runoff if they do not contribute at all to the areal rainfall (if Thiessen polygons are applied)? Or did I miss a certain point in the description of the areal rainfall determination? Is the areal rainfall for the framework analysis estimated only by one (or less than all 22 stations)?*

**Response 29:**

Thiessen polygons are not applied for the evaluation represented in Fig. 5, which summarises the outcome of Step 1 of the virtual hydrological framework procedure. In Step 1 of the proposed framework, rainfall from each site is applied independently and repeated for each site, so that any discrepancy between virtual-observed streamflow and simulated streamflow can be attributed to a deficiency in rainfall parameters at that site. Also see the response to comment 34.

**Comment 30:**

*P13l5-6 To which streamflow data were the model parameters calibrated? Mean flow (on a seasonal basis), annual extreme flows or something different?*

**Response 30:**

The rainfall-runoff model was calibration to daily streamflow time series. The text has been amended as follows:

> Page 13, Line 4: '*The GR4J model was calibrated according to the procedure set out in Westra et al. (2014b) for the stationary version of the GR4J hydrological model. The details are provided in (Westra et al., 2014a) and a short summary is provided here. The multi-site rainfall gauges were Thiessen weighted to calculate the catchment average rainfall. The hydrological model was calibrated to the daily streamflow data at Houlgrave Weir (see Fig. 5) using model calibration period of 15 years (1985-1999). The model parameters were estimated using maximum likelihood estimation procedure with a weighted least squares likelihood function. The set of hydrological model parameters that maximised the likelihood function were found using a multi-start quasi-Newton optimisation procedure with 100 random starts. Overall, the GR4J model was able to simulate streamflow with a good fit to the observed daily streamflow, with a Nash-Sutcliffe efficiency of 0.8.*'

**Comment 31:**

*P13l11 "The same set" – For all further analysis all 100 parameter sets are kept or only the best one?*

**Response 31:**

The hydrological model parameters (i.e. the set of parameters) that maximised the likelihood function were used for both the unit and integrated test. This has been clarified in the manuscript:

> Page 13, Line 8: '*The model parameters were estimated using maximum likelihood estimation procedure with a weighted least squares likelihood function. The set of hydrological model parameters that maximised the likelihood function were found using a multi-start quasi-Newton optimisation procedure with 100 random starts ... The same set of hydrological model parameters are used for both the unit and integrated tests so that the same transformation of rainfall to flow is used*.'

**Comment 32:**

*P13l Providing a NSC-value is only useful, if the authors state for which variable it was calculated. I assume for simulated daily discharge values?*

**Response 32:**

Yes, the NSE value was calculated based on comparing observed and simulated daily streamflow. This detail has been added to the paper.

> Page 13, Line 11: '*Overall, the GR4J model was able to simulate streamflow with a good fit to the observed daily streamflow, with a Nash-Sutcliffe efficiency of 0.8.*'

**Comment 33:**

*Table 3 The abbreviations used in the table header include event-based characteristics. How do the authors define an event (when does it start and end)? This information is important for the reader to understand the characteristics.*

**Response 33:**

A note has been added to Table 3 to indicate that a wet day is defined as a day with rainfall above 0.1 mm such that the number of days in a row above this threshold are deemed a wet-spell (and vice versa for dry-spells). The note is produced below:

> Page 14, Bottom of Table 3: '*Note: Wet days are defined as days where the rainfall exceeded a 0.1 mm threshold with wet-spells defined as the number of days in a row above the threshold (and vice versa for dry-spells).*'

**Comment 34:**

*P15l6 I can't follow how 10 out of 22 rain gauges are categorized as "poor", if the areal mean is used as input for r-r modelling. Or are the r-r models driven by single-station input only?*

**Response 34:**

Yes, the hydrological model is driven by single-station input only. Also see response to comment 29.

**Comment 35:**

*P15l9-10 I would prefer to see the results for all 22 sites to test the framework more in detail.*

**Response 35:**

The full rainfall-based evaluation is provided in Bennett et al. (2018). The focus of the virtual hydrological evaluation framework (and in particular Steps 2-3) is to identify sources of error in streamflow-based evaluation. Therefore it logically follows that we would choose to apply this to sites where there are streamflow errors. This is why the analysis focuses on the 10 sites with poor streamflow performance. If we had undertaken the subsequent analysis on the other 12 sites, it is unlikely to result in additional knowledge or the identification of actionable changes to the SRM because it performs well in terms of both rainfall and resultant streamflow. This analysis would have taken additional space in an already long manuscript (13 Figures, 3 Tables, ~10,300 words).

**Comment 36:**

*P15l19-27 I struggle with the description of Fig. 7. The first column includes simulated rainfall statistics: mean of daily rainfall amounts and mean number of wet days, while in the second one the standard deviation for both is shown. How is in the first column for one sites decided, if it is a good, fair or poor quality, when two criteria are taken into account? Do both criteria for one site have to be "good", to result in a final "good?" Or is the mean of both relative errors chosen for the final decision? A short explanation would be helpful.*

**Response 36:**

In Figure 7 each quantitative criteria is applied separately such that a site may be deemed 'good' in terms of the simulation of daily rainfall means but 'poor' in terms of the standard deviation. However, in the first column it shows the performance for the sites for three separate statistics (mean daily rainfall amounts, the standard deviation of daily rainfall amounts and the mean number of wet day) as all of these statistics are classified as 'good' for 100% of sites for all the months (and at the annual level). This was done to avoid the repetition of three columns of 100% 'good' bar-charts.

Descriptive text explaining this has been added to the manuscript to avoid confusion:

> Page 15, Line 19: '*The first to fourth columns of Fig. 7 summarise the observed-rainfall evaluation and the fifth and sixth of Fig. 7 summarise the virtual hydrological evaluation. The first column of Fig. 7 indicates that of the poor performing sites the SRM exhibited 'good' performance in simulating daily rainfall means and standard deviations as well as the mean number of wet days for all sites and months and at an annual level according to the observed-rainfall evaluation. Each of the three statistics presented in the first column are assessed separately but are presented together to avoid repetition.*'

**Comment 37:**

*P16l1-3 The small deviation from the observed runoff in January result from the very low runoff generated in January (see Fig. 6), that the r-r model is trained to simulate-the rainfall has no effect in that months. It would be useful if the authors provide an example from the wettest periods (as done later).*

**Response 37:**

An example of the type requested (i.e. discussion of wetter months) appears in the subsequent paragraph.

**Comment 38:**

*Fig. 7 caption: Since the rainfall and runoff characteristics are mentioned before (p15l15-18), they can be removed from the caption.*

**Response 38:**

Fixed. The previously described rainfall and runoff characteristics have been removed from the caption.

**Comment 39:**

*Fig. 8 Why is for the standard deviation (b) and d)) no range? There should be a range from the number of months used for the simulation (15 years) and from the 100 parameter sets mentioned before.*

**Comment 40:**

*Fig. 9 Again, shouldn't there be a range for the virtual observations as well?*

**Responses 39–40:**

As mentioned in the response to comment 31, only the 'best' set of hydrological model parameters (i.e. those that maximise the likelihood function) are used in the simulation of virtual-observed and simulated streamflow.

The red crosses in Figure 8 (b) and (d) represent the standard deviations of the observed rainfall and the virtual-observed flow for each month calculated from the 73 years. The standard deviation represents the variability. They are single numbers. If we were presenting all the monthly values, then they would be a range, but we are presenting statistics, the mean (left-hand side) and standard deviation (right-hand side) for each month of the

year. The boxplots show the range of these monthly statistics for the 10,000 stochastic rainfall model replicates. This is a standard way of evaluating stochastic models (e.g. Bennett et al. 2018, Khedhaouiria et al. 2018, Evin et al. 2018, Frost et al. 2011, Frost et al. 2004, Srikanthan et al. 2004, etc.).

This issue regarding the range has also been dealt with in the first round of revisions. See text below.

**Excerpt from Response to Reviewer #2 Comment 6, Revision Round 1:**

**[**'*… In Figures 6 and 7 (now Figures 8 and 11) we present a higher-level summary to evaluate the model performance considering both mean conditions and their variability (i.e. standard deviations). The statistics presented in these figures are the observed rain and the virtual-observed flow means (left column) and standard deviations (right column) over the full 73 years, calculated for the 12 months respectively. There are 12 monthly means and 12 monthly standard deviations per realisation. We are not calculating a separate statistic for each year of the timeseries. The boxplots show the range of these monthly statistics for the 10,000 stochastic rainfall model replicates. This convention is common to other papers in the field (e.g. Bennett et al. 2018, Khedhaouiria et al. 2018, Evin et al. 2018, Frost et al. 2011, Frost et al. 2004, Srikanthan et al. 2004, etc.)*'**] End of excerpt.**
* * *
**Comment 41:**

*P22l20 Do the 10 % refer to the 10 bad out of 22 stations or to all 22 stations?*
* * *
**Response 41:**

Thank you for pointing this out. It refers to the set of 'poor' sites/months (identified in Step 2) in terms of the attributed cause. Clarification has been added.

> Page 22, Line 19: '*The category where streamflow errors originate from rainfall model deficiencies in a preceding month represents about 10% of the evaluated site/month combinations (i.e. those identified in Step 2).*'
* * *
**Comment 42:**

*P26l11 I'm not familiar with the reference of Ang & Tang (2007) and the information provided in the reference list are insufficient to find it (is it a book? v1 sounds like a handbook or a model description). However, the r-r transformation is strongly non-linear, hence an error propagation as mentioned in the text should not be carried out (even not as an example).*
* * *
**Response 42:**

Yes, Ang & Tang (2007) is a textbook commonly used as reference text to teach statistics to engineering students. The reference list has been updated to make this book easier to identify and find. The reference supplied is as follows:

> ANG, A. & TANG, W. 2007. Probability Concepts in Engineering: Emphasis on Applications to Civil and Environmental Engineering (2nd Edition), Hoboken, New Jersey, USA, John Wiley & Sons Inc., hardback ISBN-10 0-471-72064-X.

While we agree with the reviewer, that the rainfall-runoff transformation is non-linear, this does not invalidate this discussion point. The example provided is used to demonstrate the potential amplification of the errors and is likely to be a lower bound, because it assumes linearity in the transformation to perform the error propagation. The revised text now highlights this point.  This discussion point simply serves as an illustration of the advantages of the using streamflow-based evaluation approaches. The analysis of error propagation is not a key point of the paper.

> Page 26, Line 9: '*In terms of amplification, the elasticity of the rainfall-streamflow response (Chiew, 2006) suggests that catchments can have strong sensitivities to discrepancies in rainfall. Given that the rainfall elasticity of streamflow to rainfall is a factor of 2 to 3.5 (Chiew, 2006), using the principles of error propagation (Ang and Tang, 2007), assuming linearity it follows that a 10% error in mean/standard deviation of rainfall could potentially be amplified to 20-35% error in the mean/standard of streamflow. This estimate represents a lower-bound of the potential amplification,*

*since the non-linear nature of the rainfall-runoff transformation will likely produce a larger potential amplification of errors. This indicates that streamflow-based evaluation of rainfall models provides a stronger test than observed-rainfall evaluation in terms of the sensitivity of the statistics.'*
* * *
**Comment 43:**

*P28l21-22 This sentence sounds as if it was possible to determine, which rainfall characteristics are most important for the r-r modelling. Indeed, this was not done in the study. A few rainfall characteristics have been analysed, but there was no detailed and quantitative investigation which one has higher impacts on the simulated runoff.*
* * *
**Response 43:**

Fixed. The sentence has been reworded to remove the implication that it determines which rainfall characteristics should be modified.

> Page 28, Line 24: '*This paper has introduced a virtual hydrologic evaluation framework that enables targeted hydrological evaluation of SRMs*.'
* * *
**Comment 44:**

*P29l1 It would be useful if the authors can provide references for conventional approaches.*
* * *
**Response 44:**

We agree, conventional approaches were not well defined and have removed the term. The wording of this part of the conclusion, has been modified to be more direct and consistent with the discussion in Section 5.2.

> Page 29, Line 2: '*The unit test identified the importance of the simulated rainfall in the transition months of May and June (late autumn/early winter) during the 'wetting-up' phase of the catchment cycle for producing low errors in subsequent high streamflow months (July/August/September) and the annual streamflow distribution. The virtual hydrologic evaluation framework provides valuable additional diagnostic ability for the development and application of SRMs, not available by using rainfall-based evaluation techniques alone.'*
* * *
*Technical corrections:*

**Comment 45:**

*P2l9 The reference of Bennett et al. (2018) is missing and could not be proven regarding its content and relevance for the current manuscript.*
* * *
**Response 45:**

We do not believe that the reference was missing, it could be found on page 40, line 1 of the marked-up version of the manuscript provided as well as page 29, line 24 of the un-annotated version as part of the first round of manuscript revisions. Furthermore, this reference was a part of the original submission to HESS and has been consistently included in all revisions.

The reference Bennett et al. (2018) is provided on Page 29, Line 27 of the revised manuscript and is also reproduced below for convenience:

> Bennett, B., Thyer, M., Leonard, M., Lambert, M., and Bates, B. (2018). A comprehensive and systematic evaluation framework for a parsimonious daily rainfall field model, Journal of Hydrology, 556, 1123-1138.
* * *
**Comment 46:**

*Fig. 9, 10, 13 "cummulative" -> "cumulative"*
* * *
**Response 46:**

Fixed. The x-axis labels in panel (c) of Figures 9, 10, 12 and 13 have been amended.

**Comment 47:**

*P13l6 Fig. 5 instead of Fig. 4? But I can't see the Houlgrave weir in Fig. 5 anyway.*

**Response 47:**

The figure cross-reference has been fixed. Figure 5 has been amended so that Houlgrave Weir is explicitly labelled to make it easier to locate on the map. The streamflow gauge position is indicated by the purple square. The figure is reproduced below:

[Figure]

**Figure 4 - Onkaparinga catchment, South Australia. Sites indicated by blue triangles are explored in greater detail in this paper due to the relatively poorer ability of simulated rainfall to reproduce annual streamflow totals at these sites.**

**Comment 48:**

*P24l6 "sits" is an inappropriate verb in this context*

**Response 48:**

The verb 'sits' has been removed and replaced with 'is located'.

**Comment 49:**

*P27l8 "identifying" -> "identifies"*

**Response 49:**

Fixed.

**Comment 50:**

*P27l13 Remove the comma.*

**Response 50:**

Fixed.

**Comment 51:**

*P27l25 "a hydrological model" -> "a single hydrological model" Maybe this is what the authors want to say?*

**Response 51:**

Done – 'a single hydrological model' has been inserted in place of 'a hydrological model'.

**Comment 52:**

*Comments from the first review:*

*Reply to comment 2: "This is the first time the virtual-observed streamflow evaluation approach has been formalised using a Comprehensive and Systematic Evaluation (CASE) framework (pioneered by Bennett et al., 2018 and used by Evin et al. 2018, Khedhaouiria et al. 2018) to evaluate stochastic rainfall models in terms of the ability to produce key runoff statistics of interest"*

*From the manuscript it remains unclear what the CASE framework is exactly (is it the applied framework illustrated in Fig. 2?), the reference for explanations (Bennett et al., 2018) is missing.*

*„(iii) systematically categorise aggregate performance over multiple spatial and/or temporal scales"*

*-> Only one spatial and temporal scale are analyzed.*

**Response 52:**

Clarification surrounding the CASE framework as it pertains to the virtual hydrological evaluation framework has been added to the manuscript (see Sections 1 and 2.2; also see the responses to comments 15, 20, 22 and 45) as well as clarification around the spatial and temporal scales addressed (see response to comment 13).

[revised manuscript text omitted]

(a) Observed-rainfall evaluation

| True rainfall | Observed rainfall | Observed-rainfall evaluation | Stochastic rainfall model / Simulated rainfall |

(b) Observed-streamflow evaluation

True rainfall → Catchment processes → True streamflow | Observed streamflow | Observed-streamflow evaluation | Stochastic rainfall model → Simulated rainfall → Hydrologic model → Simulated streamflow

(c) Virtual hydrological evaluation

True rainfall → Observed rainfall → Hydrologic model → Virtual-observed streamflow | Virtual hydrological evaluation | Stochastic rainfall model → Simulated rainfall → Hydrologic model → Simulated streamflow

Key: Observed · True · Simulated · Model

[revised manuscript text omitted]

---

## Author Response (AR3)

**Response to the Editor**

*Comments to the Author:*

*Dear Bree Bennett and co-authors,*

*The referees are satisfied with the revised version of the manuscript. Reading it once again, I have a few comments and suggestions, listed below, that I feel might contribute to the message of the paper. I hope this will be the last round of revisions before accepting the manuscript for publication.*

*Best regards,*

*Nadav*

This document describes our response to the referee comments and our revision of the paper entitled 'A virtual hydrological framework for evaluation of stochastic rainfall models' (HESS 2018-489).

We are pleased that the reviewers are satisfied with the revision and response to queries.

We agreed with the majority of the issues raised by the editor. The key modifications we have made in the revised paper include:

- Added a new section to the discussion (Section 5.3) that discusses identified deficiencies in the SRM with reference to the corresponding rainfall properties (Comment #1)
- Revised the paper objectives to clarify the space and time scales evaluated (Comment #2)
- Clarified the term 'site' throughout the paper (Comment #3)
- Revised Figure 5 to include the rainfall gradient (Comment # 5)

We were somewhat surprised to see comments in this third review that, by our impression, have re-raised issues otherwise satisfactorily addressed in the responses to the first review (further details given below in response to Comment #1). We appreciate that the reviewers have put in substantial effort providing thorough and thoughtful reviews. They have greatly improved the quality of this paper. We have responded by putting in substantial effort to address reviewer concerns, and in each of the three revisions of this paper the reviewers appear to be happy with our responses (e.g. Reviewer #1 '*The authors addressed all my comments and answered all my questions in a sufficient way').*  We look forward to obtaining a decision soon regarding this paper.

Our item-by-item responses to the specific comments appear below.

**Comment 1:**

*Therefore, to improve September flows, September rainfall should be improved in preference to all other months" [18 7-8] and again "therefore, to improve July flows, it is not just the July rainfall that should be improved but also the preceding two months" [20 9-10]. The fact that only the problematic months are detected and that no information is given on the properties that need to be corrected reduces the importance of this study. As it is later mentioned "this framework can be extended to provide more direct guidance on which rainfall features (in terms of components of the SRM) should be modified to improve streamflow performance. This can be done by comparing multiple rainfall model variants (parametrically, or via bootstrap techniques) which are designed to have contrasting features of a key characteristic (e.g. intermittency, rainfall correlation)" [28 4-6]. I strongly recommend that this will be demonstrated (using bootstrapping method or alike) already in this paper on one of the cases that are discussed, as I think it will emphasize the importance of using the suggested tests. This will make the methodology presented in this paper practical and more useful.*

**Response 1:**

We agree with the editor that identifying the key rainfall properties that should be corrected will improve the paper. In order to achieve this we have now added a new section in the discussion (Section 5.3. Identifying the key deficiencies in the rainfall properties of the SRM) where we relate the insights from the virtual hydrological evaluation to the observed-rainfall evaluation and highlight that it is the variability in the rainfall occurrences in two months (May and June) that will likely provide important improvements for reproducing the virtual streamflow.

> Page 28, Line 23: '*The previous section highlighted that simulated rainfall in the 'wetting up' period May-June was a key cause of errors in the streamflow. Returning to the observed-rainfall evaluation of the SRM (see Fig. 7) enables identification of the rainfall properties that are likely to be the cause of these errors in the streamflow. In the May-June months, the mean and standard deviation of rainfall amounts on wet days and the mean number of wet days shows 100% 'good' performance, while the standard deviation of the number of wet days shows a high proportion of 'poor' performance (50% in May and 40% in June). This identifies that rainfall amounts on wet days are reproduced well, but there is not enough variability in the rainfall occurrences in the key months of May-June. The observed-rainfall evaluation identifies this is a common problem for this SRM with eight months having some proportion of 'poor' sites for the standard deviation of the number of wet days–similar results were found in Bennett et al. (2018). However, the virtual hydrological evaluation identified that it is the rainfall occurrences in two key months (May and June) that will provide the most benefit in terms of virtual hydrological evaluation. This provides clear guidance on the rainfall properties that need to be improved for this SRM.*'

We agree with the editor that including multiple rainfall model variants would make the methodology in the paper practical and more useful. However, we do not think it is feasible to include another SRM or variant in this current study due to limited space. Over four different submissions, in response to the numerous issues raised by the reviewers we have included a large amount of additional information into the paper (i.e. analysis, figures and discussion) regarding the explanation of the virtual hydrological evaluation framework, the integrated tests and the innovative unit tests. As a result this paper has improved substantially and grown from being ~8,480 words, 7 Figures and 2 Tables in the original submission to now being quite lengthy (current paper revision ~10,400 words, 13 Figures and 2 Tables). Including another SRM variant would essentially be a fourth objective, require substantial analysis and further increase the text and figures. Based on our experience with introducing SRMs (e.g. Bennett et al., 2018), it would likely take another 2,800 words to explain the theory and model calibration techniques and another 3-4 Figures to present the results. Hence, to present this type of analysis in a meaningful way, it is better suited to dedicated separate future paper.

A similar comment was raised in the first review by Reviewer #2, who commented that '*Generally, I think that the demonstration would be more illuminating if the authors used it to compare two or more SRGs…*' and ' *It is useful to know whether the SRG performs well for some months than others, but what next?*' (as part of Comments 3 and 8). The responses are included below for completeness. We note that in the second review

this issue was not raised again by the reviewers or the editor and hence this gave the impression that the issue was resolved. We are somewhat surprised that this issue has be re-raised in the third review.

**The response to Comment 3 by Reviewer #2 from the first review is reproduced below for convenience:**

*["We support this idea. In the future, the framework can be used to compare two or more SRG's for particular hydrological applications. Furthermore, by utilising two or more hydrological models in the virtual evaluation framework, it would reduce the dependence on the choice of hydrological model (which was raised by reviewer #1 see comment 10), because one could look for patterns of errors for a single SRG across two hydrological models.*

*However, our preference is not to include this in this paper, for the following reasons:*

*1.        To include the details of a second SRG and/or a second hydrological model, as well as providing a complete explanation of the details of the framework, observed-rainfall evaluation, virtual observed streamflow evaluation, two different tests, the integrated and unit tests, would make the paper and/or analysis overly long. Reviewer #1 has asked for extra details and additional figures/tables to explain a wide range of details – including the hydrological model calibration (see reviewer #1 comment 15), extra streamflow analysis (see reviewer #1 comment 3). This is over and above the seven figures already included. If we included another SRG and hydrological model, the number of the figures could increase dramatically making the paper overly long and lose focus on the presentation of the framework.*

*2.        The framework has not been presented before, in particular the unit test. Therefore, in the paper we present the evaluation of a single stochastic rainfall model to demonstrate the framework. This application to a single model has demonstrated some new insights; that the errors in streamflow for a particular month can be affected by errors in the in rainfall from the previous 2 to 3 months. This innovation is something that has not been previously identified in the literature.*

*Once the framework has been established and explained in this paper, future work will undertake multiple SRG and/or multi-hydrological model comparisons, as suggested by the reviewer. A comment on this has been incorporated into the discussion (Section 5).*

*Page 27, Line 22 - Page 29, Line 7: "The formalisation of the virtual hydrological framework for SRM evaluation provides the opportunity for further improvements in the future, including:*

*(i) Using multiple, well-tested hydrological models - a potential limitation of the virtual hydrologic evaluation framework is that it is reliant on the use of a hydrological model. Hydrological structural errors may potentially skew interpretation of the SRM evaluation if the hydrological model poorly represents the catchment processes. To reduce these impacts the steps taken in this study included (a) using a well-tested hydrological model that has demonstrated good performance on a wide range of catchments (e.g. the GR4J model has been widely tested , see (Perrin et al., 2003, Coron et al., 2012); (b) calibrating and evaluating the hydrological model on a catchment close to the observed rainfall sites to ensure it provided sufficiently good performance  (e.g. GR4J was calibrated to the Onkaparinga catchment - see (Westra et al., 2014a, Westra et al., 2014b). Future research will use multiple, well-tested hydrological models with sufficiently good performance to reduce the reliance on a single hydrological model and ensure the identification of SRM deficiencies is not dependent on a single hydrological model.*

*(ii) Comparison of SRMs – this framework can be extended to provide more direct guidance on which rainfall features (in terms of components of the SRM) should be modified to improve streamflow performance. This can be done by comparing multiple rainfall model variants (parametrically, or via bootstrap techniques) which are designed to have contrasting features of a key characteristic (e.g. intermittency, rainfall correlation). Such an approach was undertaken by Evin et al. (2018) using an observed-rainfall evaluation approach. If the SRMs have monthly/seasonal autocorrelation (these were not significant for the rainfall in the Onkaparinga catchment) the unit testing approach would need to be extended by conditionally sample the simulated rainfall in a manner that preserves monthly correlations." ]end of excerpt*

**The response to Comment 8 by Reviewer #2 from the first review is reproduced below for convenience:**

*[*"It is useful to know whether the SRG performs well for some months than others, but what next?" – We agree that it is a useful feature and it provides much more information than observed-rainfall evaluation alone. This is a key innovation of the paper. Following an observed-rainfall evaluation the focus would have been on the months Jan, Feb, Nov, Dec, May and June (Bennett et al., 2018). However, based on the results of the virtual hydrologic framework, we now know that May-July are the key months when considering the hydrology and that the problems with modelled rainfall in Jan-Feb, Nov-Dec are less important. Also, we now know that rainfall in preceding months is important and not just the month in which the flow is evaluated, which is more information than before.*

*We agree that it does not tell us exactly which rainfall characteristics to focus on. However, it is unlikely to be that simple – a single rainfall statistic is unlikely to translate into a single runoff statistic because streamflow integrates a range of rainfall processes (see also reviewer #1, comment 21).*

*Now that we know in which months deficiencies originate, we can focus on those months and trial various alternatives to the rainfall model to address the problem. This is left for future research, as mentioned in Section 5.3.*

*Page 27, Line 22 - Page 28, Line 7: "The formalisation of the virtual hydrological framework for SRM evaluation provides the opportunity for further improvements in the future, including: …*

*(ii) Comparison of SRMs – this framework can be extended to provide more direct guidance on which rainfall features (in terms of components of the SRM) should be modified to improve streamflow performance. This can be done by comparing multiple rainfall model variants (parametrically, or via bootstrap techniques) which are designed to have contrasting features of a key characteristic (e.g. intermittency, rainfall correlation). Such an approach was undertaken by Evin et al. (2018) using an observed-rainfall evaluation approach. If the SRMs have monthly/seasonal autocorrelation (these were not significant for the rainfall in the Onkaparinga catchment) the unit testing approach would need to be extended by conditionally sample the simulated rainfall in a manner that preserves monthly correlations." ] end of excerpt*
* * *
**Comment 2:**

***Introduction - the scales of rainfall that are examined in space (single-site, multi-site, gridded rainfall) and time (daily, sub-daily) should be explicitly mentioned in the objectives of the paper. The fact that the model can be used in the context of gridded rainfall, for example, is mentioned too late, almost at the end of the discussion.***
* * *
**Response 2:**

We have modified the third objective to clarify the space and time scales (i.e. daily, multi-site) that are evaluated in this paper. The modified objective is reproduced below.

> Page 5, Line 15: '*3. To demonstrate the framework on a daily SRM at multiple sites to evaluate performance at daily, monthly and annual time scales and to contrast the outcomes with conventional evaluation methods*.'

Our preference is to keep the objectives focussed on the analysis that was undertaken in the paper and leave future extensions and applications of the model and/or framework at different space and time scales (e.g. spatial rainfall, sub-daily rainfall) for discussion (see Section 5.4).
* * *
**Comment 3:**

*The term "site" should be better defined. From reading the text, it seems that sometimes "site" refers to a catchment or sub-catchment and sometimes it refers to a point in space (e.g. a rain-gauge).*
* * *
**Response 3:**

Thank you for pointing this out. In the paper the term 'site' refers to a point in space (e.g. a rain gauge). We have clarified the meaning of the term 'site' early in the revised paper (in Section 2, paragraph 4):

> Page 6, Line 14: '*The formal implementation of the virtual hydrological evaluation framework is summarised in Fig.2. It uses a series of steps to identify poor performing sites (i.e. specific locations in space, for example, the location of a rainfall gauge), then poor performing time periods and then the key deficiencies in the SRM for those sites and time periods*.'
* * *
**Comment 4:**

*[7 12] Why daily? The integrated test can be applied also to, e.g. hourly scales, am I right? I suggest that in the methods section the scales should be discussed in general, and in the case study presented later the specific scales that were examined should be specified. Same for monthly in [8 9], for example.*
* * *
**Response 4:**

Yes, the integrated test can be applied at a range of scales (e.g. hourly, daily, monthly, annual). Here a daily time step was used to explain the integrated test, because it was used to test a daily SRM. In a similar vein as our response to Comment #2, our preference is to keep this paper focussed on the analysis undertaken in the paper, rather than potential future generalisations. Generalisations and extensions of the evaluation framework (to other time scales) can be undertaken when the framework is applied to a SRM at the particular time scale as part of future work (see discussion in Section 5.4). We now also mention in Section 2.2.2, that this evaluation framework can be extended to other time and space scales.

> Page 7, Line 23: '*The integrated test procedure is explained in terms of daily rainfall and daily streamflow at a single site, as it is herein applied to evaluate a daily rainfall model (see Section 3). This test procedure can be extended to other time and spatial scales (e.g. spatial rainfall, subdaily rainfall). These extensions are discussed in Section 5.4.*'

**Comment 5:**

*[12 16-17] The rainfall gradient should be presented in Figure 5. Table 3 can then be moved to the SI.*

**Response 5:**

The rainfall gradient has been added to Figure 5 (denoted using a colour ramp) and the contents of Table 3 have been moved to Supplementary Material A. The revised Figure 5 is reproduced below for convenience.

[Figure]

**Figure 5 - Onkaparinga catchment, South Australia. Sites indicated by triangles are explored in greater detail in this paper due to the relatively poorer ability of simulated rainfall to reproduce virtual-observed annual streamflow totals at these sites.**

**Comment 6:**

*[12 21-25] The rainfall is simulated independently at each site, or is the model simulates the rainfall at multi-site simultaneously, i.e. accounting for the cross-correlation between locations?*

**Response 6:**

Yes, the rainfall is simulated independently at each site. This has been clarified in the text:

> Page 13, Line 19: '*The evaluated daily rainfall was simulated using the latent variable autoregressive daily rainfall model of Bennett et al. (2018) using at-site calibrated parameters to simulate daily rainfall independently at each of the 22 sites … Full details of the calibration procedure are provided in Bennett et al. (2018). In this study the daily rainfall model was calibrated and simulated at 22 locations independently throughout the catchment that have long, high-quality records (Fig. 5). 10,000 replicates of simulated rainfall covering a 73 year period (1914-1986) were used.*'

We have also emphasised this point in the introduction:

> Page 5, Line 18: '*Daily SRMs have been developed for 22 sites in the Onkaparinga catchment, South Australia (Section 3), and are used to illustrate the procedure (Section 4).*'

**Comment 7:**

*[13 6-7] A bit confusing as the multi-site gauges are used to calculate the average rainfall over the catchments, and "sites" are later mentioned in the context of the catchments.*

**Response 7:**

To avoid confusion we have removed the word 'site' from this sentence and added clarifying text to explain that the observed data from the multiple gauges in the Onkaparinga catchment were used to calculate average annual rainfall for the purpose of hydrological model calibration only. The modified text is reproduced below:

> Page 14, Line 5: '*The observed data from the multiple rainfall gauges in the Onkaparinga catchment were Thiessen weighted to calculate the catchment average rainfall, which was used purely for the purposes of calibrating the hydrological model. The hydrological model was calibrated to the daily streamflow data at Houlgrave Weir (see Fig. 5) using a model calibration period of 15 years (1985-1999).*'

In response to Comment #3 above we have now provided a clear definition of a 'site', which is simply the point location of a rainfall gauge where we want to evaluate the SRM.

We believe this clarifies any confusion between the observed rainfall data used for hydrological model calibration and the observed rainfall data from multiple sites used for other purposes in this paper.

**Comment 8:**

*[Fig. 7] Please add the sd and m abbreviations in the figure caption.*

**Response 8:**

Fixed. The 'sd' and 'm' abbreviations are now defined in the caption of Figure 7.

**Comment 9:**

*Choose of hydrological model – in the text it is mentioned that "it is also important that the selected hydrological model is fit for purpose so that it can simulate the streamflow characteristics of interest" [6 2-3], "the hydrological model should be selected on the basis that it is capable of simulating streamflow for the timescales, magnitudes and physical processes of interest to the intended application" [7 2-4] and the benefits of using different hydrological models than the one presented here are discussed in 5.3i. However, it should also be mentioned in the text that the "right" hydrological model in terms of representing the rainfall properties should be chosen. For example, no point of using a distributed hydrological model if multi-site rainfall locations are examined, and on the other hand, no point of using a lumped model that averages 15 rainfall locations in a single site...*

**Response 9:**

Following your suggestion, we have added text to clarify that there are further requirements that should be considered in the selection of an appropriate hydrological model, including the compatibility of the rainfall model and hydrological model.

> Page 6, Line 3: *'It is also important that the selected hydrological model is fit for purpose so that it can simulate the streamflow characteristics of interest. This selection process should also ensure that the hydrological model is compatible with tested the rainfall model and the objectives of the test (i.e. a distributed hydrological model would not be selected to evaluate a single-site rainfall model for deficiencies).'*

**Response to Reviewer 1**

**Comment 1:**

*From my point of view the manuscript can be published without further modifications. The authors addressed all my comments and answered all my questions in a sufficient way.*

*-Hannes Müller-Thomy*

**Response 1:**

[revised manuscript text omitted]